# Approximation Bounds for Transformer Networks with Application to Regression

**Yuling Jiao** [1 2]  **Yanming Lai** [3]  **Defeng Sun** [3]  **Yang Wang** [4]  **Bokai Yan** [5]

## Abstract

We develop approximation and statistical theory for standard Transformer networks in sequence modeling. Given a sequence-to-sequence target on $[0,1]^{d_x \times n}$ whose entries are $\gamma$-Hölder for $\gamma \in (0,1]$ or belong to a first-order Sobolev class, we establish explicit $L^p$-approximation bounds for all $p \in [1, \infty]$, including the previously elusive endpoint $p = \infty$ under softmax attention. In particular, achieving error $\varepsilon$ in $L^p$-norm requires $\mathcal{O}(\varepsilon^{-d_x n/\gamma})$ parameters for $\gamma$-Hölder targets and $\mathcal{O}(\varepsilon^{-d_x n})$ parameters for Sobolev targets, matching the best known scalings in ambient dimension $d_x n$. We further study nonparametric regression with sequential and dependent observations using Transformer networks. Assuming stationary $\beta$-mixing covariates whose temporal dependence weakens over time, we analyze a sliding-window empirical risk minimization procedure and establish excess-risk guarantees for the Transformer-based estimators. Our analysis clarifies the role of attention and enables extensions beyond softmax.

## 1. Introduction

Transformers (Vaswani et al., 2017) are widely used in modern machine learning, yet their approximation and statistical properties remain far from fully understood. This paper focuses on two questions in approximation theory and nonparametric regression: (i) what sequence-to-sequence relationship can Transformers approximate efficiently, and (ii)

Authors are listed in alphabetical order. [1]School of Artificial Intelligence, Wuhan University, Wuhan, China [2]Hubei Key Laboratory of Computational Science, Wuhan University, Wuhan, China [3]Department of Applied Mathematics, The Hong Kong Polytechnic University, Hong Kong, China [4]Department of Mathematics, The University of Hong Kong, Hong Kong, China [5]Department of Mathematics, The Hong Kong University of Science and Technology, Hong Kong, China. Correspondence to: Bokai Yan <byanac@connect.ust.hk>.

*Proceedings of the 43$^{rd}$ International Conference on Machine Learning*, Seoul, South Korea. PMLR 306, 2026. Copyright 2026 by the author(s).

what statistical rates can be achieved in regression when the observations are generated from a dependent sequence.

A central question in the theory of Transformers is their expressive capacity, which refers to their ability to effectively approximate target functions. The study of expressivity for neural networks dates back to the 1980s, when the universal approximation property was established for feed-forward neural networks (FNNs) (Cybenko, 1989; Hornik et al., 1989). More recently, a large body of work has moved beyond universality and quantified approximation rates for structured function classes, most notably for ReLU FNNs and related architectures; see, e.g., Yarotsky (2017), Shen et al. (2020), Lu et al. (2021) and Jiao et al. (2023) for deep FNNs, Yang & Zhou (2024a), Yang & Zhou (2024b) and Siegel (2025) for shallow FNNs, and Jiao et al. (2024b) for RNNs. In contrast, approximation theory for Transformer architectures is comparatively less developed, partly because attention introduces more intricate, highly nonlinear interactions and enables substantially richer sequence-to-sequence mappings than coordinate-wise feed-forward models. Representative results for Transformers include universality in $L^p$-norms for $1 \le p < \infty$ (Yun et al., 2020; Kajitsuka & Sato, 2024), qualitative uniform approximation of continuous in-context mappings over probability-measure contexts (Furuya et al., 2025), analyses for specialized settings such as infinite-dimensional inputs (Takakura & Suzuki, 2023) or restricted target families (Takeshita & Imaizumi, 2025), and several recent attempts to quantify approximation rates via architectural constraints (Xu & Sato, 2025; Jiang & Li, 2024). Another related line of work replaces the softmax attention with more tractable alternatives to facilitate analysis (Gurevych et al., 2022; Jiao et al., 2024a; Havrilla & Liao, 2024). Despite these advances, sharp approximation rates for general target functions, especially in the $L^\infty$-norm, for the standard softmax-activated Transformer architecture remain largely open. This paper aims to fill this gap by establishing $L^\infty$-norm approximation guarantees for Transformers.

A second theme concerns Transformer performance in sequence modeling, namely, how Transformer captures dependencies and interactions within sequential data. Recent theoretical work has approached this question from sev-

eral complementary angles: inductive biases and sample complexity of attention mechanisms (Edelman et al., 2022), architectural constructions enabling exact recognition of formal languages and long-range dependencies (Chiang & Cholak, 2022), interpretations of attention and weight matrices through the lens of associative memory (Bietti et al., 2023), and universality results for prompted or prefix-tuned Transformers (Petrov et al., 2024; Hu et al., 2025). Related analyses further quantify how architectural choices (e.g., depth, number of heads, and feed-forward width) affect performance in tasks requiring large and sparse memories (Wang & E, 2024). On the statistical side, most existing nonparametric regression theory for neural networks is developed under the simplifying assumption of i.i.d. observations (Suzuki, 2019; Schmidt-Hieber, 2020; Nakada & Imaizumi, 2020; Kohler & Langer, 2021; Farrell et al., 2021; Jiao et al., 2023; Yang & Zhou, 2024a; He et al., 2026). Sequential data, however, typically exhibit temporal dependence, making the i.i.d. model too restrictive. A growing line of work therefore studies regression under stationary mixing processes, where dependence decays over time (Feng et al., 2023; Ren et al., 2024; Jiao et al., 2024b; 2025). Despite these developments, the statistical theory for Transformer-based regression with dependent observations, a setting directly aligned with the design goal of Transformers, remains comparatively under-explored. This paper aims to address this gap by analyzing Transformer estimators under temporal dependence.

In this paper, we study approximation guarantees for Transformer networks over Hölder and Sobolev classes, and derive statistical rates for nonparametric regression with dependent observations. Our main contributions are as follows:

- **Approximation bounds for standard softmax-activated Transformers.** We establish explicit upper bounds for approximating Hölder- and Sobolev-smooth sequence-to-sequence mappings by standard (softmax-activated) Transformer architectures. In particular, consider a target mapping whose component functions are $\gamma$-Hölder with $\gamma \in (0, 1]$. To achieve approximation error at most $\varepsilon$ in $L^p$-norm for any $p \in [1, \infty]$, it suffices to use a Transformer whose total number of parameters scales as $\varepsilon^{-d_x n/\gamma}$. This extends existing approximation results by providing an explicit rate and, crucially, covering the endpoint case $p = \infty$. We also derive analogous bounds for Sobolev targets.

- **Nonparametric regression under temporal dependence.** We develop a unified excess-risk analysis for Transformer-based regression trained via a sliding-window scheme on weakly dependent sequences. Up to logarithmic factors, we obtain the rate $m^{-\frac{\gamma}{\gamma + d_x n}}$ under geometrically $\beta$-mixing dependence, the rate $m^{-\frac{r\gamma}{(r+2)\gamma + (r+1)d_x n}}$ under algebraically $\beta$-mixing dependence, and the (i.i.d.) benchmark rate $m^{-\frac{\gamma}{\gamma + d_x n}}$,

where $m$ is the sample size and $r$ controls the strength of dependence. In addition, we provide sample-complexity upper bounds for Transformer estimators without imposing explicit constraints on weight magnitudes.

- **A Kolmogorov-Arnold-inspired proof strategy and beyond-softmax attention.** We introduce a proof approach inspired by the Kolmogorov-Arnold representation theorem. A key observation in our analysis is that the self-attention layers effectively reduce to column-wise averaging operators, which allows us to decouple approximation from specific properties of the softmax. As a consequence, our arguments extend to a broader class of attention mechanisms obtained by replacing softmax with more general alternatives, offering a new, interpretable viewpoint on self-attention.

*Organization.* The rest of the paper is organized as follows. In Section 2, we define the Transformer architecture, describe the setup of the problem, and list our main results. In Section 3, we present discussions and related works. In Section 4, we conclude and discuss future directions. All proofs are provided in Appendix A.

## 2. Main Results

*Notation.* We use bold lowercase letters to represent vectors and bold uppercase letters to represent matrices. For any vector $\boldsymbol{v} \in \mathbb{R}^d$, we denote by $v_i$ the $i$-the element of $\boldsymbol{v}$. For any matrix $\boldsymbol{A} \in \mathbb{R}^{d \times n}$, we denote its $i$-th row by $\boldsymbol{A}_{i,:}$, its $j$-th column by $\boldsymbol{A}_{:,j}$ and the element at its $i$-th row and $j$-th column by $A_{i,j}$. We denote the all-zero and all-one vectors of length $n$ by $\boldsymbol{0}_n$ and $\boldsymbol{1}_n$, respectively. The identity matrix of size $n$ is denoted by $\boldsymbol{I}_n$. The zero matrix of size $m \times n$ is denoted by $\boldsymbol{O}_{m,n}$. When the dimensions are clear from the context, we omit the subscripts for brevity. For $m \in \mathbb{N}$, we write $[m] := \{1, \ldots, m\}$. We use $\mathbb{N}_0$ to denote the set of nonnegative integers and $\mathbb{N}_0^d = \{(\alpha_1, \alpha_2, \ldots, \alpha_d) : \alpha_k \in \mathbb{N}_0, \forall k \in [d]\}$ to denote the set of $d$-dimensional multi-index. For a multi-index $\boldsymbol{\alpha} \in \mathbb{N}_0^d$, we denote $\|\boldsymbol{\alpha}\|_1 = \alpha_1 + \alpha_2 + \cdots + \alpha_d$. For a finite set $\mathbb{G}$, we use $|\mathbb{G}|$ to denote its cardinality. For two sequences $\{a_n\}$ and $\{b_n\}$, we use the notation $a_n \lesssim b_n$ and $a_n \gtrsim b_n$ to indicate $a_n \leq c_1 b_n$ and $a_n \geq c_2 b_n$, respectively, for some constants $c_1, c_2 > 0$ that are independent of $n$. Furthermore, $a_n \asymp b_n$ means that both $a_n \lesssim b_n$ and $a_n \gtrsim b_n$ hold. In our analysis, we use $\sigma_S$ and $\sigma_H$ to denote the column-wise softmax and hardmax functions, respectively. Specifically, for a matrix $\boldsymbol{A} \in \mathbb{R}^{d \times n}$, $\sigma_S[\boldsymbol{A}] \in \mathbb{R}^{d \times n}$ is defined by $\sigma_S[\boldsymbol{A}]_{i,j} := \frac{\exp(A_{i,j})}{\sum_{k=1}^d \exp(A_{k,j})}$, and $\sigma_H[\boldsymbol{A}] \in \mathbb{R}^{d \times n}$ is defined by $\sigma_H[\boldsymbol{A}]_{i,j} := \frac{\mathbb{1}\{A_{i,j} = \max_{k \in [d]} A_{k,j}\}}{\sum_{\ell=1}^d \mathbb{1}\{A_{\ell,j} = \max_{k \in [d]} A_{k,j}\}}$. The ReLU activation function is denoted by $\sigma_R[x] := \max\{x, 0\}$. In

contrast to $\sigma_S$ and $\sigma_H$, $\sigma_R$ operates element-wise, regardless of whether the input is a vector or a matrix. Let $\Omega \subseteq \mathbb{R}^{d \times n}$ be a bounded domain. For $1 \leq p < \infty$, the $L^p$-norm of a real-valued function $f : \mathbb{R}^{d \times n} \to \mathbb{R}$ is defined as $\|f\|_{L^p(\Omega)} := (\int_\Omega |f(\boldsymbol{X})|^p \, d\boldsymbol{X})^{1/p}$, and for $p = \infty$, it is given by $\|f\|_{L^\infty(\Omega)} := \operatorname{ess\,sup}_{\boldsymbol{X} \in \Omega} |f(\boldsymbol{X})|$. For a matrix-valued function $\boldsymbol{F} : \mathbb{R}^{d \times n} \to \mathbb{R}^{m \times n}$, the $L^p$-norm is defined as $\|\boldsymbol{F}\|_{L^p(\Omega)} := (\int_\Omega \|\boldsymbol{F}(\boldsymbol{X})\|_F^p \, d\boldsymbol{X})^{1/p}$ for $1 \leq p < \infty$, and for $p = \infty$, $\|\boldsymbol{F}\|_{L^\infty(\Omega)} := \operatorname{ess\,sup}_{\boldsymbol{X} \in \Omega} \|\boldsymbol{F}(\boldsymbol{X})\|_F$.

## 2.1. Approximation Rates for Hölder and Sobolev Functions

We begin by introducing the architecture of Transformers, following the notations in Kim et al. (2023) and Kajitsuka & Sato (2025). A Transformer network is in general a sequence-to-sequence function $\mathbb{R}^{d_x \times n} \to \mathbb{R}^{d_y \times n}$, comprising three main components: the self-attention layer, the (token-wise) feed-forward layer, and the embedding layer.

**Embedding and projection layers.** For embedding dimension $D \in \mathbb{N}$, the embedding and projection layers connect the input, hidden, and output spaces. The embedding layer $\mathcal{E}_{in} : \mathbb{R}^{d_x \times n} \to \mathbb{R}^{D \times n}$ is defined as

$$\mathcal{E}_{in}(\boldsymbol{X}) := \boldsymbol{E}_{in}\boldsymbol{X} + \boldsymbol{P} \in \mathbb{R}^{D \times n},$$

where $\boldsymbol{E}_{in} \in \mathbb{R}^{D \times d_x}$ is a learnable weight matrix, and $\boldsymbol{P} \in \mathbb{R}^{D \times n}$ is a trainable positional encoding matrix. Since self-attention and feed-forward layers are permutation equivariant, $\boldsymbol{P}$ is introduced to provide positional information and break this equivariance. The projection layer $\mathcal{E}_{out} : \mathbb{R}^{D \times n} \to \mathbb{R}^{d_x \times n}$ is defined as

$$\mathcal{E}_{out}(\boldsymbol{Y}) := \boldsymbol{E}_{out}\boldsymbol{Y} \in \mathbb{R}^{d_y \times n},$$

where $\boldsymbol{E}_{out} \in \mathbb{R}^{d_y \times D}$ maps the high-dimensional hidden representation onto the output space.

**Self-attention layer.** Given a sequence $\boldsymbol{Z} \in \mathbb{R}^{D \times n}$, composed of $n$ tokens, each with an embedding dimension $D$, the $l$-th self-attention layer $\mathcal{F}_l^{(SA)} : \mathbb{R}^{D \times n} \to \mathbb{R}^{D \times n}$ is defined as

$$\mathcal{F}_l^{(SA)}(\boldsymbol{Z}) := \boldsymbol{Z} + \sum_{h=1}^{H} \boldsymbol{W}_{h,l}^{(O)} \left( \boldsymbol{W}_{h,l}^{(V)} \boldsymbol{Z} \right) \sigma_S \left[ \left( \boldsymbol{W}_{h,l}^{(K)} \boldsymbol{Z} \right)^\top \left( \boldsymbol{W}_{h,l}^{(Q)} \boldsymbol{Z} \right) \right],$$

where $\boldsymbol{W}_{h,l}^{(V)}, \boldsymbol{W}_{h,l}^{(K)}, \boldsymbol{W}_{h,l}^{(Q)} \in \mathbb{R}^{S \times D}$ and $\boldsymbol{W}_{h,l}^{(O)} \in \mathbb{R}^{D \times S}$ are the value, key, query, and projection matrices for head $h \in [H]$ with head size $S$, respectively.

**Feed-forward layer.** The output $\boldsymbol{Z} \in \mathbb{R}^{D \times n}$ from the previous self-attention layer is then passed to the feed-forward layer, computed by

$$\mathcal{F}_l^{(FF)}(\boldsymbol{Z}) := \boldsymbol{Z} + \boldsymbol{W}_l^{(2)} \sigma_R \left[ \boldsymbol{W}_l^{(1)} \boldsymbol{Z} + \boldsymbol{b}_l^{(1)} \mathbf{1}_n^\top \right] + \boldsymbol{b}_l^{(2)} \mathbf{1}_n^\top,$$

where $\boldsymbol{W}_l^{(1)} \in \mathbb{R}^{W \times D}$ and $\boldsymbol{W}_l^{(2)} \in \mathbb{R}^{D \times W}$ are weight matrices with hidden dimension $W$, and $\boldsymbol{b}_l^{(1)} \in \mathbb{R}^W$, $\boldsymbol{b}_l^{(2)} \in \mathbb{R}^D$ are bias terms.

The class of Transformer networks is then defined as

$$\mathcal{T}_{d_x,d_y}(D, H, S, W, L) := \left\{ \mathcal{E}_{out} \circ \mathcal{F}_L^{(FF)} \circ \mathcal{F}_L^{(SA)} \circ \cdots \circ \mathcal{F}_1^{(FF)} \circ \mathcal{F}_1^{(SA)} \circ \mathcal{E}_{in} \right\},$$

where $D$ is the embedding dimension, $H$ is the number of attention heads, $S$ is the head size, $W$ is the hidden dimension in the feed-forward layer, and $L$ is the number of Transformer layers, each consisting of a self-attention and a feed-forward sublayer. We list some basic properties of the Transformer class in Proposition A.2. Let

$$\begin{aligned} N = {} & Dd_x + Dn + d_yD \\ & + L\left(4HSD + 2WD + W + D\right) \qquad (1) \\ & \lesssim (HS + W)DL \end{aligned}$$

be the total number of training parameters in the Transformer network.

We now define the Hölder and Sobolev balls on $\Omega$ that will be used throughout. The domain $\Omega$ will typically be taken as the unit cube $[0, 1]^{d_x \times n}$ in this paper.

**Definition 2.1** ($\gamma$-Hölder functions). Let $\Omega$ be a bounded domain in $\mathbb{R}^{d_x \times n}$ and $\gamma \in (0, 1]$. Given $K_\mathcal{H} > 0$, we denote the Hölder class $\mathcal{H}^\gamma(\Omega, K_\mathcal{H})$ as

$$\mathcal{H}^\gamma(\Omega, K_\mathcal{H}) := \left\{ f : \Omega \to \mathbb{R} : \right.$$

$$\left. \|f\|_{L^\infty(\Omega)} + \sup_{\boldsymbol{X}, \boldsymbol{Y} \in \Omega, \boldsymbol{X} \neq \boldsymbol{Y}} \frac{|f(\boldsymbol{X}) - f(\boldsymbol{Y})|}{\|\boldsymbol{X} - \boldsymbol{Y}\|_F^\gamma} \leq K_\mathcal{H} \right\}.$$

**Definition 2.2** (Sobolev functions). Let $\Omega$ be a bounded domain in $\mathbb{R}^{d_x \times n}$. For $p \in [1, \infty)$ and $K_\mathcal{W} > 0$, we denote the Sobolev class $\mathcal{W}^{1,p}(\Omega, K_\mathcal{W})$ as

$$\mathcal{W}^{1,p}(\Omega, K_\mathcal{W})$$

$$:= \left\{ f : \Omega \to \mathbb{R} : \left( \sum_{\|\boldsymbol{\alpha}\|_1 \leq 1} \int_\Omega |D^{\boldsymbol{\alpha}} f|^p \, d\boldsymbol{X} \right)^{1/p} \leq K_\mathcal{W} \right\},$$

and for $p = \infty$, the Sobolev class $\mathcal{W}^{1,\infty}(\Omega, K_\mathcal{W})$ is defined as

$$\mathcal{W}^{1,\infty}(\Omega, K_\mathcal{W})$$

$$:= \left\{ f : \Omega \to \mathbb{R} : \sum_{\|\boldsymbol{\alpha}\|_1 \leq 1} \operatorname{ess\,sup}_\Omega |D^{\boldsymbol{\alpha}} f| \leq K_\mathcal{W} \right\},$$

where $\boldsymbol{\alpha} \in \mathbb{N}_0^{d_x \times n}$ is a multi-index and $D^{\boldsymbol{\alpha}}$ is the weak derivative of order $\boldsymbol{\alpha}$.

Hölder and Sobolev classes are standard smoothness models in approximation theory, closely tied to polynomial and spline approximation (DeVore, 1998). They are also related through classical embedding and interpolation results. Under standard regularity assumptions on $\Omega$ (and in particular when $p > d$), the Sobolev embedding yields $\mathcal{W}^{1,p}(\Omega, K_{\mathcal{W}}) \subseteq \mathcal{H}^{1-\frac{d}{p}}(\Omega, K_{\mathcal{H}})$ for a suitable $K_{\mathcal{H}}$ depending on $K_{\mathcal{W}}$ and the geometry of $\Omega$, and in the endpoint case $p = \infty$ the Sobolev space coincides with the Lipschitz space on $\Omega$ (up to norm equivalence), so that $\mathcal{W}^{1,\infty}(\Omega, K_{\mathcal{W}}) \subseteq \mathcal{H}^1(\Omega, K_{\mathcal{H}})$ and $\mathcal{H}^1(\Omega, K_{\mathcal{H}}) \subseteq \mathcal{W}^{1,\infty}(\Omega, K_{\mathcal{W}})$ with constants depending at most on $\Omega$.

We now present our main results on the approximation capabilities of Transformer networks for Hölder and Sobolev functions. We defer the proofs to Appendix A.1.

**Theorem 2.3.** *Given* $\gamma \in (0, 1]$ *and* $K_{\mathcal{H}} > 0$, *assume that the target function* $\boldsymbol{F} : [0, 1]^{d_x \times n} \to \mathbb{R}^{d_x \times n}$ *satisfies* $F_{i,j} \in \mathcal{H}^\gamma([0,1]^{d_x \times n}, K_{\mathcal{H}})$ *for each* $i \in [d_x], j \in [n]$. *For any* $\varepsilon \in (0, 1)$ *and* $p \in [1, \infty]$, *there exists a Transformer network* $\mathcal{N} \in \mathcal{T}_{d_x, d_x}(D = C_1, H = C_2, S = C_3, W = C_4 \cdot \lceil \varepsilon^{-\frac{d_x n}{\gamma}} \rceil, L = C_5)$ *such that*

$$\|\mathcal{N} - \boldsymbol{F}\|_{L^p([0,1]^{d_x \times n})} \leq 4(d_x n)^2 K_{\mathcal{H}} \varepsilon,$$

*where*

1. $C_1 = d_x$, $C_2 = 1$, $C_3 = 1$, $C_4 = 5n$ *and* $C_5 = 2$ *if* $p \in [1, \infty)$;

2. $C_1 = 5d_x 3^{d_x n}$, $C_2 = 3^{d_x n}$, $C_3 = 1$, $C_4 = 5n3^{d_x n}$ *and* $C_5 = 2 + 2d_x n$ *if* $p = \infty$.

**Theorem 2.4.** *Given* $p \in [1, \infty)$ *and* $K_{\mathcal{W}} > 0$, *assume that the target function* $\boldsymbol{F} : [0, 1]^{d_x \times n} \to \mathbb{R}^{d_x \times n}$ *satisfies* $F_{i,j} \in \mathcal{W}^{1,p}([0,1]^{d_x \times n}, K_{\mathcal{W}})$ *and* $\|F_{i,j}\|_{L^\infty([0,1]^{d_x \times n})} \leq K_{\mathcal{W}}$ *for each* $i \in [d_x], j \in [n]$. *For any* $\varepsilon \in (0, 1)$, *there exists a Transformer network* $\mathcal{N} \in \mathcal{T}_{d_x, d_x}(D = d_x, H = 1, S = 1, W = 5n \cdot \lceil \varepsilon^{-d_x n} \rceil, L = 2)$ *such that*

$$\|\mathcal{N} - \boldsymbol{F}\|_{L^p([0,1]^{d_x \times n})} \leq 4CK_{\mathcal{W}} \varepsilon,$$

*where* $C$ *is a constant depending only on* $d_x n$.

We make several remarks regarding our results. First, ever since Vaswani et al. (2017) proposed the Transformer architecture, there have been various theoretical analyses on its expressive capacity. A series of works established the universal approximation for sequence-to-sequence continuous functions in the $L^p$-norm for $1 \leq p < \infty$. More precisely, Yun et al. (2020) showed that

$$\sup_{\boldsymbol{F}: F_{i,j} \in \mathcal{C}(\Omega)} \inf_{\mathcal{N} \in \mathcal{T}(D,H,S,W,L)} \|\mathcal{N} - \boldsymbol{F}\|_{L^p(\Omega)} \to 0$$

for fixed and sufficiently large $D, H, S, W$ and as $L \to \infty$, where $\mathcal{C}(\Omega)$ denotes the space of continuous functions on

a bounded domain $\Omega$, and subsequently Kajitsuka & Sato (2024) showed that

$$\sup_{\boldsymbol{F}: F_{i,j} \in \mathcal{C}(\Omega)} \inf_{\mathcal{N} \in \mathcal{T}(D,H,S,W,L)} \|\mathcal{N} - \boldsymbol{F}\|_{L^p(\Omega)} \to 0$$

for fixed and sufficiently large $D, H, S, L$ and as $W \to \infty$. Theorems 2.3 and 2.4 give that for all $1 \leq p \leq \infty$,

$$\sup_{\boldsymbol{F}: F_{i,j} \in \mathcal{H}^\gamma(\Omega, K_{\mathcal{H}})} \inf_{\mathcal{N} \in \mathcal{T}(D,H,S,W,L)} \|\mathcal{N} - \boldsymbol{F}\|_{L^p(\Omega)}$$
$$\lesssim K_{\mathcal{H}} W^{-\gamma/(d_x n)}$$

and for $1 \leq p < \infty$,

$$\sup_{\substack{\boldsymbol{F}: F_{i,j} \in \mathcal{W}^{1,p}(\Omega, K_{\mathcal{W}}), \\ \|F_{i,j}\|_{L^\infty(\Omega)} \leq K_{\mathcal{W}}}} \inf_{\mathcal{N} \in \mathcal{T}(D,H,S,W,L)} \|\mathcal{N} - \boldsymbol{F}\|_{L^p(\Omega)}$$
$$\lesssim K_{\mathcal{W}} W^{-1/(d_x n)},$$

for fixed $D, H, S, L$ and adjustable $W$. Our results not only provide explicit approximation rates for general Hölder and Sobolev functions, but also extend to the case $p = \infty$, which previous methods were unable to address. These improvements are largely attributed to the use of the horizontal shift technique, which was originally introduced by Lu et al. (2021) and further developed in Shen et al. (2020) and Zhang et al. (2022). We summarize the most related results in Table 1.

Second, we would like to remark that our approximation results are established in a sequence-to-sequence sense; that is, every entry of the Transformer network $\mathcal{N}$ simultaneously approximates the corresponding entry of the matrix-valued target function $\boldsymbol{F}$. It is not hard to extend the target function $\boldsymbol{F} : [0, 1]^{d_x \times n} \to \mathbb{R}^{d_y \times n}$ to general output dimension $d_y \in \mathbb{N}$ in Theorems 2.3 and 2.4.

Our results further demonstrate that Transformer networks have stronger expressive power than RNNs in approximating sequence-to-sequence functions. As discussed in hoon Song et al. (2023) and Jiao et al. (2024b), RNNs are inherently limited to approximating past-dependent sequence-to-sequence functions because, at each time step, only the current and past tokens are utilized, leaving future tokens unprocessed. In contrast, Transformers have the advantage of accessing the entire input sequence. In other words, even the first output token of a Transformer depends on the entire input sequence, whereas in an RNN the first output token depends only on the first input token, the second on the first two, and so forth, owing to the sequential nature of RNNs. This distinction underpins our assertion that Transformer architectures outperform RNNs in terms of expressive power.

Third, we observe that achieving an approximation error $\varepsilon$ requires on the order of $\varepsilon^{-d_x n/\gamma}$ trainable parameters for $\gamma$-Hölder targets, and on the order of $\varepsilon^{-d_x n}$ parameters for

Sobolev targets. These scalings match the previously best known upper bounds for fixed-depth FNNs and RNNs in ambient input dimension $d_x n$ (Yarotsky, 2017; Shen et al., 2020; Lu et al., 2021; Jiao et al., 2023; Siegel, 2023; Jiao et al., 2024b).

## 2.2. Nonparametric Regression

We then study the regression problem, which seeks to estimate an unknown target regression function from finite observations. We consider the following $n$-step prediction model

$$Y = f^*(X_1, X_2, \ldots, X_n) + \varepsilon, \tag{2}$$

where $Y \in \mathbb{R}$ is a response, $f^*(\boldsymbol{x}_1, \ldots, \boldsymbol{x}_n) = \mathbb{E}[Y | X_1 = \boldsymbol{x}_1, \ldots, X_n = \boldsymbol{x}_n] : [0,1]^{d_x \times n} \to \mathbb{R}$ is an unknown regression function and $\varepsilon$ is a sub-Gaussian noise, independent of $X_i, i = 1, \ldots, n$, with $\mathbb{E}[\varepsilon] = 0$ and

$$\mathbb{E}[\exp(s\varepsilon)] \leq \exp\left(\frac{\sigma^2 s^2}{2}\right) \quad \text{for any } s \in \mathbb{R}.$$

Our purpose is to estimate the unknown target regression function $f^*$ given observations $\mathcal{D}_m = \{(\boldsymbol{x}_1, y_1), \ldots, (\boldsymbol{x}_m, y_m)\}$ which may not be i.i.d.

As observed in real sequence modeling applications, the sequential observations often exhibit temporal dependence, rendering the usual i.i.d. assumption inapplicable. This motivates us to consider dependent data. A frequently used alternative is to assume that observations are drawn from a stationary mixing distribution, where the dependence between observations weakens over time. This scenario has become standard and has been discussed extensively in previous studies (Yu, 1994; Meir, 2000; Mohri & Rostamizadeh, 2008; Steinwart & Christmann, 2009; Mohri & Rostamizadeh, 2010; Agarwal & Duchi, 2013; Shalizi & Kontorovich, 2013; Kuznetsov & Mohri, 2017; Ren et al., 2024). We now introduce the relevant definitions.

**Definition 2.5** (Stationarity). A sequence of random variables $\{\boldsymbol{x}_t\}_{t=-\infty}^{\infty}$ is said to be stationary if for any $t$ and non-negative integers $m$ and $k$, the random vectors $(\boldsymbol{x}_t, \ldots, \boldsymbol{x}_{t+m})$ and $(\boldsymbol{x}_{t+k}, \ldots, \boldsymbol{x}_{t+m+k})$ have the same distribution.

**Definition 2.6** ($\beta$-mixing). Let $\{\boldsymbol{x}_t\}_{t=-\infty}^{\infty}$ be a stationary sequence of random variables. For any $i, j \in \mathbb{Z} \cup \{-\infty, +\infty\}$, let $\sigma_i^j$ denote the $\sigma$-algebra generated by the random variables $\boldsymbol{x}_k, i \leq k \leq j$. Then, for any positive integer $k$, the $\beta$-mixing coefficient of the stochastic process $\{\boldsymbol{x}_t\}_{t=-\infty}^{\infty}$ is defined as

$$\beta(k) = \sup_n \mathbb{E}_{B \in \sigma_{-\infty}^n} \left[ \sup_{A \in \sigma_{n+k}^\infty} |\mathbb{P}(A \mid B) - \mathbb{P}(A)| \right].$$

$\{\boldsymbol{x}_t\}_{t=-\infty}^{\infty}$ is said to be $\beta$-mixing if $\beta(k) \to 0$ as $k \to \infty$. It is said to be algebraically $\beta$-mixing if there exist real numbers $\beta_0 > 0$ and $r > 0$ such that $\beta(k) \leq \beta_0 / k^r$ for all $k$, and geometrically $\beta$-mixing if there exist real numbers $\beta_0, \beta_1 > 0$ and $r > 0$ such that $\beta(k) \leq \beta_0 \exp(-\beta_1 k^r)$ for all $k$.

In this work, we assume that the sequence of random variables $\mathcal{X} = \{\boldsymbol{x}_t\}_{t=1}^m$ is drawn from a stationary $\beta$-mixing process. By Definition 2.5, the time index $t$ does not affect the distribution of $\boldsymbol{x}_t$ in a stationary sequence. Moreover, any $n$ consecutive observations, $(\boldsymbol{x}_{t-n+1}, \ldots, \boldsymbol{x}_t)$, share the same joint distribution, which we denote by $\Pi$. We assume that $\Pi$ is supported on $[0,1]^{d_x \times n}$ and is absolutely continuous with respect to the Lebesgue measure, with its probability density function uniformly bounded by a finite constant on $[0,1]^{d_x \times n}$. Definition 2.6 states that a sequence of random variables is mixing if the influence of past events on future events diminishes as the temporal gap increases. This definition provides a standard measure of the dependence among the random variables $\{\boldsymbol{x}_t\}$ within a stationary sequence. We note that in certain special cases, such as Markov chains, the mixing coefficients admit upper bounds that can be estimated from data (Hsu et al., 2015). If $\{\boldsymbol{x}_t\}_{t=1}^m$ are i.i.d. random variables, then by definition, $\beta(k) = 0$ for all $k$.

A standard approach to estimating $f^*$ is to minimize the population risk. Define

$$\mathcal{R}(f) := \mathbb{E}_{(X_1, \ldots, X_n) \sim \Pi, \, Y}[(f(X_1, \ldots, X_n) - Y)^2].$$

It is known that the risk $\mathcal{R}(f)$ is minimized by the regression function $f^*$ given the assumption that $\mathbb{E}[\varepsilon | X_1, \ldots, X_n] = 0$, i.e.,

$$f^* \in \arg\min_f \mathcal{R}(f).$$

However, in practice the joint distribution of $((X_1, \ldots, X_n), Y)$ is typically unknown, and only a random sample $\mathcal{D}_m = \{(\boldsymbol{x}_i, y_i)\}_{i=1}^m$ is available, where $m$ denotes the sample size. Given that each evaluation of the Transformer requires a sequence of length $n$, we consider a sliding window training approach. Specifically, we estimate $f^*$ using the empirical risk minimizer

$$\hat{f}_m \in \arg\min_{f \in \mathcal{F}} \mathcal{R}_m(f) \tag{3}$$

$$\mathcal{R}_m(f) := \frac{1}{m-n+1} \sum_{t=n}^m (f(\boldsymbol{x}_{t-n+1}, \ldots, \boldsymbol{x}_t) - y_t)^2,$$

where we choose the hypothesis class

$$\mathcal{F} = \mathcal{F}(D_m, H_m, S_m, W_m, L_m)$$
$$= \{\langle \mathcal{N}(X), \boldsymbol{E} \rangle : \mathcal{N} \in \mathcal{T}_{d_x, d_x}(D_m, H_m, S_m, W_m, L_m)\}.$$

*Table 1.* Comparison of approximation rates.

| Reference | Type | Target Function[1] | Metric in $L^p$-norm | Activation in Self-Attention Layer[2] | Width[3] | Depth[4] |
|---|---|---|---|---|---|---|
| Yun et al. (2020) | Universality | $\mathcal{C}^0$ | $p \in [1, \infty)$ | $\sigma_S$ with bias | | |
| Kajitsuka & Sato (2024) | Universality | $\mathcal{C}^0$ | $p \in [1, \infty)$ | $\sigma_S$ | | |
| Fang et al. (2022) | Universality | $\mathcal{C}^0$ | $p \in [1, \infty]$ | $\sigma_H$ | | |
| Jiao et al. (2024a) | Rate | $\mathcal{H}^\gamma$ $\mathcal{C}^m$ | $p \in [1, \infty]$ | $X \odot \sigma_H(X)$ | $\mathcal{O}(\varepsilon^{-d_x n/\gamma})$ $\mathcal{O}(\varepsilon^{-d_x n/m})$ | $\mathcal{O}(\log \frac{1}{\varepsilon})$ $\mathcal{O}(\log \frac{1}{\varepsilon})$ |
| Havrilla & Liao (2024) | Rate | $\mathcal{H}^\gamma$ | $p \in [1, \infty]$ | $\sigma_R$ | $\mathcal{O}(\varepsilon^{-d_x n/\gamma})$ | $\mathcal{O}(\log \frac{1}{\varepsilon})$ |
| **Ours** (Theorem 2.3) | Rate | $\mathcal{H}^\gamma$ | $p \in [1, \infty]$ | $\sigma_S$ | $\mathcal{O}(\varepsilon^{-d_x n/\gamma})$ | $\mathcal{O}(1)$ |
| **Ours** (Theorem 2.4) | Rate | $\mathcal{W}^{1,p}$ | $p \in [1, \infty]$ | $\sigma_S$ | $\mathcal{O}(\varepsilon^{-d_x n})$ | $\mathcal{O}(1)$ |

[1]The space $\mathcal{C}^m$ consists of all functions whose first $m$ derivatives exist and are continuous, and $\mathcal{C}^0$ denotes the space of continuous functions. [2]Different Transformer architectures are obtained by replacing the softmax function in the self-attention layer with various activation functions. The symbol $\odot$ denotes the Hadamard product. [3]Following Kim et al. (2023), the width of a Transformer network is defined as $\max\{D, HS, W\}$. We use $\mathcal{O}(\cdot)$ to omit constants independent of $\varepsilon$. [4]We denote by $L$ the depth of the Transformer.

Here, $\langle \cdot, \cdot \rangle$ denotes the matrix inner product, and $\boldsymbol{E} \in \mathbb{R}^{d_x \times n}$ is a trainable weight matrix. The performance of the estimator is evaluated by the excess risk, defined as the difference between the population risks of $\hat{f}_m$ and $f^*$, given by

$$\mathcal{R}(\hat{f}_m) - \mathcal{R}(f^*)$$
$$= \mathbb{E}_{(X_1, \ldots, X_n)}[(\hat{f}_m(X_1, \ldots, X_n) - f^*(X_1, \ldots, X_n))^2].$$

To control the sample complexity, it is often required that the hypothesis class is uniformly bounded. We define the truncation operator $\mathcal{C}_B$ with level $B > 0$ for a real-valued function $f$ as

$$\mathcal{C}_B f(x) := \begin{cases} f(x) & \text{if } |f(x)| \leq B, \\ \operatorname{sgn}(f(x)) \cdot B & \text{if } |f(x)| > B. \end{cases}$$

For a class of real-valued functions $\mathcal{F}$, we use the notation $\mathcal{C}_B \mathcal{F} := \{\mathcal{C}_B f : f \in \mathcal{F}\}$. Note that the truncation can be implemented by a feed-forward layer that applies the operation $\sigma_R[x] - \sigma_R[-x] - \sigma_R[x - B] + \sigma_R[-x - B]$ element-wise. Our next theorem provides a rate of convergence for estimating the target function $f^*$ using the truncated empirical risk minimizer $\mathcal{C}_{B_m} \hat{f}_m$. We defer the proof to Appendix A.2.

**Theorem 2.7.** *Under model (2), assume that the regression function $f^* \in \mathcal{H}^\gamma([0,1]^{d_x \times n}, K_{\mathcal{H}})$ for some $\gamma \in (0, 1]$ and $K_{\mathcal{H}} > 0$, and that the probability measure of the covariate*

$\Pi$ *is supported on $[0,1]^{d_x \times n}$ and is absolutely continuous with respect to the Lebesgue measure, with its density function uniformly bounded by a finite constant. Let $\hat{f}_m$ be the empirical risk minimizer defined in (3) over a random sample $\mathcal{D}_m = \{(\boldsymbol{x}_i, y_i)\}_{i=1}^m$. Then, the following bounds hold:*

*1. If $\{\boldsymbol{x}_i\}_{i=1}^m$ is a **geometrically $\beta$-mixing** sequence, i.e., $\beta(k) \leq \beta_0 \exp(-\beta_1 k^r)$ for some $r, \beta_0, \beta_1 > 0$, then by choosing $B_m \asymp \log m$ and the hypothesis class $\mathcal{F}(D_m \asymp 1, H_m \asymp 1, S_m \asymp 1, W_m \asymp m^{\frac{d_x n}{2\gamma + 2d_x n}}, L_m \asymp 1)$, we have*

$$\mathbb{E}_{\mathcal{D}_m}[\mathcal{R}(\mathcal{C}_{B_m} \hat{f}_m) - \mathcal{R}(f^*)] \lesssim m^{-\frac{\gamma}{\gamma + d_x n}} (\log m)^{3 + 1/r}.$$

*2. If $\{\boldsymbol{x}_i\}_{i=1}^m$ is an **algebraically $\beta$-mixing** sequence, i.e., $\beta(k) \leq \beta_0 / k^r$ for some $r, \beta_0 > 0$, then by choosing $B_m \asymp \log m$ and the hypothesis class $\mathcal{F}(D_m \asymp 1, H_m \asymp 1, S_m \asymp 1, W_m \asymp m^{\frac{r d_x n}{2(r+2)\gamma + 2(r+1)d_x n}}, L_m \asymp 1)$, we have*

$$\mathbb{E}_{\mathcal{D}_m}[\mathcal{R}(\mathcal{C}_{B_m} \hat{f}_m) - \mathcal{R}(f^*)] \lesssim m^{-\frac{r\gamma}{(r+2)\gamma + (r+1)d_x n}} (\log m)^3.$$

*3. If $\{\boldsymbol{x}_i\}_{i=1}^m$ is a sequence of **i.i.d.** random variables, then by choosing $B_m \asymp \log m$ and the hypothesis class $\mathcal{F}(D_m \asymp 1, H_m \asymp 1, S_m \asymp 1, W_m \asymp m^{\frac{d_x n}{2\gamma + 2d_x n}}, L_m \asymp 1)$, we have*

$$\mathbb{E}_{\mathcal{D}_m}[\mathcal{R}(\mathcal{C}_{B_m} \hat{f}_m) - \mathcal{R}(f^*)] \lesssim m^{-\frac{\gamma}{\gamma + d_x n}} (\log m)^3.$$

It is well known that the optimal rate of convergence in nonparametric regression with squared loss for i.i.d. data is

*Table 2.* Comparison of convergence rates.

| Reference | Network Architecture | Dependence Assumption | Rate of Convergence[1] |
|---|---|---|---|
| Feng et al. (2023) | FNN | geometrically $\beta$-mixing | $\widetilde{\mathcal{O}}(m^{-\frac{\gamma}{2\gamma+2d+2}})$ |
| Ren et al. (2024) | FNN | geometrically $\beta$-mixing | $\widetilde{\mathcal{O}}(m^{-\frac{2\gamma}{2\gamma+d}})$ |
| Jiao et al. (2024b) | RNN | geometrically $\beta$-mixing | $\widetilde{\mathcal{O}}(m^{-\frac{2\gamma}{2\gamma+d_x n}})$ |
| | | algebraically $\beta$-mixing | $\widetilde{\mathcal{O}}(m^{-\frac{2r\gamma}{(2r+4)\gamma+(r+1)d_x n}})$ |
| | | i.i.d. | $\widetilde{\mathcal{O}}(m^{-\frac{2\gamma}{2\gamma+d_x n}})$ |
| **Ours** (Theorem 2.7) | Transformer | geometrically $\beta$-mixing | $\widetilde{\mathcal{O}}(m^{-\frac{\gamma}{\gamma+d_x n}})$ |
| | | algebraically $\beta$-mixing | $\widetilde{\mathcal{O}}(m^{-\frac{r\gamma}{(r+2)\gamma+(r+1)d_x n}})$ |
| | | i.i.d. | $\widetilde{\mathcal{O}}(m^{-\frac{\gamma}{\gamma+d_x n}})$ |

[1]We use $\widetilde{\mathcal{O}}(\cdot)$ to omit constants independent of $m$ and logarithmic factors in $m$. $d$ and $d_x n$ denote the input dimensions for vector and sequence inputs, respectively.

$m^{-2\gamma/(d_x n+2\gamma)}$ (Stone, 1982; Donoho & Johnstone, 1998), and that the same rate remains optimal for certain $\beta$-mixing sequences (Yu, 1993; Viennet, 1997). The rates in Theorem 2.7 are suboptimal. We attribute this suboptimality to the loose upper bound on the VC-dimension (see Lemma A.12). We consider the i.i.d. case for an illustration. Classical empirical process techniques yield a decomposition of the excess risk into an approximation error and a statistical error, and by trading off these two errors one obtains the optimal rate of convergence, as discussed in Jiao et al. (2023). For the approximation error, to approximate a $\gamma$-Hölder function up to accuracy $\varepsilon$, it suffices to use a ReLU FNN with a total number of adjustable parameters $N \lesssim \varepsilon^{-d/\gamma}$ (up to logarithmic factors), where $d$ denotes the input dimension and in our setting $d = d_x n$. Meanwhile, the VC-dimension, which governs the statistical error, grows linearly with $N$ (assuming fixed depth) due to the piecewise linear nature of ReLU FNNs. In fact, for FNNs with piecewise polynomial activations (such as $\text{ReLU}^k$), or for self-attention layers with piecewise polynomial activations (e.g., replacing the softmax $\sigma_S[\boldsymbol{Z}]$ with the hardmax $\sigma_H[\boldsymbol{Z}]$ (Kajitsuka & Sato, 2025) or $\boldsymbol{Z} \odot \sigma_H[\boldsymbol{Z}]$ (Gurevych et al., 2022; Jiao et al., 2024a)), the VC-dimension grows linearly in the total number of parameters $N$. However, for function classes involving exponential operations, such as those defined by sigmoid networks or radial basis function networks, the best known upper bounds on the VC-dimension grow polynomially in $N$ (Karpinski & Macintyre, 1997; Anthony & Bartlett, 1999). We use the same method to establish an upper bound on the VC-dimension of Transformer networks, and hence it exhibits quadratic growth in $N$. As noted in Bartlett & Maass (2003), there is a gap between the best known upper and lower bounds for function classes that involve exponential operations, and it remains open whether these bounds are optimal. Karpinski & Macintyre (1997) conjectured that the upper bounds are not sharp. To prove

Theorem 2.7, we decompose the excess risk into the approximation error and the statistical error. By Theorem 2.3 and (1), to achieve an approximation error of at most $\varepsilon$, it suffices to use a Transformer network with total parameters $N \lesssim \varepsilon^{-d_x n/\gamma}$. However, since the VC-dimension scales as $N^2$, it grows faster than in the ReLU FNN case, leading to the suboptimal rate after trade-off. We leave possible improvements to this gap as an open problem for future study.

We observe that, ignoring logarithmic factors, the rates of convergence for the geometrically $\beta$-mixing and i.i.d. cases are identical. In addition, for the algebraically $\beta$-mixing case the rate of convergence is given by $m^{-\frac{r\gamma}{(r+2)\gamma+(r+1)d_x n}}$, which improves as the mixing parameter $r$ increases. When $r$ is sufficiently large, this rate approaches $m^{-\frac{\gamma}{\gamma+d_x n}}$, matching that of the geometrically $\beta$-mixing and i.i.d. cases. We remark that a completely analogous result holds for estimating a Sobolev target $f^* \in \mathcal{W}^{1,p}([0,1]^{d_x \times n}, K_{\mathcal{W}})$. We summarize the most related results in Table 2.

### 2.3. Approximation by Generalized Transformer Networks

We extend the original definition of Transformer networks to enable more flexible functional representations.

**Generalized feed-forward layer.** We define the generalized feed-forward layer as

$$\mathcal{F}_l^{(GFF)}(\boldsymbol{Z}) :=$$
$$\boldsymbol{Z} + \boldsymbol{W}_l^{(2)} \sigma_R \left[ \boldsymbol{W}_l^{(1)} \boldsymbol{Z} + \boldsymbol{B}_l^{(1)} \right] + \boldsymbol{B}_l^{(2)} \in \mathbb{R}^{D \times n},$$

where $\boldsymbol{W}_l^{(1)} \in \mathbb{R}^{W \times D}$ and $\boldsymbol{W}_l^{(2)} \in \mathbb{R}^{D \times W}$ are weight matrices, and $\boldsymbol{B}_l^{(1)} \in \mathbb{R}^{W \times n}$, $\boldsymbol{B}_l^{(2)} \in \mathbb{R}^{D \times n}$ are bias matrices. In contrast to the standard feed-forward layer, we allow different bias terms for each column, thereby generalizing

the original formulation.

**Generalized self-attention layer.** We define the generalized self-attention layer as

$$\mathcal{F}_l^{(GSA)}(\boldsymbol{Z}) := \boldsymbol{Z} + \sum_{h=1}^{H} \boldsymbol{W}_{h,l}^{(O)} \sigma_G[\boldsymbol{Z}] \in \mathbb{R}^{D \times n},$$

where $\boldsymbol{W}_{h,l}^{(O)} \in \mathbb{R}^{D \times D}$ is a weight matrix with rank at most $S$ for all $h$ and $l$, and $\sigma_G[\boldsymbol{Z}]$ is a general function (may vary across different $h$ and $l$) with the only requirement that, for a particular parameter choice, it computes the column average of $\boldsymbol{Z}$, namely,

$$\sigma_G[\boldsymbol{Z}] = \left( \frac{1}{n} \sum_{j=1}^{n} \boldsymbol{Z}_{:,j}, \ldots, \frac{1}{n} \sum_{j=1}^{n} \boldsymbol{Z}_{:,j} \right).$$

Clearly, both softmax-activated self-attention (Vaswani et al., 2017)

$$\sigma_G[\boldsymbol{Z}] = \boldsymbol{Z} \cdot \sigma_S \left[ \left( \boldsymbol{W}^{(K)} \boldsymbol{Z} \right)^{\top} \left( \boldsymbol{W}^{(Q)} \boldsymbol{Z} \right) \right]$$

and (averaging) hardmax-activated self-attention (Pérez et al., 2021)

$$\sigma_G[\boldsymbol{Z}] = \boldsymbol{Z} \cdot \sigma_H \left[ \left( \boldsymbol{W}^{(K)} \boldsymbol{Z} \right)^{\top} \left( \boldsymbol{W}^{(Q)} \boldsymbol{Z} \right) \right]$$

satisfy the above definition, since they compute the column average of $\boldsymbol{Z}$ when $\boldsymbol{W}^{(K)} = \boldsymbol{W}^{(Q)} = \boldsymbol{O}$.

The class of generalized Transformer networks is then defined as

$$\mathcal{GT}_{d_x,d_y}(D, H, S, W, L) :=$$
$$\left\{ \mathcal{E}_{out} \circ \mathcal{F}_L^{(GFF)} \circ \mathcal{F}_L^{(GSA)} \circ \cdots \circ \mathcal{F}_1^{(GFF)} \circ \mathcal{F}_1^{(GSA)} \circ \mathcal{E}_{in} \right\}.$$

The following theorem provides explicit approximation bounds for Hölder continuous functions using generalized Transformer networks. We defer the proof to Appendix A.3.

**Theorem 2.8.** *Given $\gamma \in (0,1]$ and $K_{\mathcal{H}} > 0$, assume that the target function $\boldsymbol{F} : [0,1]^{d_x \times n} \to \mathbb{R}^{d_x \times n}$ satisfies $F_{i,j} \in \mathcal{H}^{\gamma}([0,1]^{d_x \times n}, K_{\mathcal{H}})$ for each $i \in [d_x], j \in [n]$. For any $\varepsilon \in (0,1)$ and $p \in [1,\infty)$, there exists a generalized Transformer network $\mathcal{N} \in \mathcal{GT}_{d_x,d_x}(D = 4d_x n, H = 1, S = d_x, W = 3d_x n \cdot \lceil \varepsilon^{-\frac{d_x n}{\gamma}} \rceil, L = 6 \lceil \frac{1}{\gamma} \log_2 \frac{1}{\varepsilon} \rceil)$, such that*

$$\| \mathcal{N} - \boldsymbol{F} \|_{L^p([0,1]^{d_x \times n})} \leq 4(d_x n)^3 K_{\mathcal{H}} \varepsilon.$$

## 3. Related Work

**Nonparametric regression using neural networks.** The rates of convergence of neural network regression estimators

have been extensively analyzed in the literature. Minimax optimal rates have been established across various neural network architectures, including under-parameterized sparse deep FNNs (Schmidt-Hieber, 2020; Suzuki, 2019), under-parameterized fully connected deep FNNs (Jiao et al., 2023), over-parameterized shallow FNNs (Yang & Zhou, 2024b;a), and RNNs (Jiao et al., 2024b). In contrast, convergence rates for Transformer-based estimators have rarely been observed. Takakura & Suzuki (2023) investigated the approximation and estimation capabilities of Transformers as sequence-to-sequence functions operating on infinite-dimensional inputs, where variable-length sliding window attention was considered. Additionally, modifications to the Transformer architecture have been explored, such as replacing the standard softmax function $\sigma_S[\boldsymbol{Z}]$ in the self-attention layer by $\boldsymbol{Z} \odot \sigma_H[\boldsymbol{Z}]$ (Gurevych et al., 2022) and by $\sigma_R[\boldsymbol{Z}]$ (Havrilla & Liao, 2024). Although these studies provide insightful constructions and analyses, their avoidance of the standard softmax function does not fully explain the successes observed ever since the introduction of the Transformer mechanism. Our result (Theorem 2.7) directly addresses this gap by analyzing convergence rates for the standard Transformer architecture explicitly using the original softmax attention. It has also been shown that neural networks are able to circumvent the curse of dimensionality under certain conditions, for example, when the intrinsic dimension of the regression function is low (Nakada & Imaizumi, 2020; Chen et al., 2022; Jiao et al., 2023; Havrilla & Liao, 2024), or the regression function has certain hierarchical structures (Schmidt-Hieber, 2020; Kohler & Langer, 2021). Exploring the conditions under which Transformers similarly mitigate the curse of dimensionality within our framework is an important direction for future research.

**Assumption on target smoothness.** Throughout this work we assume that the target function is either $\gamma$-Hölder for some $\gamma \in (0,1]$, or belongs to a first-order Sobolev class. It is an interesting open direction to develop adaptive variants of our analysis for higher-order smoothness. In the proofs of Theorems 2.3 and 2.4, we first approximate the target by a piecewise-constant function on a uniform partition, and then realize this discretized map via a Transformer that performs a lookup-type "memorization" of the grid values. We uniformly partition the unit cube $[0,1]^{d_x \times n}$ into $K^{d_x n}$ cells, whose mesh size is $h \asymp K^{-1}$; this yields the approximation orders $K^{-\gamma}$ in (4) and $K^{-1}$ in (10), which are sharp for piecewise constant approximation in general. The limitation is a manifestation of the classical saturation phenomenon: for instance, if a Lipschitz function $f$ satisfies

$$\inf_{s \in \mathcal{S}_K} \| f - s \|_{L^{\infty}([0,1]^{d_x \times n})} = o(K^{-1}) \quad \text{as } K \to \infty,$$

where $\mathcal{S}_K$ denotes the space of piecewise constant functions on the uniform $K$-partition, then $f$ must be (essentially) constant; see DeVore (1998, Section 6.2) for saturation and

inverse theorems of this type. Consequently, the piecewise constant discretization constitutes the main bottleneck in our current proofs and prevents faster rates even when the target enjoys higher regularity.

## 4. Conclusion

To conclude, this paper establishes explicit approximation guarantees for standard softmax-activated Transformers over Hölder and Sobolev classes, including $L^\infty$ bounds, and provides excess-risk guarantees for Transformer-based nonparametric regression with temporally dependent (stationary $\beta$-mixing) data. Our analysis clarifies the functional role of attention and extends naturally to broader, beyond-softmax attention mechanisms, suggesting several directions for sharper statistical bounds and adaptivity to higher-order smoothness.

## Acknowledgements

The work of Yuling Jiao was supported in part by the National Key Research and Development Program of China under Grant No. 2024YFA1014202, in part by the National Natural Science Foundation of China under Grant Nos. 12371441 and 12526216, and in part by the Fundamental Research Funds for the Central Universities. The work of Defeng Sun was supported by the Research Center for Intelligent Operations Research and RGC Senior Research Fellow Scheme No. SRFS2223-5S02. We would also like to thank the anonymous reviewers for their valuable feedback on the manuscript.

## Impact Statement

This paper presents work whose goal is to advance the field of Machine Learning. There are many potential societal consequences of our work, none which we feel must be specifically highlighted here.

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

# A. Proofs

Before proceeding, we clarify some simplifications used in the proof.

1. In a self-attention layer, if we set $\boldsymbol{W}^{(O)} = \boldsymbol{O}$, then the layer behaves as an identity mapping due to the presence of the skip connection. Similarly, in a feed-forward layer, setting $\boldsymbol{W}^{(2)} = \boldsymbol{O}$ causes the layer to degenerate into an identity mapping. Therefore, as long as identity mappings are appropriately introduced, the composition of multiple self-attention layers or feed-forward layers remains consistent with our definition of a Transformer.

2. Since a feed-forward layer applies the same operation to each column of the input matrix, we do not distinguish between matrix and vector inputs when the context is clear, with a slight abuse of notation. For example, given a feed-forward layer $\mathcal{F}^{(FF)} : \mathbb{R}^{D \times n} \to \mathbb{R}^{D \times n}$ defined as

$$\mathcal{F}^{(FF)}(\boldsymbol{H}) = \boldsymbol{H} + \boldsymbol{W}^{(2)} \sigma_R \left[ \boldsymbol{W}^{(1)} \boldsymbol{H} + \boldsymbol{b}^{(1)} \mathbf{1}_n^\top \right] + \boldsymbol{b}^{(2)} \mathbf{1}_n^\top,$$

we also define $\mathcal{F}^{(FF)} : \mathbb{R}^D \to \mathbb{R}^D$ as

$$\mathcal{F}^{(FF)}(\boldsymbol{H}_{:,i}) = \boldsymbol{H}_{:,i} + \boldsymbol{W}^{(2)} \sigma_R \left[ \boldsymbol{W}^{(1)} \boldsymbol{H}_{:,i} + \boldsymbol{b}^{(1)} \right] + \boldsymbol{b}^{(2)}$$

for each $i$, so that $\mathcal{F}^{(FF)}(\boldsymbol{H}) = (\mathcal{F}^{(FF)}(\boldsymbol{H}_{:,1}), \ldots, \mathcal{F}^{(FF)}(\boldsymbol{H}_{:,n}))$.

If all self-attention layers degenerate into identity mappings, the resulting Transformer reduces to a token-wise ResNet (He et al., 2016). We will demonstrate that any token-wise FNN can be represented by a token-wise ResNet, thereby naturally extending existing results on FNNs to Transformers. Specifically, an FNN $\mathcal{N} : \mathbb{R}^{d_x} \to \mathbb{R}^{d_y}$ is a function that can be parameterized in the form

$$\begin{aligned}
\mathcal{N}_0(\boldsymbol{x}) &= \boldsymbol{x}, \\
\mathcal{N}_{l+1}(\boldsymbol{x}) &= \sigma_R[\boldsymbol{A}_l \mathcal{N}_l(\boldsymbol{x}) + \boldsymbol{b}_l], \quad l = 0, \ldots, L-1, \\
\mathcal{N}(\boldsymbol{x}) &= \boldsymbol{A}_L \mathcal{N}_L(\boldsymbol{x}) + \boldsymbol{b}_L,
\end{aligned}$$

where $\boldsymbol{A}_l \in \mathbb{R}^{W_{l+1} \times W_l}, \boldsymbol{b}_l \in \mathbb{R}^{W_{l+1}}$ with $W_0 = d_x$, $W_{L+1} = d_y$ and $W_l = W$ for $l = 1, \ldots, L$. The parameters $W$ and $L$ are referred to as the width and depth of the neural network, respectively. We denote by $\mathcal{FNN}_{d_x, d_y}(W, L)$ the set of functions that can be parameterized in this form with width $W$ and depth $L$.

**Lemma A.1.** *Let $d_x, d_y$ be positive integers. For any $\mathcal{N} \in \mathcal{FNN}_{d_x, d_y}(W, L)$ with width $W \geq \max\{d_x, d_y\}$ and depth $L \geq 2$, there exist an embedding map $\mathcal{E}_{in} : \boldsymbol{X} \in \mathbb{R}^{d_x \times n} \mapsto \begin{pmatrix} \boldsymbol{X} \\ \boldsymbol{O} \end{pmatrix} \in \mathbb{R}^{W \times n}$, a projection map $\mathcal{E}_{out} : \begin{pmatrix} \boldsymbol{Y} \\ \boldsymbol{O} \end{pmatrix} \in \mathbb{R}^{W \times n} \mapsto \boldsymbol{Y} \in \mathbb{R}^{d_y \times n}$, and $L$ feed-forward layers with width at most $3W$, such that for any $\boldsymbol{X} \in \mathbb{R}^{d_x \times n}$,*

$$\mathcal{E}_{out} \circ \mathcal{F}_L^{(FF)} \circ \cdots \circ \mathcal{F}_1^{(FF)} \circ \mathcal{E}_{in}(\boldsymbol{X}) = (\mathcal{N}(\boldsymbol{X}_{:,1}), \ldots, \mathcal{N}(\boldsymbol{X}_{:,n})) \in \mathbb{R}^{d_y \times n}.$$

*Proof.* The idea is to use the identity $\sigma_R[x] - \sigma_R[-x] = x$ to eliminate the skip connection. For any $\boldsymbol{x} \in \mathbb{R}^{d_x}$, direct computation yields

$$\begin{aligned}
\mathcal{F}_1^{(FF)} \begin{pmatrix} \boldsymbol{x} \\ \boldsymbol{0} \end{pmatrix} &= \begin{pmatrix} \boldsymbol{x} \\ \boldsymbol{0} \end{pmatrix} + \begin{pmatrix} \boldsymbol{I}_{d_x} & \boldsymbol{O} & -\boldsymbol{I}_{d_x} & \boldsymbol{I}_{d_x} \\ \boldsymbol{O} & \boldsymbol{I}_{W-d_x} & \boldsymbol{O} & \boldsymbol{O} \end{pmatrix} \sigma_R \left[ \begin{pmatrix} \boldsymbol{A}_0 & \boldsymbol{O} \\ \boldsymbol{I}_{d_x} & \boldsymbol{O} \\ -\boldsymbol{I}_{d_x} & \boldsymbol{O} \end{pmatrix} \begin{pmatrix} \boldsymbol{x} \\ \boldsymbol{0} \end{pmatrix} + \begin{pmatrix} \boldsymbol{b}_0 \\ \boldsymbol{0} \\ \boldsymbol{0} \end{pmatrix} \right] \\
&= \sigma_R[\boldsymbol{A}_0 \boldsymbol{x} + \boldsymbol{b}_0] + \begin{pmatrix} \boldsymbol{x} + -\sigma_R[\boldsymbol{x}] + \sigma_R[-\boldsymbol{x}] \\ \boldsymbol{0} \end{pmatrix} \\
&= \mathcal{N}_1(\boldsymbol{x}),
\end{aligned}$$

where we have used the identity $\sigma_R[\boldsymbol{x}] - \sigma_R[-\boldsymbol{x}] = \boldsymbol{x}$.

Now, assuming that $\mathcal{F}_l^{(FF)} \circ \cdots \circ \mathcal{F}_1^{(FF)} \circ \mathcal{E}_{in}(\boldsymbol{x}) = \mathcal{N}_l(\boldsymbol{x})$, we define the $(l+1)$-th feed-forward layer $\mathcal{F}_{l+1}^{(FF)}$ as

$$\mathcal{F}_{l+1}^{(FF)}(\mathcal{N}_l(\boldsymbol{x})) = \mathcal{N}_l(\boldsymbol{x}) + \left(\boldsymbol{I}_W, -\boldsymbol{I}_W, \boldsymbol{I}_W\right) \sigma_R \left[ \begin{pmatrix} \boldsymbol{A}_l \\ \boldsymbol{I}_W \\ -\boldsymbol{I}_W \end{pmatrix} \mathcal{N}_l(\boldsymbol{x}) + \begin{pmatrix} \boldsymbol{b}_l \\ \boldsymbol{0} \\ \boldsymbol{0} \end{pmatrix} \right]$$

$$= \sigma_R \left[ \boldsymbol{A}_l \mathcal{N}_l(\boldsymbol{x}) + \boldsymbol{b}_l \right] = \mathcal{N}_{l+1}(\boldsymbol{x}).$$

By induction, it follows that

$$\mathcal{F}_{l+1}^{(FF)} \circ \mathcal{F}_l^{(FF)} \circ \cdots \circ \mathcal{F}_1^{(FF)} \circ \mathcal{E}_{in}(\boldsymbol{x}) = \mathcal{N}_{l+1}(\boldsymbol{x}).$$

By the principle of induction, we establish that $\mathcal{F}_{L-1}^{(FF)} \circ \cdots \circ \mathcal{F}_1^{(FF)} \circ \mathcal{E}_{in}(\boldsymbol{x}) = \mathcal{N}_{L-1}(\boldsymbol{x})$. For the last feed-forward layer, we calculate that

$$\mathcal{F}_L^{(FF)}(\mathcal{N}_{L-1}(\boldsymbol{x}))$$

$$= \mathcal{N}_{L-1}(\boldsymbol{x}) + \begin{pmatrix} \boldsymbol{A}_L & -\boldsymbol{I}_{d_y} & \boldsymbol{O} & \boldsymbol{I}_{d_y} & \boldsymbol{O} \\ \boldsymbol{O} & \boldsymbol{O} & -\boldsymbol{I}_{W-d_y} & \boldsymbol{O} & \boldsymbol{I}_{W-d_y} \end{pmatrix} \sigma_R \left[ \begin{pmatrix} \boldsymbol{A}_{L-1} \\ \boldsymbol{I}_W \\ -\boldsymbol{I}_W \end{pmatrix} \mathcal{N}_{L-1}(\boldsymbol{x}) + \begin{pmatrix} \boldsymbol{b}_{L-1} \\ \boldsymbol{0} \\ \boldsymbol{0} \end{pmatrix} \right] + \begin{pmatrix} \boldsymbol{b}_L \\ \boldsymbol{0} \end{pmatrix}$$

$$= \begin{pmatrix} \boldsymbol{A}_L \sigma_R[\boldsymbol{A}_{L-1}\mathcal{N}_{L-1}(\boldsymbol{x}) + \boldsymbol{b}_{L-1}] + \boldsymbol{b}_L \\ \boldsymbol{0} \end{pmatrix}$$

$$= \begin{pmatrix} \mathcal{N}(\boldsymbol{x}) \\ \boldsymbol{0} \end{pmatrix}.$$

Thus, we obtain

$$\mathcal{E}_{out} \circ \mathcal{F}_L^{(FF)} \circ \cdots \circ \mathcal{F}_1^{(FF)} \circ \mathcal{E}_{in}(\boldsymbol{x}) = \mathcal{N}(\boldsymbol{x}).$$

Since each feed-forward layer in our construction has width at most $3W$, the proof is complete by considering $\boldsymbol{x} = \boldsymbol{X}_{:,i}$ for $i \in [n]$. $\qquad\square$

The following proposition gives basic properties of Transformer networks that enable the recursive construction of complex architectures.

**Proposition A.2.** *Let* $\mathcal{N}_i \in \mathcal{T}_{d_i,k_i}(D_i, H_i, S_i, W_i, L_i)$ *for* $i = 1, 2$.

1. *If* $d_1 = d_2$, $k_1 = k_2$, *and* $D_1 \leq D_2, H_1 \leq H_2, S_1 \leq S_2, W_1 \leq W_2, L_1 \leq L_2$, *then*

$$\mathcal{T}_{d_1,k_1}(D_1, H_1, S_1, W_1, L_1) \subseteq \mathcal{T}_{d_2,k_2}(D_2, H_2, S_2, W_2, L_2).$$

2. *(Concatenation) If define* $\mathcal{N}\begin{pmatrix} \boldsymbol{X} \\ \boldsymbol{Y} \end{pmatrix} = \begin{pmatrix} \mathcal{N}_1(\boldsymbol{X}) \\ \mathcal{N}_2(\boldsymbol{Y}) \end{pmatrix}$, *then*

$$\mathcal{N} \in \mathcal{T}_{d_1+d_2,k_1+k_2}(D_1 + D_2, H_1 + H_2, \max\{S_1, S_2\}, W_1 + W_2, \max\{L_1, L_2\}).$$

3. *(Summation) If* $d_1 = d_2$ *and* $k_1 = k_2$, *then*

$$\mathcal{N}_1 + \mathcal{N}_2 \in \mathcal{T}_{d_1,k_1}(D_1 + D_2, H_1 + H_2, \max\{S_1, S_2\}, W_1 + W_2, \max\{L_1, L_2\}).$$

*Proof.* We provide the proof for (2), as the arguments for (1) and (3) follow analogously.

Let

$$\mathcal{N}_i = \mathcal{E}_{i,out} \circ \mathcal{F}_{i,L_i}^{(FF)} \circ \mathcal{F}_{i,L_i}^{(SA)} \circ \cdots \circ \mathcal{F}_{i,1}^{(FF)} \circ \mathcal{F}_{i,1}^{(SA)} \circ \mathcal{E}_{i,in} \in \mathcal{T}_{d_i,k_i}(D_i, H_i, S_i, W_i, L_i), \quad i = 1, 2.$$

Without loss of generality, assume that $L_1 = L_2 = L$, since the identity mapping can be viewed as a special self-attention layer or a feed-forward layer. We define the following components:

1. Input Embedding:

$$\mathcal{E}_{in}\begin{pmatrix} \boldsymbol{X} \\ \boldsymbol{Y} \end{pmatrix} = \begin{pmatrix} \boldsymbol{E}_{1,in} & \\ & \boldsymbol{E}_{2,in} \end{pmatrix}\begin{pmatrix} \boldsymbol{X} \\ \boldsymbol{Y} \end{pmatrix} + \begin{pmatrix} \boldsymbol{P}_1 \\ \boldsymbol{P}_2 \end{pmatrix} = \begin{pmatrix} \boldsymbol{E}_{1,in}\boldsymbol{X} + \boldsymbol{P}_1 \\ \boldsymbol{E}_{2,in}\boldsymbol{Y} + \boldsymbol{P}_2 \end{pmatrix} = \begin{pmatrix} \mathcal{E}_{1,in}(\boldsymbol{X}) \\ \mathcal{E}_{2,in}(\boldsymbol{Y}) \end{pmatrix}$$

2. Feed-forward Layer:

$$\mathcal{F}_l^{(FF)}\begin{pmatrix} \boldsymbol{X} \\ \boldsymbol{Y} \end{pmatrix}$$

$$= \begin{pmatrix} \boldsymbol{X} \\ \boldsymbol{Y} \end{pmatrix} + \begin{pmatrix} \boldsymbol{W}_{1,l}^{(2)} & \\ & \boldsymbol{W}_{2,l}^{(2)} \end{pmatrix} \sigma_R\left[\begin{pmatrix} \boldsymbol{W}_{1,l}^{(1)} & \\ & \boldsymbol{W}_{2,l}^{(1)} \end{pmatrix}\begin{pmatrix} \boldsymbol{X} \\ \boldsymbol{Y} \end{pmatrix} + \begin{pmatrix} \boldsymbol{b}_{1,l}^{(1)} \\ \boldsymbol{b}_{2,l}^{(1)} \end{pmatrix}\boldsymbol{1}_n^\top\right] + \begin{pmatrix} \boldsymbol{b}_{1,l}^{(2)} \\ \boldsymbol{b}_{2,l}^{(2)} \end{pmatrix}\boldsymbol{1}_n^\top$$

$$= \begin{pmatrix} \boldsymbol{X} + \boldsymbol{W}_{1,l}^{(2)}\sigma_R[\boldsymbol{W}_{1,l}^{(1)}\boldsymbol{X} + \boldsymbol{b}_{1,l}^{(1)}\boldsymbol{1}_n^\top] + \boldsymbol{b}_{1,l}^{(2)}\boldsymbol{1}_n^\top \\ \boldsymbol{Y} + \boldsymbol{W}_{2,l}^{(2)}\sigma_R[\boldsymbol{W}_{2,l}^{(1)}\boldsymbol{Y} + \boldsymbol{b}_{2,l}^{(1)}\boldsymbol{1}_n^\top] + \boldsymbol{b}_{2,l}^{(2)}\boldsymbol{1}_n^\top \end{pmatrix}$$

$$= \begin{pmatrix} \mathcal{F}_{1,l}^{(FF)}(\boldsymbol{X}) \\ \mathcal{F}_{2,l}^{(FF)}(\boldsymbol{Y}) \end{pmatrix}$$

3. Self-Attention Layer:

$$\mathcal{F}_l^{(SA)}\begin{pmatrix} \boldsymbol{X} \\ \boldsymbol{Y} \end{pmatrix}$$

$$= \begin{pmatrix} \boldsymbol{X} \\ \boldsymbol{Y} \end{pmatrix} + \sum_{h=1}^{H_1}\begin{pmatrix} \boldsymbol{W}_{1,h,l}^{(O)} \\ \boldsymbol{O} \end{pmatrix}\left(\boldsymbol{W}_{1,h,l}^{(V)},\boldsymbol{O}\right)\begin{pmatrix} \boldsymbol{X} \\ \boldsymbol{Y} \end{pmatrix}\sigma_S\left[\left(\left(\boldsymbol{W}_{1,h,l}^{(K)},\boldsymbol{O}\right)\begin{pmatrix} \boldsymbol{X} \\ \boldsymbol{Y} \end{pmatrix}\right)^\top\left(\boldsymbol{W}_{1,h,l}^{(Q)},\boldsymbol{O}\right)\begin{pmatrix} \boldsymbol{X} \\ \boldsymbol{Y} \end{pmatrix}\right]$$

$$+ \sum_{h=1}^{H_2}\begin{pmatrix} \boldsymbol{O} \\ \boldsymbol{W}_{2,h,l}^{(O)} \end{pmatrix}\left(\boldsymbol{O},\boldsymbol{W}_{2,h,l}^{(V)}\right)\begin{pmatrix} \boldsymbol{X} \\ \boldsymbol{Y} \end{pmatrix}\sigma_S\left[\left(\left(\boldsymbol{O},\boldsymbol{W}_{2,h,l}^{(K)}\right)\begin{pmatrix} \boldsymbol{X} \\ \boldsymbol{Y} \end{pmatrix}\right)^\top\left(\boldsymbol{O},\boldsymbol{W}_{2,h,l}^{(Q)}\right)\begin{pmatrix} \boldsymbol{X} \\ \boldsymbol{Y} \end{pmatrix}\right]$$

$$= \begin{pmatrix} \boldsymbol{X} + \sum_{h=1}^{H_1}\boldsymbol{W}_{1,h,l}^{(O)}\left(\boldsymbol{W}_{1,h,l}^{(V)}\boldsymbol{X}\right)\sigma_S\left[\left(\boldsymbol{W}_{1,h,l}^{(K)}\boldsymbol{X}\right)^\top\left(\boldsymbol{W}_{1,h,l}^{(Q)}\boldsymbol{X}\right)\right] \\ \boldsymbol{Y} + \sum_{h=1}^{H_2}\boldsymbol{W}_{2,h,l}^{(O)}\left(\boldsymbol{W}_{2,h,l}^{(V)}\boldsymbol{Y}\right)\sigma_S\left[\left(\boldsymbol{W}_{2,h,l}^{(K)}\boldsymbol{Y}\right)^\top\left(\boldsymbol{W}_{2,h,l}^{(Q)}\boldsymbol{Y}\right)\right] \end{pmatrix}$$

$$= \begin{pmatrix} \mathcal{F}_{1,l}^{(SA)}(\boldsymbol{X}) \\ \mathcal{F}_{2,l}^{(SA)}(\boldsymbol{Y}) \end{pmatrix}$$

4. Output Projection:

$$\mathcal{E}_{out}\begin{pmatrix} \boldsymbol{X} \\ \boldsymbol{Y} \end{pmatrix} = \begin{pmatrix} \boldsymbol{E}_{1,out} \\ \boldsymbol{E}_{2,out} \end{pmatrix}\begin{pmatrix} \boldsymbol{X} \\ \boldsymbol{Y} \end{pmatrix} = \begin{pmatrix} \boldsymbol{E}_{1,out}\boldsymbol{X} \\ \boldsymbol{E}_{2,out}\boldsymbol{Y} \end{pmatrix} = \begin{pmatrix} \mathcal{E}_{1,out}(\boldsymbol{X}) \\ \mathcal{E}_{2,out}(\boldsymbol{Y}) \end{pmatrix}$$

By direct verification, we obtain

$$\mathcal{N}\begin{pmatrix} \boldsymbol{X} \\ \boldsymbol{Y} \end{pmatrix} := \mathcal{E}_{out} \circ \mathcal{F}_L^{(FF)} \circ \mathcal{F}_L^{(SA)} \circ \cdots \circ \mathcal{F}_1^{(FF)} \circ \mathcal{F}_1^{(SA)} \circ \mathcal{E}_{in}\begin{pmatrix} \boldsymbol{X} \\ \boldsymbol{Y} \end{pmatrix} = \begin{pmatrix} \mathcal{N}_1(\boldsymbol{X}) \\ \mathcal{N}_2(\boldsymbol{Y}) \end{pmatrix}.$$

Furthermore, it follows that $\mathcal{N} \in \mathcal{T}_{d_1+d_2,k_1+k_2}(D_1 + D_2, H_1 + H_2, \max\{S_1, S_2\}, W_1 + W_2, L)$, thus completing the proof. $\qquad\square$

### A.1. Proof of Theorems 2.3 and 2.4

Given $K \in \mathbb{N}$ and $\delta \in (0, \frac{1}{K})$, we define a trifling region $\Omega([0, 1]^{D \times n}, K, \delta) \subseteq [0, 1]^{D \times n}$ as

$$\Omega([0, 1]^{D \times n}, K, \delta) := \left\{ \boldsymbol{X} \in [0, 1]^{D \times n} : \exists X_{i,j} \in \cup_{t=1}^{K-1}(\tfrac{t}{K}, \tfrac{t}{K} + \delta) \right\}.$$

The purpose of introducing the trifling region is to pinpoint the "bad" subsets where discrepancies arise when a discontinuous multi-step function is approximated by continuous piecewise linear functions, the latter being realizable through a feed-forward layer. Since the trifling region can be chosen to have arbitrarily small Lebesgue measure by taking $\delta$ sufficiently small, we focus on approximating the function over the "good" region, namely the complementary set $[0,1]^{D \times n} \setminus \Omega([0,1]^{D \times n}, K, \delta)$.

**Proposition A.3.** *Given $\gamma \in (0,1]$ and $K_{\mathcal{H}} > 0$, assume that $\boldsymbol{F} : [0,1]^{d_x \times n} \to \mathbb{R}^{d_x \times n}$ satisfies $F_{i,j} \in \mathcal{H}^\gamma([0,1]^{d_x \times n}, K_{\mathcal{H}})$ for each $i \in [d_x], j \in [n]$. For any $K \in \mathbb{N}$ and $\delta \in (0, \frac{1}{K})$, there exists a Transformer network $\mathcal{N} \in \mathcal{T}_{d_x,d_x}(d_x, 1, 1, 5nK^{d_x n}, 2)$ such that*

1. *$|\mathcal{N}_{i,j}(\boldsymbol{X}) - F_{i,j}(\boldsymbol{X})| \leq K_{\mathcal{H}}(d_x n)^{\gamma/2} K^{-\gamma}$ for any $i \in [d_x]$, $j \in [n]$ and $\boldsymbol{X} \in [0,1]^{d_x \times n} \setminus \Omega([0,1]^{d_x \times n}, K, \delta)$,*

2. *$\|\mathcal{N}(\boldsymbol{X})\|_F \leq \sqrt{d_x n} K_{\mathcal{H}}$ for any $\boldsymbol{X} \in \mathbb{R}^{d_x \times n}$.*

*Proof.* We basically follow the proof of Kajitsuka & Sato (2024, Proposition 1).

**Step 1:** We begin by uniformly partitioning the domain $[0,1]^{d_x \times n}$ into $K^{d_x n}$ subregions and constructing a piecewise constant function $\overline{\boldsymbol{F}}$ that approximates the target function $\boldsymbol{F}$, with an approximation error scales as $K^{-\gamma}$. Specifically, let $K \in \mathbb{N}$ denote the granularity of the grid

$$\mathbb{G}_K = \left\{ \tfrac{1}{K}, \tfrac{2}{K}, \ldots, 1 \right\}^{d_x \times n}.$$

We define each subregion as

$$\omega_{\boldsymbol{G}} := \prod_{i \in [d_x], j \in [n]} \begin{cases} [G_{i,j} - \frac{1}{K}, G_{i,j}], & \text{if } G_{i,j} = \frac{1}{K} \\ (G_{i,j} - \frac{1}{K}, G_{i,j}], & \text{otherwise} \end{cases}$$

associated with $\boldsymbol{G} \in \mathbb{G}_K$. Clearly, these subregions form a partition of the domain such that $[0,1]^{d_x \times n} = \bigcup_{\boldsymbol{G} \in \mathbb{G}_K} \omega_{\boldsymbol{G}}$. Given a target function $\boldsymbol{F}$ with $F_{i,j} \in \mathcal{H}^\gamma([0,1]^{d_x \times n}, K_{\mathcal{H}})$, we define a piecewise constant approximation of $\boldsymbol{F}$ as

$$\overline{\boldsymbol{F}}(\boldsymbol{X}) = \sum_{\boldsymbol{G} \in \mathbb{G}_K} \boldsymbol{F}(\boldsymbol{G}) \mathbb{1}_{\omega_{\boldsymbol{G}}}(\boldsymbol{X}),$$

where $\mathbb{1}_\omega$ denotes the indicator function of set $\omega$. That is, within each subregion $\omega_{\boldsymbol{G}}$, we approximate $\boldsymbol{F}$ using its value at the grid point $\boldsymbol{G}$. By the regularity of $\boldsymbol{F}$, we have the error estimate

$$
\begin{aligned}
|F_{i,j}(\boldsymbol{X}) - \overline{F}_{i,j}(\boldsymbol{X})| &= \left| \sum_{\boldsymbol{G} \in \mathbb{G}_K} (F_{i,j}(\boldsymbol{X}) - \overline{F}_{i,j}(\boldsymbol{X})) \mathbb{1}_{\omega_{\boldsymbol{G}}}(\boldsymbol{X}) \right| \\
&= \left| \sum_{\boldsymbol{G} \in \mathbb{G}_K} (F_{i,j}(\boldsymbol{X}) - F_{i,j}(\boldsymbol{G})) \mathbb{1}_{\omega_{\boldsymbol{G}}}(\boldsymbol{X}) \right| \\
&\leq \sum_{\boldsymbol{G} \in \mathbb{G}_K} |(F_{i,j}(\boldsymbol{X}) - F_{i,j}(\boldsymbol{G}))| \mathbb{1}_{\omega_{\boldsymbol{G}}}(\boldsymbol{X}) \\
&\leq \sum_{\boldsymbol{G} \in \mathbb{G}_K} K_{\mathcal{H}} \|\boldsymbol{X} - \boldsymbol{G}\|_F^\gamma \mathbb{1}_{\omega_{\boldsymbol{G}}}(\boldsymbol{X}) \\
&\leq K_{\mathcal{H}}(d_x n)^{\gamma/2} K^{-\gamma} \sum_{\boldsymbol{G} \in \mathbb{G}_K} \mathbb{1}_{\omega_{\boldsymbol{G}}}(\boldsymbol{X}) \\
&= K_{\mathcal{H}}(d_x n)^{\gamma/2} K^{-\gamma},
\end{aligned}
\tag{4}
$$

for any $i \in [d_x]$, $j \in [n]$ and $\boldsymbol{X} \in [0,1]^{d_x \times n}$.

**Step 2:** Given a positional encoding matrix $\boldsymbol{P}$ and a spatial discretization function $\overline{\mathcal{F}}_1^{(FF)}$ satisfying

$$\overline{\mathcal{F}}_1^{(FF)}(\boldsymbol{X} + \boldsymbol{P}) = \boldsymbol{G} + \boldsymbol{P}, \quad \text{for all } \boldsymbol{X} \in \omega_{\boldsymbol{G}},$$

our objective is to construct a feed-forward layer $\mathcal{F}_1^{(FF)}$, with width at most $2nd_x(K+1)$, that accurately represents $\overline{\mathcal{F}}_1^{(FF)}$ outside the trifling region $\Omega([0,1]^{d_x \times n}, K, \delta)$, that is,

$$\mathcal{F}_1^{(FF)}(\boldsymbol{X} + \boldsymbol{P}) = \overline{\mathcal{F}}_1^{(FF)}(\boldsymbol{X} + \boldsymbol{P}), \quad \text{for any } \boldsymbol{X} \in [0,1]^{d_x \times n} \setminus \Omega([0,1]^{d_x \times n}, K, \delta).$$

To achieve this goal, we first approximate a univariate multiple-step function using a piecewise linear function, and then extend this function to matrix elements by stacking.

We define the embedding layer as

$$\mathcal{E}_{in}(\boldsymbol{X}) = \boldsymbol{X} + \boldsymbol{P} \in \mathbb{R}^{d_x \times n},$$

where the positional encoding matrix $\boldsymbol{P}$ is chosen as

$$\boldsymbol{P} = \begin{pmatrix} 0 & 2 & \cdots & 2(n-1) \\ \vdots & \vdots & & \vdots \\ 0 & 2 & \cdots & 2(n-1) \end{pmatrix}.$$

Since $\boldsymbol{X} \in [0,1]^{d_x \times n}$, the positional encoding ensures that the columns of $\boldsymbol{X} + \boldsymbol{P}$ are mapped to distinct intervals, that is, $[\boldsymbol{X} + \boldsymbol{P}]_{i,j} \in [2j-2, 2j-1]$ for each $j \in [n]$. Now, consider a multiple-step function $\text{step}_K(z)$ defined on $[0,1]$ as

$$\text{step}_K(z) = \begin{cases} \frac{1}{K}, & 0 \leq z \leq \frac{1}{K} \\ \frac{2}{K}, & \frac{1}{K} < z \leq \frac{2}{K} \\ \frac{3}{K}, & \frac{2}{K} < z \leq \frac{3}{K} \\ \vdots & \vdots \\ 1, & 1 - \frac{1}{K} < z \leq 1 \end{cases}.$$

Given $\delta \in (0, \frac{1}{K})$, by translations, scalings and summations of the $\delta$-approximated step function

$$\sigma_R[z/\delta] - \sigma_R[z/\delta - 1] = \begin{cases} 0 & z \leq 0 \\ z/\delta & 0 < z < \delta \\ 1 & \delta \leq z \end{cases},$$

we define

$$f(z) = \frac{1}{K} + \sum_{j=1}^{n} \sum_{t=1}^{K-1} \frac{1}{K} \left( \sigma_R \left[ \frac{z - 2(j-1)}{\delta} - \frac{t}{\delta K} \right] - \sigma_R \left[ \frac{z - 2(j-1)}{\delta} - 1 - \frac{t}{\delta K} \right] \right)$$

$$+ \sum_{j=1}^{n-1} \left( 1 + \frac{1}{K} \right) \left( \sigma_R[z - (2j-1)] - \sigma_R[z - 2j] \right).$$

We see that $f(z + (2j-2)) = \text{step}_K(z) + (2j-2)$ for all $z \in [0,1] \setminus \Omega([0,1], K, \delta)$ and $j \in [n]$. Moreover, the function $f$ can be represented by a shallow ReLU network with $2nK - 2$ units in the hidden layer.

We then concatenate multiple $f$ in parallel to construct a feed-forward layer $\mathcal{F}_1^{(FF)} : \mathbb{R}^{d_x \times n} \to \mathbb{R}^{d_x \times n}$ with width at most $2nd_x(K+1)$, satisfying

$$\mathcal{F}_1^{(FF)}(\boldsymbol{X} + \boldsymbol{P}) = \begin{pmatrix} f(X_{1,1} + P_{1,1}) & \cdots & f(X_{1,n} + P_{1,n}) \\ \vdots & \ddots & \vdots \\ f(X_{d_x,1} + P_{d_x,1}) & \cdots & f(X_{d_x,n} + P_{d_x,n}) \end{pmatrix}$$

$$\approx \begin{pmatrix} \text{step}_K(X_{1,1}) + P_{1,1} & \cdots & \text{step}_K(X_{1,n}) + P_{1,n} \\ \vdots & \ddots & \vdots \\ \text{step}_K(X_{d_x,1}) + P_{d_x,1} & \cdots & \text{step}_K(X_{d_x,n}) + P_{d_x,n} \end{pmatrix}$$

$$= \begin{pmatrix} \text{step}_K(X_{1,1}) & \cdots & \text{step}_K(X_{1,n}) \\ \vdots & \ddots & \vdots \\ \text{step}_K(X_{d_x,1}) & \cdots & \text{step}_K(X_{d_x,n}) \end{pmatrix} + \boldsymbol{P}.$$

Noting that

$$\Omega([0,1]^{d_x \times n}, K, \delta) = \bigcup_{i \in [d_x], j \in [n]} \{\boldsymbol{X} : X_{i,j} \in \Omega([0,1], K, \delta)\},$$

we conclude $\mathcal{F}_1^{(FF)}(\boldsymbol{X} + \boldsymbol{P}) = \overline{\mathcal{F}}_1^{(FF)}(\boldsymbol{X} + \boldsymbol{P})$ for any $\boldsymbol{X} \in [0,1]^{d_x \times n} \setminus \Omega([0,1]^{d_x \times n}, K, \delta)$.

**Step 3:** Since $\{\boldsymbol{G} + \boldsymbol{P} : \boldsymbol{G} \in \mathbb{G}_K\}$ can be regarded as sequences, each of which has no duplicate token due to positional encoding, it follows from Kajitsuka & Sato (2024, Theorem 2) that there exists a self-attention layer $\mathcal{F}^{(SA)} : \mathbb{R}^{d_x \times n} \to \mathbb{R}^{d_x \times n}$ with $H = 1$ and $s = 1$ that serves as a contextual mapping for such input sequences (see Kajitsuka & Sato (2024) and Yun et al. (2020) for further discussion). In essence, a contextual mapping is a bijection between sequences that satisfies $\mathcal{F}^{(SA)}(\boldsymbol{G}^{(i)} + \boldsymbol{P})_{:,k} \neq \mathcal{F}^{(SA)}(\boldsymbol{G}^{(j)} + \boldsymbol{P})_{:,l}$ if $\boldsymbol{G}^{(i)} \neq \boldsymbol{G}^{(j)} \in \mathbb{G}_K$ or $k \neq l \in [n]$. The remaining is to associate each output token with its corresponding function value using a feed-forward layer, which reduces to a memorization task. Lemma A.7 gives a construction of such a feed-forward layer, denoted as $\mathcal{F}_2^{(FF)}$, with width at most $5nK^{d_x n}$ (set $r = n \cdot |\mathbb{G}_K| \leq nK^{d_x n}$ therein), such that

$$\mathcal{F}_2^{(FF)}(\mathcal{F}^{(SA)}(\boldsymbol{G} + \boldsymbol{P})) = \boldsymbol{F}(\boldsymbol{G}) \quad \text{for all } \boldsymbol{G} \in \mathbb{G}_K,$$

and $\|\mathcal{F}_2^{(FF)}(\boldsymbol{Z})\|_F \leq \sqrt{d_x n} K_{\mathcal{H}}$.

Let $\mathcal{E}_{out}$ be the identity mapping. It holds that $\mathcal{E}_{out} \circ \mathcal{F}_2^{(FF)} \circ \mathcal{F}^{(SA)} \circ \mathcal{F}_1^{(FF)} \circ \mathcal{E}_{in} \in \mathcal{T}_{d_x, d_x}(d_x, 1, 1, 5nK^{d_x n}, 2)$. Note that for any $\boldsymbol{X} \in \omega_{\boldsymbol{G}} \setminus \Omega([0,1]^{d_x \times n}, K, \delta)$,

$$\begin{aligned}
&\mathcal{E}_{out} \circ \mathcal{F}_2^{(FF)} \circ \mathcal{F}^{(SA)} \circ \mathcal{F}_1^{(FF)} \circ \mathcal{E}_{in}(\boldsymbol{X}) \\
&= \mathcal{F}_2^{(FF)} \circ \mathcal{F}^{(SA)} \circ \mathcal{F}_1^{(FF)}(\boldsymbol{X} + \boldsymbol{P}) \\
&= \mathcal{F}_2^{(FF)} \circ \mathcal{F}^{(SA)} \circ \overline{\mathcal{F}}_1^{(FF)}(\boldsymbol{X} + \boldsymbol{P}) \\
&= \mathcal{F}_2^{(FF)} \circ \mathcal{F}^{(SA)}(\boldsymbol{G} + \boldsymbol{P}) \\
&= \boldsymbol{F}(\boldsymbol{G}) = \overline{\boldsymbol{F}}(\boldsymbol{X}).
\end{aligned}$$

Thus, for any $\boldsymbol{X} \in [0,1]^{d_x \times n} \setminus \Omega([0,1]^{d_x \times n}, K, \delta) = \bigcup_{\boldsymbol{G} \in \mathbb{G}_K} \omega_{\boldsymbol{G}} \setminus \Omega([0,1]^{d_x \times n}, K, \delta)$, we have

$$\mathcal{E}_{out} \circ \mathcal{F}_2^{(FF)} \circ \mathcal{F}^{(SA)} \circ \mathcal{F}_1^{(FF)} \circ \mathcal{E}_{in}(\boldsymbol{X}) = \overline{\boldsymbol{F}}(\boldsymbol{X}),$$

which completes the proof by noting (4).

$\square$

**Proposition A.4.** *Given* $\gamma \in (0,1]$ *and* $K_{\mathcal{H}} > 0$, *assume that* $\boldsymbol{F} : [0,1]^{d_x \times n} \to \mathbb{R}^{d_x \times n}$ *with each entry* $F_{i,j} \in \mathcal{H}^\gamma([0,1]^{d_x \times n}, K_{\mathcal{H}})$. *For any* $\varepsilon > 0$, $K \in \mathbb{N}$ *and* $\delta \in (0, \frac{1}{3K}]$, *if* $\widetilde{\mathcal{N}} \in \mathcal{T}_{d_x, d_x}(D, H, S, W, L)$ *is a Transformer network that satisfies*

$$|\widetilde{\mathcal{N}}_{i,j}(\boldsymbol{X}) - F_{i,j}(\boldsymbol{X})| \leq \varepsilon$$

*for any* $i \in [d_x]$, $j \in [n]$ *and* $\boldsymbol{X} \in [0,1]^{d_x \times n} \setminus \Omega([0,1]^{d_x \times n}, K, \delta)$, *then there exists a new Transformer network*

$$\mathcal{N} \in \mathcal{T}_{d_x, d_x}(3^{d_x n} \max\{D, 5d_x\}, 3^{d_x n} H, S, 3^{d_x n} \max\{W, 14d_x\}, L + 2d_x n),$$

*such that*

$$|\mathcal{N}_{i,j}(\boldsymbol{X}) - F_{i,j}(\boldsymbol{X})| \leq \varepsilon + d_x n K_{\mathcal{H}} \delta^\gamma$$

*for any* $i \in [d_x]$, $j \in [n]$ *and* $\boldsymbol{X} \in [0,1]^{d_x \times n}$.

*Proof.* We basically follow the proof of Lu et al. (2021, Theorem 2.1).

**Step 1:** We prove that, given $i \in [d_x]$, $j \in [n]$, $F_{i,j} \in \mathcal{H}^\gamma([0,1]^{d_x \times n}, K_{\mathcal{H}})$, and a general function $G_{i,j} : \mathbb{R}^{d_x \times n} \to \mathbb{R}$ satisfying

$$|G_{i,j}(\boldsymbol{X}) - F_{i,j}(\boldsymbol{X})| \leq \varepsilon \text{ for any } \boldsymbol{X} \in [0,1]^{d_x \times n} \setminus \Omega([0,1]^{d_x \times n}, K, \delta), \tag{5}$$

then

$$|\Phi_{i,j}(\boldsymbol{X}) - F_{i,j}(\boldsymbol{X})| \leq \varepsilon + d_x n K_{\mathcal{H}} \delta^\gamma \text{ for any } \boldsymbol{X} \in [0,1]^{d_x \times n}, \tag{6}$$

where $\Phi_{i,j} := \Phi_{i,j}^{(d_x n)}$ is defined by induction through

$$\Phi_{i,j}^{(k)}(\boldsymbol{X}) := \text{mid}\left(\Phi_{i,j}^{(k-1)}\left(\boldsymbol{X} - \delta \boldsymbol{E}^{(k)}\right), \Phi_{i,j}^{(k-1)}(\boldsymbol{X}), \Phi_{i,j}^{(k-1)}\left(\boldsymbol{X} + \delta \boldsymbol{E}^{(k)}\right)\right) \tag{7}$$

for $k = 1, 2, \cdots, d_x n$, $\Phi_{i,j}^{(0)} = G_{i,j}$, $\text{mid}(\cdot, \cdot, \cdot)$ is a function returning the middle value of three inputs, and $\boldsymbol{E}^{(u+(v-1)d_x)}$ denotes the matrix with 1 at the $(u,v)$-th position and 0 elsewhere for $u \in [d_x]$, $v \in [n]$. In other words, if $G_{i,j}$ provides a uniform approximation outside the trifling region, then the carefully constructed $\Phi_{i,j}$ extends this uniform approximation to the entire domain, with only a slight increase in the approximation error.

Note that $\{\boldsymbol{E}^{(k)}\}_{k=1}^{d_x n}$ defined above is a re-indexing of the standard basis in $\mathbb{R}^{d_x \times n}$. We re-index the elements of $\boldsymbol{X} = (X_{u,v})$ in the same manner. Let $X^{(u+(v-1)d_x)} = X_{u,v}$ for $u \in [d_x]$, $v \in [n]$. Using this notation, define

$$\Omega_k := \left\{ \boldsymbol{X} : X^{(i)} \in \begin{cases} [0,1], & \text{if } i \leq k \\ [0,1] \setminus \Omega([0,1], K, \delta), & \text{if } i > k \end{cases} \right\}.$$

Clearly, $\Omega_0 = [0,1]^{d_x \times n} \setminus \Omega([0,1]^{d_x \times n}, K, \delta)$ and $\Omega_{d_x n} = [0,1]^{d_x \times n}$.

We will prove by induction that for each $k \in \{0, 1, \ldots, d_x n\}$,

$$|\Phi_{i,j}^{(k)}(\boldsymbol{X}) - F_{i,j}(\boldsymbol{X})| \leq \varepsilon + k \cdot K_{\mathcal{H}} \delta^\gamma \text{ for any } \boldsymbol{X} \in \Omega_k. \tag{8}$$

As the final step of the induction, we derive

$$\begin{aligned}
|\Phi_{i,j}(\boldsymbol{X}) - F_{i,j}(\boldsymbol{X})| &= |\Phi_{i,j}^{(d_x n)}(\boldsymbol{X}) - F_{i,j}(\boldsymbol{X})| \\
&\leq \varepsilon + d_x n K_{\mathcal{H}} \delta^\gamma \text{ for any } \boldsymbol{X} \in \Omega_{d_x n} = [0,1]^{d_x \times n},
\end{aligned}$$

which completes the proof of (6).

In the base case, it follows from (5) that

$$\begin{aligned}
|\Phi_{i,j}^{(0)}(\boldsymbol{X}) - F_{i,j}(\boldsymbol{X})| &= |G_{i,j}(\boldsymbol{X}) - F_{i,j}(\boldsymbol{X})| \\
&\leq \varepsilon \text{ for any } \boldsymbol{X} \in \Omega_0 = [0,1]^{d_x \times n} \setminus \Omega([0,1]^{d_x \times n}, K, \delta).
\end{aligned}$$

Now, assume that for some $k \in \{1, 2, \ldots, d_x n\}$,

$$|\Phi_{i,j}^{(k-1)}(\boldsymbol{X}) - F_{i,j}(\boldsymbol{X})| \leq \varepsilon + (k-1) K_{\mathcal{H}} \delta^\gamma \text{ for any } \boldsymbol{X} \in \Omega_{k-1}.$$

For fixed $X^{(1)}, \ldots, X^{(k-1)} \in [0,1]$ and $X^{(k+1)}, \ldots, X^{(d_x n)} \in [0,1] \setminus \Omega([0,1], K, \delta)$, define

$$\phi(t) = \Phi_{i,j}^{(k-1)}(X^{(1)}, \ldots, X^{(k-1)}, t, X^{(k+1)}, \ldots, X^{(d_x n)})$$

and

$$f(t) = F_{i,j}(X^{(1)}, \ldots, X^{(k-1)}, t, X^{(k+1)}, \ldots, X^{(d_x n)}).$$

The induction hypothesis gives

$$|\phi(t) - f(t)| \leq \varepsilon + (k-1) \cdot K_{\mathcal{H}} \delta^\gamma \text{ for any } t \in [0,1] \setminus \Omega([0,1], K, \delta).$$

Since $F_{i,j} \in \mathcal{H}^\gamma([0,1]^{d_x \times n}, K_{\mathcal{H}})$ implies $f \in \mathcal{H}^\gamma([0,1], K_{\mathcal{H}})$, applying Lemma A.6 to the univariate functions $\phi(t)$ and $f(t)$ yields

$$|\widetilde{\phi}(t) - f(t)| \leq \varepsilon + (k-1) \cdot K_{\mathcal{H}}\delta^\gamma + K_{\mathcal{H}}\delta^\gamma = \varepsilon + k \cdot K_{\mathcal{H}}\delta^\gamma \quad \text{for any } t \in [0,1],$$

where

$$\begin{aligned}
\widetilde{\phi}(t) &= \text{mid}\,(\phi(t-\delta), \phi(t), \phi(t+\delta)) \\
&= \text{mid}(\Phi_{i,j}^{(k-1)}(X^{(1)}, \ldots, X^{(k-1)}, t-\delta, X^{(k+1)}, \ldots, X^{(d_x n)}), \\
&\qquad\quad \Phi_{i,j}^{(k-1)}(X^{(1)}, \ldots, X^{(k-1)}, t, X^{(k+1)}, \ldots, X^{(d_x n)}), \\
&\qquad\quad \Phi_{i,j}^{(k-1)}(X^{(1)}, \ldots, X^{(k-1)}, t+\delta, X^{(k+1)}, \ldots, X^{(d_x n)})) \\
&= \Phi_{i,j}^{(k)}(X^{(1)}, \ldots, X^{(k-1)}, t, X^{(k+1)}, \ldots, X^{(d_x n)})
\end{aligned}$$

by definition of $\Phi_{i,j}^{(k)}$. Since $X^{(1)}, \ldots, X^{(k-1)} \in [0,1]$, $X^{(k)} = t \in [0,1]$ and $X^{(k+1)}, \ldots, X^{(d_x n)} \in [0,1] \setminus \Omega([0,1], K, \delta)$ are arbitrary, we obtain

$$|\Phi_{i,j}^{(k)}(\boldsymbol{X}) - F_{i,j}(\boldsymbol{X})| \leq \varepsilon + k \cdot K_{\mathcal{H}}\delta^\gamma \quad \text{for any } \boldsymbol{X} \in \Omega_k.$$

This completes the induction.

**Step 2:** Recall that $\boldsymbol{\Phi} = (\Phi_{i,j})_{i\in[d_x], j\in[n]}$ is defined by (7). We now prove that if $\boldsymbol{G} \in \mathcal{T}_{d_x, d_x}(D, H, S, W, L)$, then

$$\boldsymbol{\Phi} \in \mathcal{T}_{d_x, d_x}(3^{d_x n}\max\{D, 5d_x\}, 3^{d_x n}H, S, 3^{d_x n}\max\{W, 14d_x\}, L + 2d_x n).$$

The observation here is that, to compute $\boldsymbol{\Phi} = \boldsymbol{\Phi}^{(d_x n)}$, we first evaluate

$$\boldsymbol{\Phi}^{(d_x n-1)}(\cdot + c_{d_x n}\delta\boldsymbol{E}^{(d_x n)}) \quad \text{for each } c_{d_x n} \in \{-1, 0, 1\}.$$

Each such evaluation, in turn, requires computing

$$\boldsymbol{\Phi}^{(d_x n-2)}(\cdot + c_{d_x n-1}\delta\boldsymbol{E}^{(d_x n-1)} + c_{d_x n}\delta\boldsymbol{E}^{(d_x n)}) \quad \text{for each } c_{d_x n-1} \in \{-1, 0, 1\}.$$

Continuing this process recursively, determining $\boldsymbol{\Phi}$ ultimately requires evaluating

$$\boldsymbol{\Phi}^{(0)}(\cdot + \textstyle\sum_{l=1}^{d_x n} c_l \delta\boldsymbol{E}^{(l)}) \quad \text{for every } (c_1, \ldots, c_{d_x n}) \in \{-1, 0, 1\}^{d_x n}.$$

Reversing the overall construction and noting that $\boldsymbol{\Phi}^{(0)} = \boldsymbol{G}$ by definition, if we are given access to all functions of the form $\boldsymbol{G}(\cdot + \sum_{l=1}^{d_x n} c_l \delta\boldsymbol{E}^{(l)})$, then we can repeatedly apply the $\text{mid}$ operation to $\boldsymbol{\Phi}^{(k)}$ to obtain $\boldsymbol{\Phi}^{(k+1)}$, using the same procedure as in (7). By iterating this process up to $k = d_x n$, we finally recover $\boldsymbol{\Phi} = \boldsymbol{\Phi}^{(d_x n)}$. On the other hand, each function $\boldsymbol{G}(\cdot + \sum_{k=1}^{d_x n} c_k \delta\boldsymbol{E}^{(k)})$ is a Transformer network thanks to positional encoding, and the $\text{mid}$ function can be implemented by feed-forward layers and vectorized operations, thereby completing the construction. The details are given below.

Fixing $k \in \{0, 1, \ldots, d_x n - 1\}$, we reindex the functions

$$\left\{ \boldsymbol{\Phi}^{(k)}(\cdot + \textstyle\sum_{l=k+1}^{d_x n} c_l \delta\boldsymbol{E}^{(l)}) : (c_{k+1}, \ldots, c_{d_x n}) \in \{-1, 0, 1\}^{d_x n - k} \right\}$$

as $\{\boldsymbol{\Phi}_l^{(k)}\}_{l=1}^{3^{d_x n - k}}$ (set $\boldsymbol{\Phi}_1^{(d_x n)} = \boldsymbol{\Phi}^{(d_x n)}$ for notational convenience), such that

$$\left[\boldsymbol{\Phi}_l^{(k+1)}\right]_{i,j} = \text{mid}\left(\left[\boldsymbol{\Phi}_{3l-2}^{(k)}\right]_{i,j}, \left[\boldsymbol{\Phi}_{3l-1}^{(k)}\right]_{i,j}, \left[\boldsymbol{\Phi}_{3l}^{(k)}\right]_{i,j}\right)$$

for all $i \in [d_x], j \in [n]$, which aligns with (7). Since $\text{mid}(\cdot, \cdot, \cdot) \in \mathcal{FNN}_{3,1}(14, 2)$ by Lemma A.5, there exists an FNN $\widetilde{\mathcal{N}} \in \mathcal{FNN}_{3d_x, d_x}(14d_x, 2)$, such that for all $j \in [n]$,

$$\widetilde{\mathcal{N}}\left(\begin{array}{c} \left[\boldsymbol{\Phi}_{3l-2}^{(k)}\right]_{:,j} \\ \left[\boldsymbol{\Phi}_{3l-1}^{(k)}\right]_{:,j} \\ \left[\boldsymbol{\Phi}_{3l}^{(k)}\right]_{:,j} \end{array}\right) = \left[\boldsymbol{\Phi}_l^{(k+1)}\right]_{:,j}.$$

We then concatenate $\widetilde{\mathcal{N}}$ in parallel to construct a new FNN

$$\widetilde{\mathcal{N}}^{(k)} \in \mathcal{FNN}_{3d_x \cdot 3^{d_x n - k - 1}, d_x \cdot 3^{d_x n - k - 1}}(14d_x \cdot 3^{d_x n - k - 1}, 2)$$

such that

$$\widetilde{\mathcal{N}}^{(k)} \begin{pmatrix} \left[\mathbf{\Phi}_1^{(k)}\right]_{:,j} \\ \left[\mathbf{\Phi}_2^{(k)}\right]_{:,j} \\ \left[\mathbf{\Phi}_3^{(k)}\right]_{:,j} \\ \vdots \\ \left[\mathbf{\Phi}_{3^{d_x n - k} - 2}^{(k)}\right]_{:,j} \\ \left[\mathbf{\Phi}_{3^{d_x n - k} - 1}^{(k)}\right]_{:,j} \\ \left[\mathbf{\Phi}_{3^{d_x n - k}}^{(k)}\right]_{:,j} \end{pmatrix} = \begin{pmatrix} \left[\mathbf{\Phi}_1^{(k+1)}\right]_{:,j} \\ \vdots \\ \left[\mathbf{\Phi}_{3^{d_x n - k - 1}}^{(k+1)}\right]_{:,j} \end{pmatrix}.$$

By recursively composing $\widetilde{\mathcal{N}}^{(k)}$ for all $k \in \{0, 1, \ldots, d_x n - 1\}$, we obtain

$$\widetilde{\mathcal{N}}^{(d_x n - 1)} \circ \widetilde{\mathcal{N}}^{(d_x n - 2)} \circ \cdots \circ \widetilde{\mathcal{N}}^{(0)} \in \mathcal{FNN}_{d_x 3^{d_x n}, d_x}(14d_x 3^{d_x n - 1}, 2d_x n),$$

which, by construction, satisfies

$$\widetilde{\mathcal{N}}^{(d_x n - 1)} \circ \widetilde{\mathcal{N}}^{(d_x n - 2)} \circ \cdots \circ \widetilde{\mathcal{N}}^{(0)} \begin{pmatrix} \left[\mathbf{\Phi}_1^{(0)}\right]_{:,j} \\ \left[\mathbf{\Phi}_2^{(0)}\right]_{:,j} \\ \left[\mathbf{\Phi}_3^{(0)}\right]_{:,j} \\ \vdots \\ \left[\mathbf{\Phi}_{3^{d_x n} - 2}^{(0)}\right]_{:,j} \\ \left[\mathbf{\Phi}_{3^{d_x n} - 1}^{(0)}\right]_{:,j} \\ \left[\mathbf{\Phi}_{3^{d_x n}}^{(0)}\right]_{:,j} \end{pmatrix}$$

$$= \widetilde{\mathcal{N}}^{(d_x n - 1)} \circ \widetilde{\mathcal{N}}^{(d_x n - 2)} \circ \cdots \circ \widetilde{\mathcal{N}}^{(1)} \begin{pmatrix} \left[\mathbf{\Phi}_1^{(1)}\right]_{:,j} \\ \vdots \\ \left[\mathbf{\Phi}_{3^{d_x n - 1}}^{(1)}\right]_{:,j} \end{pmatrix}$$

$$\vdots$$

$$= \left[\mathbf{\Phi}^{(d_x n)}\right]_{:,j},$$

for each $j \in [n]$. Furthermore, Lemma A.1 guarantees that any token-wise FNN can be expressed in terms of feed-forward layers. We have (set $W = 14d_x 3^{d_x n - 1}$ and $L = 2d_x n$ therein) an embedding map $\mathcal{E}_{in} : \mathbb{R}^{d_x 3^{d_x n} \times n} \to \mathbb{R}^{14d_x 3^{d_x n - 1} \times n}$, a projection map $\mathcal{E}_{out} : \mathbb{R}^{14d_x 3^{d_x n - 1} \times n} \to \mathbb{R}^{d_x \times n}$, and $2d_x n$ feed-forward layers $\mathcal{F}_{L+2d_x n}^{(FF)}, \ldots, \mathcal{F}_{L+1}^{(FF)}$, each with width at most $3 \cdot 14d_x 3^{d_x n - 1} = 14d_x 3^{d_x n}$, such that

$$\mathcal{E}_{out} \circ \mathcal{F}_{L+2d_x n}^{(FF)} \circ \cdots \circ \mathcal{F}_{L+1}^{(FF)} \circ \mathcal{E}_{in} \begin{pmatrix} \mathbf{\Phi}_1^{(0)} \\ \mathbf{\Phi}_2^{(0)} \\ \vdots \\ \mathbf{\Phi}_{3^{d_x n}}^{(0)} \end{pmatrix} = \mathbf{\Phi}^{(d_x n)}. \tag{9}$$

Due to the positional encoding, each function

$$G(\cdot + \textstyle\sum_{l=1}^{d_x n} c_k \delta E^{(l)}) \in \mathcal{T}_{d_x, d_x}(D, H, S, W, L).$$

Recall that $G = \Phi^{(0)}$ and $\{\Phi_l^{(0)}\}_{l=1}^{3^{d_x n}}$ is a reordering of $\{\Phi^{(0)}(\cdot + \sum_{l=1}^{d_x n} c_l \delta E^{(l)})\}$. By concatenation of Transformers (see Proposition A.2), there exists a Transformer network

$$\mathcal{N} \in \mathcal{T}_{d_x, d_x 3^{d_x n}}(3^{d_x n}D, 3^{d_x n}H, S, 3^{d_x n}W, L)$$

such that

$$\mathcal{N}(X) = \begin{pmatrix} \Phi_1^{(0)} \\ \Phi_2^{(0)} \\ \vdots \\ \Phi_{3^{d_x n}}^{(0)} \end{pmatrix}.$$

Together with (9) and $\Phi^{(d_x n)} = \Phi$, we have

$$\mathcal{E}_{out} \circ \mathcal{F}_{L+2d_x n}^{(FF)} \circ \cdots \circ \mathcal{F}_{L+1}^{(FF)} \circ \mathcal{E}_{in} \circ \mathcal{N}(X)$$

$$= \mathcal{E}_{out} \circ \mathcal{F}_{L+2d_x n}^{(FF)} \circ \cdots \circ \mathcal{F}_{L+1}^{(FF)} \circ \mathcal{E}_{in} \begin{pmatrix} \Phi_1^{(0)} \\ \Phi_2^{(0)} \\ \vdots \\ \Phi_{3^{d_x n}}^{(0)} \end{pmatrix}$$

$$= \Phi^{(d_x n)} = \Phi(X),$$

thereby

$$\Phi \in \mathcal{T}_{d_x, d_x}(\max\{3^{d_x n}D, 14 d_x 3^{d_x n - 1}\}, 3^{d_x n}H, S, \max\{3^{d_x n}W, 14 d_x 3^{d_x n}\}, L + 2d_x n)$$
$$\subseteq \mathcal{T}_{d_x, d_x}(3^{d_x n}\max\{D, 5d_x\}, 3^{d_x n}H, S, 3^{d_x n}\max\{W, 14d_x\}, L + 2d_x n),$$

which completes the proof. $\qquad\square$

**Lemma A.5** (Lemma 3.1 of Lu et al. (2021)). *The middle value function* $\mathrm{mid}(x_1, x_2, x_3) \in \mathcal{FNN}_{3,1}(14, 2)$.

**Lemma A.6** (Lemma 3.3 of Lu et al. (2021)). *Given any* $\varepsilon > 0$, $K \in \mathbb{N}$, *and* $\delta \in (0, \frac{1}{3K}]$, *assume that* $f \in \mathcal{H}^\gamma([0,1], K_\mathcal{H})$ *and* $g : \mathbb{R} \to \mathbb{R}$ *is a general function with*

$$|g(x) - f(x)| \leq \varepsilon, \text{ for any } x \in [0,1] \setminus \Omega([0,1], K, \delta).$$

*Then*

$$|\phi(x) - f(x)| \leq \varepsilon + K_\mathcal{H}\delta^\gamma \text{ for any } x \in [0,1],$$

*where*

$$\phi(x) := \mathrm{mid}\left(g(x - \delta), g(x), g(x + \delta)\right) \text{ for any } x \in \mathbb{R}.$$

*Proof of Theorem 2.3.* **Case 1:** $p \in [1, \infty)$. Let

$$\mathcal{N} \in \mathcal{T}_{d_x, d_x}(d_x, 1, 1, 5nK^{d_x n}, 2)$$

be as in Proposition A.3. By Proposition A.3 and noting that the Lebesgue measure of $\Omega([0,1]^{d_x \times n}, K, \delta)$ is at most $d_x n K \delta$, we have

$$\|\mathcal{N} - \boldsymbol{F}\|_{L^p([0,1]^{d_x \times n})}^p$$

$$= \int_{[0,1]^{d_x \times n}} \|\mathcal{N}(\boldsymbol{X}) - \boldsymbol{F}(\boldsymbol{X})\|_F^p \mathrm{d}\boldsymbol{X}$$

$$= \int_{\Omega([0,1]^{d_x \times n}, K, \delta)} \|\mathcal{N}(\boldsymbol{X}) - \boldsymbol{F}(\boldsymbol{X})\|_F^p \mathrm{d}\boldsymbol{X} + \int_{[0,1]^{d_x \times n} \setminus \Omega([0,1]^{d_x \times n}, K, \delta)} \|\mathcal{N}(\boldsymbol{X}) - \boldsymbol{F}(\boldsymbol{X})\|_F^p \mathrm{d}\boldsymbol{X}$$

$$\leq \int_{\Omega([0,1]^{d_x \times n}, K, \delta)} \|\mathcal{N}(\boldsymbol{X}) - \boldsymbol{F}(\boldsymbol{X})\|_F^p \mathrm{d}\boldsymbol{X}$$

$$+ \int_{[0,1]^{d_x \times n} \setminus \Omega([0,1]^{d_x \times n}, K, \delta)} (d_x n)^{\max\{0, \frac{p}{2}-1\}} \sum_{i=1}^{d_x} \sum_{j=1}^{n} |\mathcal{N}_{i,j}(\boldsymbol{X}) - F_{i,j}(\boldsymbol{X})|^p \mathrm{d}\boldsymbol{X}$$

$$\leq (2\sqrt{d_x n} K_{\mathcal{H}})^p \cdot d_x n K \delta + (d_x n)^{1+\max\{0, \frac{p}{2}-1\}} (K_{\mathcal{H}} (d_x n)^{\gamma/2} K^{-\gamma})^p$$

$$\leq 2^p (d_x n)^{2p} K_{\mathcal{H}}^p ((K\delta)^{\frac{1}{p}} + K^{-\gamma})^p,$$

using for the last inequality that $\gamma \in (0,1]$, $\max\{a,b\} \leq a+b$ for any $a, b \geq 0$, and $a^p + b^p \leq (a+b)^p$ for all $p \geq 1$ and $a, b \geq 0$. Hence,

$$\|\mathcal{N} - \boldsymbol{F}\|_{L^p([0,1]^{d_x \times n})} \leq 2(d_x n)^2 K_{\mathcal{H}} ((K\delta)^{\frac{1}{p}} + K^{-\gamma}).$$

Choosing $\delta \leq K^{-p\gamma-1}$ and $K \geq \varepsilon^{-1/\gamma}$ so that $K^{d_x n} = \lceil \varepsilon^{-\frac{d_x n}{\gamma}} \rceil$, we conclude

$$\|\mathcal{N} - \boldsymbol{F}\|_{L^p([0,1]^{d_x \times n})} \leq 4(d_x n)^2 K_{\mathcal{H}} \varepsilon$$

and

$$\mathcal{N} \in \mathcal{T}_{d_x, d_x}(d_x, 1, 1, 5n \lceil \varepsilon^{-\frac{d_x n}{\gamma}} \rceil, 2).$$

**Case 2:** $p = \infty$. By Proposition A.3, there exists a Transformer network

$$\widetilde{\mathcal{N}} \in \mathcal{T}_{d_x, d_x}(d_x, 1, 1, 5n K^{d_x n}, 2)$$

such that

$$|\widetilde{\mathcal{N}}_{i,j}(\boldsymbol{X}) - F_{i,j}(\boldsymbol{X})| \leq (d_x n)^{\gamma/2} K_{\mathcal{H}} K^{-\gamma}$$

for any $i \in [d_x]$, $j \in [n]$ and $\boldsymbol{X} \in [0,1]^{d_x \times n} \setminus \Omega([0,1]^{d_x \times n}, K, \delta)$. By Proposition A.4 (assume that $5n K^{d_x n} \geq 14 d_x$), there exists a new Transformer network

$$\mathcal{N} \in \mathcal{T}_{d_x, d_x}(5 d_x 3^{d_x n}, 3^{d_x n}, 1, 5n 3^{d_x n} K^{d_x n}, 2 + 2 d_x n),$$

such that

$$|\mathcal{N}_{i,j}(\boldsymbol{X}) - F_{i,j}(\boldsymbol{X})| \leq (d_x n)^{\gamma/2} K_{\mathcal{H}} K^{-\gamma} + d_x n K_{\mathcal{H}} \delta^{\gamma}$$

for any $i \in [d_x]$, $j \in [n]$ and $\boldsymbol{X} \in [0,1]^{d_x \times n}$. This implies

$$\|\mathcal{N} - \boldsymbol{F}\|_{L^\infty([0,1]^{d_x \times n})} = \sup_{\boldsymbol{X} \in [0,1]^{d_x \times n}} \|\mathcal{N}(\boldsymbol{X}) - \boldsymbol{F}(\boldsymbol{X})\|_F$$

$$\leq \sup_{\boldsymbol{X} \in [0,1]^{d_x \times n}} \sum_{i=1}^{d_x} \sum_{j=1}^{n} |\mathcal{N}_{i,j}(\boldsymbol{X}) - F_{i,j}(\boldsymbol{X})|$$

$$\leq \sum_{i=1}^{d_x} \sum_{j=1}^{n} \sup_{\boldsymbol{X} \in [0,1]^{d_x \times n}} |\mathcal{N}_{i,j}(\boldsymbol{X}) - F_{i,j}(\boldsymbol{X})|$$

$$\leq (d_x n)^{1+\gamma/2} K_{\mathcal{H}} K^{-\gamma} + (d_x n)^2 K_{\mathcal{H}} \delta^{\gamma}.$$

Choosing $\delta \in (0, \frac{1}{3K}]$ sufficiently small and $K \geq \varepsilon^{-1/\gamma}$ so that $K^{d_x n} = \lceil \varepsilon^{-\frac{d_x n}{\gamma}} \rceil$, we conclude

$$\|\mathcal{N} - \boldsymbol{F}\|_{L^\infty([0,1]^{d_x \times n})} \leq 4(d_x n)^2 K_{\mathcal{H}} \varepsilon$$

and

$$\mathcal{N} \in \mathcal{T}_{d_x, d_x}(5d_x 3^{d_x n}, 3^{d_x n}, 1, 5n3^{d_x n} \lceil \varepsilon^{-\frac{d_x n}{\gamma}} \rceil, 2 + 2d_x n).$$

This completes the proof. □

*Proof of Theorem 2.4.* Let $K$, $\mathbb{G}_K$ and $\omega_{\boldsymbol{G}}$ be as defined in the proof of Proposition A.3. We approximate the target function $\boldsymbol{F}$ by a piecewise constant function, where the value in each cell is given by the average of $\boldsymbol{F}$ over that cell. Define

$$\overline{\boldsymbol{F}}(\boldsymbol{X}) = \sum_{\boldsymbol{G} \in \mathbb{G}_K} \boldsymbol{F_G} \mathbb{1}_{\omega_{\boldsymbol{G}}}(\boldsymbol{X}),$$

where

$$[F_{\boldsymbol{G}}]_{i,j} = K^{d_x n} \int_{\omega_{\boldsymbol{G}}} F_{i,j}(\boldsymbol{X}) \mathrm{d}\boldsymbol{X}, \quad i \in [d_y], j \in [n].$$

Since each cell $\omega_{\boldsymbol{G}}$ is a bounded convex domain, Poincaré inequality gives, for any $p \in [1, \infty]$,

$$\|[F_{\boldsymbol{G}}]_{i,j} - F_{i,j}\|_{L^p(\omega_{\boldsymbol{G}})} \leq C \|\nabla F_{i,j}\|_{L^p(\omega_{\boldsymbol{G}})} K^{-1},$$

where $C$ is a constant depending only on $d_x n$, and $\|\nabla F\|_{L^p(\omega)}$ denotes the $L^p$-norm of the Frobenius norm of $\nabla F$ (see (Evans, 2010; Dekel & Leviatan, 2004)). Summing over all grid cells and using that $F_{i,j} \in \mathcal{W}^{1,p}([0,1]^{d_x \times n}, K_{\mathcal{W}})$ implies $\|\nabla F_{i,j}\|_{L^p([0,1]^{d_x \times n})} \leq (d_x n)^{\max\{0, \frac{1}{2} - \frac{1}{p}\}} K_{\mathcal{W}}$, we obtain

$$
\begin{aligned}
\|F_{i,j} - \overline{F}_{i,j}\|_{L^p([0,1]^{d_x \times n})} &= \begin{cases} \left( \sum_{\boldsymbol{G} \in \mathbb{G}_K} \|F_{i,j} - [F_{\boldsymbol{G}}]_{i,j}\|_{L^p(\omega_{\boldsymbol{G}})}^p \right)^{1/p} & \text{if } p < \infty \\ \sup_{\boldsymbol{G} \in \mathbb{G}_K} \|F_{i,j} - [F_{\boldsymbol{G}}]_{i,j}\|_{L^\infty(\omega_{\boldsymbol{G}})} & \text{if } p = \infty \end{cases} \\
&\leq C \|\nabla F_{i,j}\|_{L^p([0,1]^{d_x \times n})} K^{-1} \\
&\leq C(d_x n)^{\max\{0, \frac{1}{2} - \frac{1}{p}\}} K_{\mathcal{W}} K^{-1},
\end{aligned}
\tag{10}
$$

for any $p \in [1, \infty]$.

From **Step 2** and **Step 3** of Proposition A.3, there exists a Transformer network

$$\mathcal{N} \in \mathcal{T}_{d_x, d_x}(d_x, 1, 1, 5nK^{d_x n}, 2)$$

such that

$$\mathcal{N}(\boldsymbol{X}) = \overline{\boldsymbol{F}}(\boldsymbol{X}) \quad \text{for any } \boldsymbol{X} \in [0,1]^{d_x \times n} \setminus \Omega([0,1]^{d_x \times n}, K, \delta)$$

and

$$\|\mathcal{N}(\boldsymbol{X})\|_F \leq \sqrt{d_x n} K_{\mathcal{W}} \quad \text{for any } \boldsymbol{X} \in \mathbb{R}^{d_x \times n}.$$

Since the Lebesgue measure of $\Omega([0,1]^{d_x \times n}, K, \delta)$ is at most $d_x n K \delta$, for $p \in [1, \infty)$, we have

$$\|\mathcal{N} - \boldsymbol{F}\|_{L^p([0,1]^{d_x \times n})}^p$$

$$= \int_{[0,1]^{d_x \times n}} \|\mathcal{N}(\boldsymbol{X}) - \boldsymbol{F}(\boldsymbol{X})\|_F^p \mathrm{d}\boldsymbol{X}$$

$$= \int_{\Omega([0,1]^{d_x \times n}, K, \delta)} \|\mathcal{N}(\boldsymbol{X}) - \boldsymbol{F}(\boldsymbol{X})\|_F^p \mathrm{d}\boldsymbol{X} + \int_{[0,1]^{d_x \times n} \setminus \Omega([0,1]^{d_x \times n}, K, \delta)} \|\overline{\boldsymbol{F}}(\boldsymbol{X}) - \boldsymbol{F}(\boldsymbol{X})\|_F^p \mathrm{d}\boldsymbol{X}$$

$$\leq \int_{\Omega([0,1]^{d_x \times n}, K, \delta)} \|\mathcal{N}(\boldsymbol{X}) - \boldsymbol{F}(\boldsymbol{X})\|_F^p \mathrm{d}\boldsymbol{X}$$

$$+ \int_{[0,1]^{d_x \times n} \setminus \Omega([0,1]^{d_x \times n}, K, \delta)} (d_x n)^{\max\{0, \frac{p}{2}-1\}} \sum_{i=1}^{d_x} \sum_{j=1}^{n} |\overline{F}_{i,j}(\boldsymbol{X}) - F_{i,j}(\boldsymbol{X})|^p \mathrm{d}\boldsymbol{X}$$

$$\leq (2\sqrt{d_x n} K_{\mathcal{W}})^p \cdot d_x n K \delta + (d_x n)^{\max\{0, \frac{p}{2}-1\}} \sum_{i=1}^{d_x} \sum_{j=1}^{n} \left( C(d_x n)^{\max\{0, \frac{1}{2}-\frac{1}{p}\}} K_{\mathcal{W}} K^{-1} \right)^p$$

$$\leq (2C)^p (d_x n)^{2p} K_{\mathcal{W}}^p ((K\delta)^{\frac{1}{p}} + K^{-1})^p,$$

using for the last inequality that $p \geq 1$, $\max\{a, b\} \leq a + b$ for any $a, b \geq 0$, and $a^p + b^p \leq (a+b)^p$ for all $p \geq 1$ and $a, b \geq 0$. Hence,

$$\|\mathcal{N} - \boldsymbol{F}\|_{L^p([0,1]^{d_x \times n})} \leq 2C(d_x n)^2 K_{\mathcal{W}}((K\delta)^{\frac{1}{p}} + K^{-1}).$$

Choosing $\delta \leq K^{-p-1}$ and $K \geq \varepsilon^{-1}$ so that $K^{d_x n} = \lceil \varepsilon^{-d_x n} \rceil$, we conclude

$$\|\mathcal{N} - \boldsymbol{F}\|_{L^p([0,1]^{d_x \times n})} \leq 4C(d_x n)^2 K_{\mathcal{W}} \varepsilon$$

and

$$\mathcal{N} \in \mathcal{T}_{d_x, d_x}(d_x, 1, 1, 5n\lceil \varepsilon^{-d_x n} \rceil, 2).$$

This completes the proof. $\qquad \square$

**Lemma A.7.** *Let $d_x, d_y, r \in \mathbb{N}$ with $d_x \geq d_y$. Let $\{(\boldsymbol{x}_i, \boldsymbol{y}_i)\}_{i=1}^r$ be a set of input-output pairs such that $\boldsymbol{x}_i \in \mathbb{R}^{d_x}, \boldsymbol{y}_i \in \mathbb{R}^{d_y}, i \in [r]$ and $\boldsymbol{x}_i \neq \boldsymbol{x}_j$ if $i \neq j$. Then, there exists a feed-forward layer $\mathcal{F}^{(FF)} : \mathbb{R}^{d_x} \to \mathbb{R}^{d_x}$ with width at most $3r + 2d_x$ such that*

$$\mathcal{F}^{(FF)}(\boldsymbol{x}_i) = \begin{pmatrix} \boldsymbol{y}_i \\ \boldsymbol{0} \end{pmatrix} \quad \text{for all } i \in [r],$$

*and $\|\mathcal{F}^{(FF)}(\boldsymbol{z})\| \leq \max_i \|\boldsymbol{y}_i\|$ for any $\boldsymbol{z} \in \mathbb{R}^{d_x}$.*

*Proof.* Let $R > 0$ be determined later. Since $\boldsymbol{x}_i, i \in [r]$ are pairwise distinct, we can find $\boldsymbol{v} \in \mathbb{R}^{d_x}$ such that $\boldsymbol{v}^\top \boldsymbol{x}_i, i \in [r]$ are distinct. The existence of $\boldsymbol{v}$ can be found in Park et al. (2021, Lemma 13). We define

$$\boldsymbol{A}_i^{(1)} = R\boldsymbol{1}_3 \boldsymbol{v}^\top, \quad \boldsymbol{b}_i^{(1)} = \begin{pmatrix} -R\boldsymbol{v}^\top \boldsymbol{x}_i - 1 \\ -R\boldsymbol{v}^\top \boldsymbol{x}_i \\ -R\boldsymbol{v}^\top \boldsymbol{x}_i + 1 \end{pmatrix}, \quad \boldsymbol{A}_i^{(2)} = \begin{pmatrix} \boldsymbol{y}_i \\ \boldsymbol{0} \end{pmatrix} (1, -2, 1), \quad \boldsymbol{b}_i^{(2)} = \boldsymbol{0}.$$

Then, by direct calculation, we obtain

$$\boldsymbol{A}_i^{(2)} \sigma_R[\boldsymbol{A}_i^{(1)} \boldsymbol{x} + \boldsymbol{b}_i^{(1)}] + \boldsymbol{b}_i^{(2)}$$

$$= \begin{pmatrix} \boldsymbol{y}_i \\ \boldsymbol{0} \end{pmatrix} \left( \sigma_R[R\boldsymbol{v}^\top(\boldsymbol{x} - \boldsymbol{x}_i) - 1] - 2\sigma_R[R\boldsymbol{v}^\top(\boldsymbol{x} - \boldsymbol{x}_i)] + \sigma_R[R\boldsymbol{v}^\top(\boldsymbol{x} - \boldsymbol{x}_i) + 1] \right)$$

$$= \begin{pmatrix} \boldsymbol{y}_i \\ \boldsymbol{0} \end{pmatrix} I_i(\boldsymbol{x}),$$

where $I_i(\boldsymbol{x})$ is the hat function with $I_i(\boldsymbol{x}_i) = 1$ and $I_i(\boldsymbol{x}) = 0$ if $|\boldsymbol{v}^\top(\boldsymbol{x} - \boldsymbol{x}_i)| \geq 1/R$. To ensure the supports of $I_i(\boldsymbol{x})$ for all $i \in [r]$ are disjoint, we choose $R > 2/\min_{i \neq j} |\boldsymbol{v}^\top(\boldsymbol{x}_i - \boldsymbol{x}_j)|$. Define

$$
\boldsymbol{A}^{(1)} = \begin{pmatrix} \boldsymbol{A}_1^{(1)} \\ \vdots \\ \boldsymbol{A}_r^{(1)} \\ \boldsymbol{I}_{d_x} \\ -\boldsymbol{I}_{d_x} \end{pmatrix}, \quad \boldsymbol{b}^{(1)} = \begin{pmatrix} \boldsymbol{b}_1^{(1)} \\ \vdots \\ \boldsymbol{b}_r^{(1)} \\ \boldsymbol{0} \\ \boldsymbol{0} \end{pmatrix}, \quad \boldsymbol{A}^{(2)} = \left( \boldsymbol{A}_1^{(2)}, \ldots, \boldsymbol{A}_r^{(2)}, -\boldsymbol{I}_{d_x}, \boldsymbol{I}_{d_x} \right), \quad \boldsymbol{b}^{(2)} = \boldsymbol{0},
$$

and let

$$
\mathcal{F}^{(FF)}(\boldsymbol{x}) = \boldsymbol{x} + \boldsymbol{A}^{(2)} \sigma_R [\boldsymbol{A}^{(1)}\boldsymbol{x} + \boldsymbol{b}^{(1)}] + \boldsymbol{b}^{(2)}
$$
$$
= \sum_{i=1}^r \begin{pmatrix} \boldsymbol{y}_i \\ \boldsymbol{0} \end{pmatrix} I_i(\boldsymbol{x}).
$$

We complete the proof by verifying that

$$
\mathcal{F}^{(FF)}(\boldsymbol{x}_k) = \sum_{i=1}^r \begin{pmatrix} \boldsymbol{y}_i \\ \boldsymbol{0} \end{pmatrix} I_i(\boldsymbol{x}_k) = \begin{pmatrix} \boldsymbol{y}_k \\ \boldsymbol{0} \end{pmatrix}
$$

and

$$
\|\mathcal{F}^{(FF)}(\boldsymbol{x})\| = \left\| \sum_{i=1}^r \begin{pmatrix} \boldsymbol{y}_i \\ \boldsymbol{0} \end{pmatrix} I_i(\boldsymbol{x}) \right\|
$$
$$
\leq \max_i \|\boldsymbol{y}_i\| \left\| \sum_{i=1}^r I_i(\boldsymbol{x}) \right\|
$$
$$
\leq \max_i \|\boldsymbol{y}_i\|.
$$

$\square$

## A.2. Proof of Theorem 2.7

We introduce sample complexities, which measure the richness of the function class in different aspects, and use them to bound the generalization error.

**Definition A.8** (VC-dimension). Let $\mathcal{H}$ be a class of real-valued functions defined on $\Omega$. The VC-dimension of $\mathcal{H}$, denoted by $\mathrm{VCDim}(\mathcal{H})$, is the largest integer $N$ for which there exist points $x_1, \ldots, x_N \in \Omega$ such that

$$
|\{\mathrm{sgn}(h(x_1)), \ldots, \mathrm{sgn}(h(x_N)) : h \in \mathcal{H}\}| = 2^N.
$$

**Definition A.9** (Pseudo-dimension). Let $\mathcal{H}$ be a class of real-valued functions defined on $\Omega$. The pseudo-dimension of $\mathcal{H}$, denoted by $\mathrm{Pdim}(\mathcal{H})$, is the largest integer $N$ for which there exist points $x_1, \ldots, x_N \in \Omega$ and constants $c_1, \ldots, c_N \in \mathbb{R}$ such that

$$
|\{\mathrm{sgn}(h(x_1) - c_1), \ldots, \mathrm{sgn}(h(x_N) - c_N) : h \in \mathcal{H}\}| = 2^N.
$$

**Definition A.10** (Covering number). Let $\rho$ be a pseudo-metric on $\mathcal{M}$ and $S \subseteq \mathcal{M}$. For any $\delta > 0$, a set $A \subseteq \mathcal{M}$ is called a $\delta$-covering of $S$ if for any $x \in S$ there exists $y \in A$ such that $\rho(x, y) \leq \delta$. The $\delta$-covering number of $S$, denoted by $\mathcal{N}(\delta, S, \rho)$, is the minimum cardinality of any $\delta$-covering of $S$.

**Theorem A.11** (Theorem 8.14 of Anthony & Bartlett (1999)). *Let $h$ be a function from $\mathbb{R}^d \times \mathbb{R}^n$ to $\{0, 1\}$, determining the class*

$$
\mathcal{H} = \{x \mapsto h(a, x) : a \in \mathbb{R}^d\}.
$$

*Suppose that $h$ can be computed by an algorithm that takes as input the pair $(a, x) \in \mathbb{R}^d \times \mathbb{R}^n$ and returns $h(a, x)$ after no more than $t$ of the following operations:*

- *the exponential function $\alpha \mapsto e^\alpha$ on real numbers,*

- *the arithmetic operations $+$, $-$, $\times$, and $/$ on real numbers,*

- *jumps conditioned on $>$, $\geq$, $<$, $\leq$, $=$, and $\neq$ comparisons of real numbers, and*

- *output $0$ or $1$.*

*Then $\mathrm{VCdim}(\mathcal{H}) \leq t^2 d \left(d + 19 \log_2(9d)\right)$. Furthermore, if the $t$ steps include no more than $q$ in which the exponential function is evaluated, then*

$$\mathrm{VCdim}(\mathcal{H}) \leq (d(q+1))^2 + 11d(q+1)\left(t + \log_2(9d(q+1))\right).$$

Theorem A.11 gives bounds on the VC-dimension of a function class in terms of the number of arithmetic operations required to compute the functions. This result immediately implies a bound on the VC-dimension (or pseudo-dimension) for Transformer networks. By applying standard techniques in learning theory, one can further derive upper bounds for the covering number. The following lemma summarizes these bounds.

**Lemma A.12.** *Recall that $\mathcal{F} = \{f = \langle \mathcal{N}, \boldsymbol{E} \rangle : \mathcal{N} \in \mathcal{T}_{d_x, d_x}(D, H, S, W, L)\}$. Then the following bounds hold:*

- $\mathrm{VCdim}(\mathcal{F}) \lesssim (HS + W)^2 D^2 H^2 L^4$,

- $\mathrm{Pdim}(\mathcal{F}) \lesssim (HS + W)^2 D^2 H^2 L^4$,

- $\sup_{\mathcal{X}} \log \mathcal{N}(\delta, \mathcal{C}_K \mathcal{F}, d_{\mathcal{X},\infty}) \lesssim (HS + W)^2 D^2 H^2 L^4 \log \frac{mK}{\delta}$, *where $\mathcal{X} = \{\boldsymbol{X}_i\}_{i=1}^m$ and*

$$d_{\mathcal{X},\infty}(f, g) = \max_{i \in [m]} |f(\boldsymbol{X}_i) - g(\boldsymbol{X}_i)|.$$

*We hide constants that depend on $d_x$ and $n$.*

*Proof.* Recall that $\mathcal{F} = \{f = \langle \mathcal{N}, \boldsymbol{E} \rangle : \mathcal{N} \in \mathcal{T}_{d_x, d_x}(D, H, S, W, L)\}$. By carefully counting the computational steps required to evaluate any $f \in \mathcal{F}$, we deduce that

- the total number of parameters is bounded by $d \lesssim (HS + W)DL$,

- the total number of computational operations is bounded by $t \lesssim L(HDSn + HSn^2 + WDn)$,

- the number of evaluations of the exponential function is bounded by $q \lesssim LHn^2$.

Theorem A.11 immediately implies that

$$\begin{aligned}
\mathrm{VCdim}(\mathcal{F}) &\leq (d(q+1))^2 + 11d(q+1)\left(t + \log_2(9d(q+1))\right) \\
&\lesssim (HS + W)^2 D^2 H^2 L^4,
\end{aligned}$$

where we use $\log(x) \leq x$ for $x \geq 1$ and suppress constants that depend on $d_x$ and $n$.

For the pseudo-dimension, note that by definition $\mathrm{VCdim}(\{f(x) - r : f \in \mathcal{F}, r \in \mathbb{R}\}) = \mathrm{Pdim}(\mathcal{F})$. Using the same reasoning as above, we have

$$\mathrm{Pdim}(\mathcal{F}) \lesssim (HS + W)^2 D^2 H^2 L^4,$$

again by Theorem A.11.

Finally, by Theorem 12.2 of Anthony & Bartlett (1999), we have

$$\begin{aligned}
\log \mathcal{N}(\delta, \mathcal{C}_K \mathcal{F}, d_{\mathcal{X},\infty}) &\leq \mathrm{Pdim}(\mathcal{C}_K \mathcal{F}) \log \frac{emK}{\delta} \\
&\leq \mathrm{Pdim}(\mathcal{F}) \log \frac{emK}{\delta} \\
&\lesssim (HS + W)^2 D^2 H^2 L^4 \log \frac{mK}{\delta}.
\end{aligned}$$

Taking the supremum over all possible sample sets $\mathcal{X}$ completes the proof. $\qquad\square$

*Proof of Theorem 2.7.* Let $\mathcal{X}$ and $d_{\mathcal{X},\infty}$ be defined as in Lemma A.12. Similar to the proof of Jiao et al. (2024b, Theorem 5), given a random sample $\mathcal{D}_m = \{(\boldsymbol{x}_i, y_i)\}_{i=1}^m$, the excess risk can be decomposed as

$$\mathbb{E}_{\mathcal{D}_m}[\mathcal{R}(\mathcal{C}_{B_m}\hat{f}_m) - \mathcal{R}(f^*)] \lesssim \mathcal{E}_{app} + \mathcal{E}_{gen} + \mathcal{E}_{den},$$

where

$$\mathcal{E}_{app} := \inf_{f \in \mathcal{F}} \mathbb{E}[(f - f^*)^2],$$

$$\mathcal{E}_{gen} := \frac{B_m^2 k_m}{m} \sup_{|\mathcal{X}|=m} \log \mathcal{N}(m^{-1}, \mathcal{C}_{B_m}\mathcal{F}, d_{\mathcal{X},\infty}),$$

$$\mathcal{E}_{den} := \frac{B_m^2 m}{k_m} \beta(k_m).$$

Here, $k_m \in \mathbb{N}$ is a parameter to be chosen. It can be seen that the excess risk is bounded by the sum of the approximation error $\mathcal{E}_{app}$, the generalization error $\mathcal{E}_{gen}$, and the dependence error $\mathcal{E}_{den}$. Note that as $k_m$ increases, $\mathcal{E}_{den}$ decreases due to the monotonic decrease of the $\beta$-mixing coefficient $\beta(k_m)$, whereas $\mathcal{E}_{gen}$ increases. Besides, if we select a larger hypothesis class $\mathcal{F}$, then $\mathcal{E}_{app}$ decreases but $\mathcal{E}_{gen}$ increases because the covering number grows with the size of the hypothesis class. Therefore, to obtain a better rate of convergence, we must carefully trade off these three errors by choosing an appropriate hypothesis class $\mathcal{F}$ and tuning the parameter $k_m$.

Since by assumption the density of $\Pi$ is upper bounded, Theorem 2.3 implies that

$$\mathcal{E}_{app} \leq \inf_{f \in \mathcal{F}} \|f - f^*\|_{L^2([0,1]^{d_x \times n})}^2 \lesssim \varepsilon^2,$$

where the hypothesis class

$$\mathcal{F} = \mathcal{F}(D_m \lesssim 1, H_m \lesssim 1, S_m \lesssim 1, W_m \lesssim \varepsilon^{-\frac{d_x n}{\gamma}}, L_m \lesssim 1).$$

Then by Lemma A.12,

$$\mathcal{E}_{gen} \lesssim \frac{B_m^2 k_m}{m} (HS + W)^2 D^2 H^2 L^4 \log(m^2 B_m)$$

$$\lesssim \frac{(\log m)^3 k_m}{m} \varepsilon^{-\frac{2d_x n}{\gamma}},$$

where we take $B_m \asymp \log m$.

We now consider three cases for the sequence $\{\boldsymbol{x}_i\}_{i=1}^m$.

**Case 1:** if $\{\boldsymbol{x}_i\}_{i=1}^m$ is geometrically $\beta$-mixing, i.e., $\beta(k_m) \leq \beta_0 \exp\left(-\beta_1 k_m^r\right)$ for some $r, \beta_0, \beta_1 > 0$, we set $k_m \asymp (\log m)^{1/r}$ so that $\beta(k_m) \lesssim 1/m^{100}$. Then,

$$\mathbb{E}_{\mathcal{D}_m}[\mathcal{R}(\mathcal{C}_{B_m}\hat{f}_m) - \mathcal{R}(f^*)] \lesssim \varepsilon^2 + \frac{(\log m)^{3+1/r}}{m} \varepsilon^{-\frac{2d_x n}{\gamma}}$$

$$\lesssim m^{-\frac{\gamma}{\gamma + d_x n}} (\log m)^{3+1/r},$$

where $\varepsilon$ is chosen as $\varepsilon \asymp m^{-\frac{\gamma}{2\gamma + 2d_x n}}$.

**Case 2:** if $\{\boldsymbol{x}_i\}_{i=1}^m$ is algebraically $\beta$-mixing, that is, $\beta(k_m) \leq \beta_0/k_m^r$ for some $r, \beta_0 > 0$, then

$$\mathbb{E}_{\mathcal{D}_m}[\mathcal{R}(\mathcal{C}_{B_m}\hat{f}_m) - \mathcal{R}(f^*)] \lesssim \varepsilon^2 + \frac{(\log m)^3 k_m}{m} \varepsilon^{-\frac{2d_x n}{\gamma}} + \frac{(\log m)^2 m}{k_m^{r+1}}$$

$$\lesssim m^{-\frac{r\gamma}{(r+2)\gamma + (r+1)d_x n}} (\log m)^3,$$

where we use the AM-GM inequality and choose $\varepsilon \asymp m^{-\frac{r\gamma}{2(r+2)\gamma + 2(r+1)d_x n}}$ and $k_m \asymp m^{\frac{2\gamma + d_x n}{(r+2)\gamma + (r+1)d_x n}}$.

**Case 3:** if $\{x_i\}_{i=1}^m$ is a sequence of i.i.d. random variables, then $\beta(k_m) = 0$ for all $k_m \geq 1$. This implies

$$\mathbb{E}_{\mathcal{D}_m}[\mathcal{R}(\mathcal{C}_{B_m} \hat{f}_m) - \mathcal{R}(f^*)] \lesssim \varepsilon^2 + \frac{(\log m)^3}{m} \varepsilon^{-\frac{2d_x n}{\gamma}}$$

$$\lesssim m^{-\frac{\gamma}{\gamma + d_x n}} (\log m)^3,$$

where we choose $\varepsilon \asymp m^{-\frac{\gamma}{2\gamma + 2d_x n}}$. So we complete the proof.

$\square$

### A.3. Proof of Theorem 2.8

The original Kolmogorov-Arnold representation theorem states that for any continuous function $f : [0, 1]^d \to \mathbb{R}$, there exist univariate continuous functions $g_q, \psi_{p,q}$ such that

$$f(x_1, \ldots, x_d) = \sum_{q=0}^{2d} g_q \left( \sum_{p=1}^{d} \psi_{p,q}(x_p) \right).$$

Schmidt-Hieber (2021) derived modifications of this representation that transfer smoothness properties of the represented function to the outer function.

**Proposition A.13** (Theorem 2 of Schmidt-Hieber (2021))**.** *For any fixed dimension $d \geq 2$, there exists a monotone function $\phi : [0, 1] \to \mathcal{C}$ (the Cantor set) such that for any function $f \in \mathcal{H}^{\gamma}([0, 1]^d, K_{\mathcal{H}})$ with some $\gamma \in (0, 1]$, we can find a function $g \in \mathcal{H}^{\frac{\gamma \log 2}{d \log 3}}(\mathcal{C}, 2\sqrt{d}K_{\mathcal{H}})$ such that*

$$f(x_1, \ldots, x_d) = g \left( 3 \sum_{p=1}^{d} 3^{-p} \phi(x_p) \right). \tag{11}$$

*Moreover, for any $x \in [0, 1]$ with its binary representation $x = [0.a_1^x a_2^x \ldots]_2$, the function $\phi$ is given explicitly by*

$$\phi(x) = \sum_{j=1}^{\infty} \frac{2a_j^x}{3^{1+d(j-1)}} = [0.(2a_1^x)\underbrace{0\ldots\ldots0}_{(d-1)\text{-times}}(2a_2^x)\underbrace{0\ldots\ldots0}_{(d-1)\text{-times}}\ldots]_3,$$

*where $[\cdot]_B$ denotes the $B$-adic expansion of a real number.*

We note that a given real number can have multiple $B$-adic representations (for example, $[1]_{10} = [0.999\ldots]_{10}$), which may make $\phi$ not well-defined. To eliminate this ambiguity, we adopt the convention of using a unique $B$-adic representation for all real numbers. Observe that the argument of $g$ in (11) satisfies

$$3 \sum_{p=1}^{d} 3^{-p} \phi(x_p) = [0.(2a_1^{x_1})(2a_1^{x_2})\ldots(2a_1^{x_d})(2a_2^{x_1})\ldots]_3. \tag{12}$$

By construction, the Cantor set consists precisely of those numbers in $[0, 1]$ whose ternary expansion contains only the digits $0$ and $2$. This shows that the mapping $3 \sum_{p=1}^{d} 3^{-p} \phi(x_p)$ indeed defines a bijection between $[0, 1]^d$ and the Cantor set $\mathcal{C}$. Additionally, an approximation of $\phi$ with a truncation parameter $K$ is defined by

$$\phi_K(x) := \sum_{j=1}^{K} \frac{2a_j^x}{3^{1+d(j-1)}}, \tag{13}$$

which will be used in our construction.

*Proof of Theorem 2.8.* By Proposition A.13 and the fact that $\phi_K$ approximates $\phi$, there exists a function $\boldsymbol{G} : \mathbb{R}^{d_x \times n} \to \mathbb{R}^{d_x \times n}$ with each entry $G_{u,v} \in \mathcal{H}^{\frac{\gamma \log 2}{d_x n \log 3}}(\mathcal{C}, 2\sqrt{d_x n}K_{\mathcal{H}})$ such that

$$\boldsymbol{F}(\boldsymbol{X}) = \boldsymbol{G} \left( 3 \sum_{p=1}^{d_x} \sum_{q=1}^{n} a_{p,q} \phi(X_{p,q}) \right) \approx \boldsymbol{G} \left( 3 \sum_{p=1}^{d_x} \sum_{q=1}^{n} a_{p,q} \phi_K(X_{p,q}) \right). \tag{14}$$

We will construct a generalized Transformer network that approximates the latter mapping. In the proof below, for simplicity, we omit the placeholder zeros used for alignment.

**Step 1:** We first show that there exist $2K + 2$ generalized feed-forward layers $\mathcal{F}_1^{(GFF)}, \ldots, \mathcal{F}_{2K+2}^{(GFF)}$ such that

$$\mathcal{F}_{2K+2}^{(GFF)} \circ \cdots \circ \mathcal{F}_1^{(GFF)} : \boldsymbol{X} \mapsto \boldsymbol{Z}_1,$$

where

$$\boldsymbol{Z}_1 = \begin{pmatrix} 3\sum_{p=1}^{d_x} a_{p,1}\widetilde{\phi}_K(X_{p,1}) & 3\sum_{p=1}^{d_x} a_{p,2}\widetilde{\phi}_K(X_{p,2}) & \cdots & 3\sum_{p=1}^{d_x} a_{p,n}\widetilde{\phi}_K(X_{p,n}) \\ \vdots & \vdots & & \vdots \\ 3\sum_{p=1}^{d_x} a_{p,1}\widetilde{\phi}_K(X_{p,1}) & 3\sum_{p=1}^{d_x} a_{p,2}\widetilde{\phi}_K(X_{p,2}) & \cdots & 3\sum_{p=1}^{d_x} a_{p,n}\widetilde{\phi}_K(X_{p,n}) \end{pmatrix}.$$

Here, $a_{p,q} = \frac{1}{3^{(q-1)d_x+p}}$ and $\widetilde{\phi}_K$ is a function that satisfies

$$\widetilde{\phi}_K(x) = \begin{cases} 0, & \text{if } x < 0, \\ \phi_K(x), & \text{if } x \in \Omega_K \subseteq [0,1], \\ 1, & \text{if } x > 1, \end{cases}$$

where $\phi_K$ is defined in (13) and $\Omega_K \subseteq [0,1]$ has Lebesgue measure at least $1 - 2^{-K\gamma p}$. Schmidt-Hieber (2021, Theorem 3) guarantees the existence of an FNN $\widetilde{\phi}_K \in \mathcal{FNN}_{1,1}(4, 2K)$ with the above properties.

By parallel computation, we can construct an FNN $\widetilde{\mathcal{N}}_1 \in \mathcal{FNN}_{1,1}(4n, 2K)$ such that

$$\widetilde{\mathcal{N}}_1(x) = \left(1, 3^{-d_x}, 3^{-2d_x}, \ldots, 3^{-(n-1)d_x}\right) \begin{pmatrix} \widetilde{\phi}_K(x) \\ \widetilde{\phi}_K(x-2) \\ \widetilde{\phi}_K(x-4) \\ \vdots \\ \widetilde{\phi}_K(x-2(n-1)) \end{pmatrix}$$

$$= \sum_{q=1}^{n} 3^{-(q-1)d_x}\widetilde{\phi}_K(x-2(q-1)).$$

If $x \in [2(j-1), 2j-1]$ for some $j \in [n]$, then

$$\widetilde{\mathcal{N}}_1(x) = \sum_{q=1}^{j-1} 3^{-(q-1)d_x}\widetilde{\phi}_K(x-2(q-1)) + 3^{-(j-1)d_x}\widetilde{\phi}_K(x-2(j-1))$$

$$+ \sum_{q=j+1}^{n} 3^{-(q-1)d_x}\widetilde{\phi}_K(x-2(q-1))$$

$$= \sum_{q=1}^{j-1} 3^{-(q-1)d_x} + 3^{-(j-1)d_x}\widetilde{\phi}_K(x-2(j-1))$$

$$= 3^{-(j-1)d_x}\widetilde{\phi}_K(x-2(j-1)) + b_j,$$

where we define $b_1 = 0$ and $b_j = \sum_{q=1}^{j-1} 3^{-(q-1)d_x}$ for $j \geq 2$.

Now consider $\boldsymbol{x} \in \mathbb{R}^{d_x}$. Fixing $j \in [n]$, if $x_i \in [2(j-1), 2j-1]$ for all $i \in [d_x]$, we can construct an FNN $\widetilde{\mathcal{N}}_2 \in$

$\mathcal{FNN}_{d_x,d_x}(4d_xn, 2K)$ such that

$$\widetilde{\mathcal{N}}_2(\boldsymbol{x}) = \mathbf{1}_{d_x}\left(1, 3^{-1}, \ldots, 3^{1-d_x}\right)\begin{pmatrix}\widetilde{\mathcal{N}}_1(x_1)\\ \widetilde{\mathcal{N}}_1(x_2)\\ \vdots\\ \widetilde{\mathcal{N}}_1(x_{d_x})\end{pmatrix}$$

$$= \mathbf{1}_{d_x}\left(1, 3^{-1}, \ldots, 3^{1-d_x}\right)\begin{pmatrix}3^{-(j-1)d_x}\widetilde{\phi}_K(x_1 - 2(j-1)) + b_j\\ 3^{-(j-1)d_x}\widetilde{\phi}_K(x_2 - 2(j-1)) + b_j\\ \vdots\\ 3^{-(j-1)d_x}\widetilde{\phi}_K(x_{d_x} - 2(j-1)) + b_j\end{pmatrix}$$

$$= \left(\sum_{p=1}^{d_x} 3^{1-p-(j-1)d_x}\widetilde{\phi}_K(x_p - 2(j-1))\right)\mathbf{1}_{d_x} + \left(b_j\sum_{p=1}^{d_x} 3^{1-p}\right)\mathbf{1}_{d_x}$$

$$= \left(3\sum_{p=1}^{d_x} a_{p,j}\widetilde{\phi}_K(x_p - 2(j-1))\right)\mathbf{1}_{d_x} + c_j\mathbf{1}_{d_x},$$

where we define $c_j = b_j\sum_{p=1}^{d_x} 3^{1-p}$. Using that $X_{p,j} \in [0,1]$ implies $X_{p,j} + 2(j-1) \in [2(j-1), 2j-1]$, set $\boldsymbol{x} = \boldsymbol{X}_{:,j} + 2(j-1)\mathbf{1}_{d_x}$ in the above equality to obtain

$$\widetilde{\mathcal{N}}_2(\boldsymbol{X}_{:,j} + 2(j-1)\mathbf{1}_{d_x}) = \left(3\sum_{p=1}^{d_x} a_{p,j}\widetilde{\phi}_K(X_{p,j})\right)\mathbf{1}_{d_x} + c_j\mathbf{1}_{d_x}.$$

By Lemma A.1, there exist $2K$ feed-forward layers $\mathcal{F}_2^{(FF)}, \ldots, \mathcal{F}_{2K+1}^{(FF)}$, each with width at most $3 \cdot 4d_xn = 12d_xn$, such that

$$\mathcal{F}_{2K+1}^{(FF)} \circ \cdots \circ \mathcal{F}_2^{(FF)}\left(\boldsymbol{X}_{:,1}, \ldots, \boldsymbol{X}_{:,n} + 2(n-1)\mathbf{1}_{d_x}\right) = \left(\widetilde{\mathcal{N}}_2(\boldsymbol{X}_{:,1}), \ldots, \widetilde{\mathcal{N}}_2(\boldsymbol{X}_{:,n} + 2(n-1)\mathbf{1}_{d_x})\right),$$

where we omit placeholder zeros for simplicity. Finally, to add and then remove the bias terms, we use two generalized feed-forward layers. We define

$$\mathcal{F}_1^{(GFF)}(\boldsymbol{X}_{:,1}, \ldots, \boldsymbol{X}_{:,n}) := (\boldsymbol{X}_{:,1}, \ldots, \boldsymbol{X}_{:,n} + 2(n-1)\mathbf{1}_{d_x})$$

and

$$\mathcal{F}_{2K+2}^{(GFF)}(\boldsymbol{Z}_{:,1}, \ldots, \boldsymbol{Z}_{:,n}) := (\boldsymbol{Z}_{:,1} - c_1\mathbf{1}_{d_x}, \ldots, \boldsymbol{Z}_{:,n} - c_n\mathbf{1}_{d_x}).$$

It is straightforward to verify that

$$\mathcal{F}_{2K+2}^{(GFF)} \circ \mathcal{F}_{2K+1}^{(FF)} \circ \cdots \circ \mathcal{F}_2^{(FF)} \circ \mathcal{F}_1^{(GFF)}(\boldsymbol{X})$$
$$= \mathcal{F}_{2K+2}^{(GFF)} \circ \mathcal{F}_{2K+1}^{(FF)} \circ \cdots \circ \mathcal{F}_2^{(FF)}\left(\boldsymbol{X}_{:,1}, \ldots, \boldsymbol{X}_{:,n} + 2(n-1)\mathbf{1}_{d_x}\right)$$
$$= \mathcal{F}_{2K+2}^{(GFF)}\left(\widetilde{\mathcal{N}}_2(\boldsymbol{X}_{:,1}), \ldots, \widetilde{\mathcal{N}}_2(\boldsymbol{X}_{:,n} + 2(n-1)\mathbf{1}_{d_x})\right)$$
$$= \mathcal{F}_{2K+2}^{(GFF)}\left(\left(3\sum_{p=1}^{d_x} a_{p,1}\widetilde{\phi}_K(X_{p,1})\right)\mathbf{1}_{d_x} + c_1\mathbf{1}_{d_x}, \ldots, \left(3\sum_{p=1}^{d_x} a_{p,n}\widetilde{\phi}_K(X_{p,n})\right)\mathbf{1}_{d_x} + c_n\mathbf{1}_{d_x}\right)$$
$$= \left(\left(3\sum_{p=1}^{d_x} a_{p,1}\widetilde{\phi}_K(X_{p,1})\right)\mathbf{1}_{d_x}, \ldots, \left(3\sum_{p=1}^{d_x} a_{p,n}\widetilde{\phi}_K(X_{p,n})\right)\mathbf{1}_{d_x}\right)$$
$$= \boldsymbol{Z}_1.$$

**Step 2:** We show that there exist a generalized self-attention layer $\mathcal{F}^{(GSA)}$ and a generalized feed-forward layer $\mathcal{F}_{2K+3}^{(GFF)}$ such that

$$\mathcal{F}_{2K+3}^{(GFF)} \circ \mathcal{F}^{(GSA)} : \begin{pmatrix} \mathbf{Z}_1 \\ \mathbf{O} \end{pmatrix} \mapsto \begin{pmatrix} \mathbf{Z}_2 \\ \mathbf{O} \end{pmatrix},$$

where

$$\mathbf{Z}_2 = \begin{pmatrix} 3\sum_{p=1}^{d_x}\sum_{q=1}^{n} a_{p,q}\widetilde{\phi}_K(X_{p,q}) & 3\sum_{p=1}^{d_x}\sum_{q=1}^{n} a_{p,q}\widetilde{\phi}_K(X_{p,q}) + 2 & \cdots & 3\sum_{p=1}^{d_x}\sum_{q=1}^{n} a_{p,q}\widetilde{\phi}_K(X_{p,q}) + 2(n-1) \\ \vdots & \vdots & & \vdots \\ 3\sum_{p=1}^{d_x}\sum_{q=1}^{n} a_{p,q}\widetilde{\phi}_K(X_{p,q}) & 3\sum_{p=1}^{d_x}\sum_{q=1}^{n} a_{p,q}\widetilde{\phi}_K(X_{p,q}) + 2 & \cdots & 3\sum_{p=1}^{d_x}\sum_{q=1}^{n} a_{p,q}\widetilde{\phi}_K(X_{p,q}) + 2(n-1) \end{pmatrix}.$$

Note that $\mathbf{Z}_2$ is obtained by summing the columns of $\mathbf{Z}_1$ and then adding different bias terms to each column.

We now prove the existence of such layers by first considering a standard self-attention layer. In fact, we only require the softmax function to compute the column average, so it can be replaced by a generalized self-attention layer. We define a self-attention layer by choosing the parameters as follows:

$$H = 1, \quad S = d_x, \quad \mathbf{W}^{(O)} = n\begin{pmatrix} \mathbf{O}_{d_x} \\ \mathbf{I}_{d_x} \end{pmatrix}, \quad \mathbf{W}^{(V)} = (\mathbf{I}_{d_x}, \mathbf{O}_{d_x}), \quad \mathbf{W}^{(K)} = \mathbf{O}, \quad \mathbf{W}^{(Q)} = \mathbf{O}.$$

Then, by direct calculation based on the definition, we have

$$\mathcal{F}^{(SA)}\begin{pmatrix} \mathbf{Z}_1 \\ \mathbf{O} \end{pmatrix} = \begin{pmatrix} \left(3\sum_{p=1}^{d_x} a_{p,1}\widetilde{\phi}_K(X_{p,1})\right)\mathbf{1}_{d_x} & \cdots & \left(3\sum_{p=1}^{d_x} a_{p,n}\widetilde{\phi}_K(X_{p,n})\right)\mathbf{1}_{d_x} \\ \left(3\sum_{p=1}^{d_x}\sum_{q=1}^{n} a_{p,q}\widetilde{\phi}_K(X_{p,q})\right)\mathbf{1}_{d_x} & \cdots & \left(3\sum_{p=1}^{d_x}\sum_{q=1}^{n} a_{p,q}\widetilde{\phi}_K(X_{p,q})\right)\mathbf{1}_{d_x} \end{pmatrix}.$$

Next, we define a generalized feed-forward layer with the following parameters:

$$\mathbf{W}^{(1)} = \begin{pmatrix} \mathbf{I}_{d_x} & \mathbf{O}_{d_x} \\ -\mathbf{I}_{d_x} & \mathbf{O}_{d_x} \\ \mathbf{O}_{d_x} & \mathbf{I}_{d_x} \\ \mathbf{O}_{d_x} & -\mathbf{I}_{d_x} \end{pmatrix}, \quad \mathbf{B}^{(1)} = \mathbf{O},$$

$$\mathbf{W}^{(2)} = \begin{pmatrix} -\mathbf{I}_{d_x} & \mathbf{I}_{d_x} & \mathbf{I}_{d_x} & -\mathbf{I}_{d_x} \\ \mathbf{O}_{d_x} & \mathbf{O}_{d_x} & -\mathbf{I}_{d_x} & \mathbf{I}_{d_x} \end{pmatrix}, \quad \mathbf{B}^{(2)} = \begin{pmatrix} \mathbf{0}_{d_x} & 2\mathbf{1}_{d_x} & \cdots & 2(n-1)\mathbf{1}_{d_x} \\ \mathbf{0}_{d_x} & \mathbf{0}_{d_x} & \cdots & \mathbf{0}_{d_x} \end{pmatrix}.$$

It can then be verified that

$$\mathcal{F}_{2K+3}^{(GFF)} \circ \mathcal{F}^{(SA)}\begin{pmatrix} \mathbf{Z}_1 \\ \mathbf{O} \end{pmatrix} = \begin{pmatrix} \mathbf{Z}_2 \\ \mathbf{O} \end{pmatrix},$$

where we have used the identity $x = \sigma_R[x] - \sigma_R[-x]$.

**Step 3:** We construct a generalized feed-forward layer $\mathcal{F}_{2K+4}^{(GFF)}$ interpolating the outer function $\mathbf{G}$ in (14) at the $2^{d_x nK} + 1$ interpolation points

$$3\sum_{p=1}^{d_x}\sum_{q=1}^{n} a_{p,q}\phi_K(X_{p,q}) \in \left\{ \sum_{j=1}^{d_x nK} 2t_j 3^{-j} : (t_1, \ldots, t_{d_x nK}) \in \{0,1\}^{d_x nK} \right\} \bigcup \{1\}.$$

Denote these points by $0 =: s_0 < s_1 < \cdots < s_{2^{d_x nK}-1} < s_{2^{d_x nK}} := 1$ and fix $u \in [d_x]$. For any $x \in \mathbb{R}$, we define a scalar

function

$$
\widetilde{G}_u(x) := G_{u,1}(s_0) + \sum_{j=1}^{2^{d_x n K}} \frac{G_{u,1}(s_j) - G_{u,1}(s_{j-1})}{s_j - s_{j-1}} \left(\sigma_R[x - s_{j-1}] - \sigma_R[x - s_j]\right)
$$

$$
+ \frac{G_{u,2}(s_0) - G_{u,1}(s_{2^{d_x n K}})}{s_0 + 2 - s_{2^{d_x n K}}} \left(\sigma_R[x - s_{2^{d_x n K}}] - \sigma_R[x - (s_0 + 2)]\right)
$$

$$
+ \sum_{j=1}^{2^{d_x n K}} \frac{G_{u,2}(s_j) - G_{u,2}(s_{j-1})}{s_j - s_{j-1}} \left(\sigma_R[x - (s_{j-1} + 2)] - \sigma_R[x - (s_j + 2)]\right) + \cdots
$$

$$
+ \frac{G_{u,n}(s_0) - G_{u,n-1}(s_{2^{d_x n K}})}{s_0 + 2 - s_{2^{d_x n K}}} \left(\sigma_R[x - (s_{2^{d_x n K}} + 2(n - 2))] - \sigma_R[x - (s_0 + 2(n - 1))]\right)
$$

$$
+ \sum_{j=1}^{2^{d_x n K}} \frac{G_{u,n}(s_j) - G_{u,n}(s_{j-1})}{s_j - s_{j-1}} \left(\sigma_R[x - (s_{j-1} + 2(n - 1))] - \sigma_R[x - (s_j + 2(n - 1))]\right)
$$

$$
= G_{u,1}(s_0)
$$

$$
+ \sum_{v=1}^{n} \sum_{j=1}^{2^{d_x n K}} \frac{G_{u,v}(s_j) - G_{u,v}(s_{j-1})}{s_j - s_{j-1}} \left(\sigma_R[x - (s_{j-1} + 2(v - 1))] - \sigma_R[x - (s_j + 2(v - 1))]\right)
$$

$$
+ \sum_{v=2}^{n} \frac{G_{u,v}(s_0) - G_{u,v-1}(s_{2^{d_x n K}})}{s_0 + 2 - s_{2^{d_x n K}}} \left(\sigma_R[x - (s_{2^{d_x n K}} + 2(v - 2))] - \sigma_R[x - (s_0 + 2(v - 1))]\right).
$$

In other words, $\widetilde{G}_u$ is defined as the piecewise linear interpolation of the points

$$
\left\{(s_j + 2(v - 1), G_{u,v}(s_j)) : j = 0, 1, \ldots, 2^{d_x n K}, \ v = 1, \ldots, n\right\},
$$

with the function being constant outside the interval $[0, 2n - 1]$. We observe that

- $\widetilde{G}_u(s_j + 2(v - 1)) = G_{u,v}(s_j)$ for every $j \in \{0\} \cup [2^{d_x n K}]$, $u \in [d_x]$, $v \in [n]$,

- $\|\widetilde{G}_u\|_{L^\infty(\mathbb{R})} \leq \max_{v \in [n]} \|G_{u,v}\|_{L^\infty(\mathcal{C})}$,

- $\widetilde{G}_u \in \mathcal{FNN}_{1,1}(n(2^{d_x n K} + 1), 1)$.

By stacking the functions $\widetilde{G}_u$ for $u \in [d_x]$ vertically, we obtain a feed-forward layer, with width at most $d_x n(2^{d_x n K} + 1) + 2d_x$, such that

$$
\mathcal{F}_{2K+4}^{(GFF)}(\boldsymbol{Z}) = \begin{pmatrix} \widetilde{G}_1(Z_{1,1}) & \widetilde{G}_1(Z_{1,2}) & \cdots & \widetilde{G}_1(Z_{1,n}) \\ \widetilde{G}_2(Z_{2,1}) & \widetilde{G}_2(Z_{2,2}) & \cdots & \widetilde{G}_2(Z_{2,n}) \\ \vdots & \vdots & & \vdots \\ \widetilde{G}_{d_x}(Z_{d_x,1}) & \widetilde{G}_{d_x}(Z_{d_x,2}) & \cdots & \widetilde{G}_{d_x}(Z_{d_x,n}) \end{pmatrix}.
$$

Together with **Step 1** and **Step 2**, we define the overall generalized Transformer network as

$$
\mathcal{N} := \mathcal{E}_{out} \circ \mathcal{F}_{2K+4}^{(GFF)} \circ \mathcal{F}_{2K+3}^{(GFF)} \circ \mathcal{F}^{(GSA)} \circ \mathcal{F}_{2K+2}^{(GFF)} \circ \cdots \circ \mathcal{F}_1^{(GFF)} \circ \mathcal{E}_{in}
$$
$$
\in \mathcal{GT}_{d_x,d_x}(D = 4d_x n, H = 1, S = d_x, W = d_x n(2^{d_x n K} + 1) + 2d_x, L = 2K + 4),
$$

(15)

where $\mathcal{E}_{in}$ and $\mathcal{E}_{out}$ are appropriately chosen to add and remove zeros to match the hidden dimension $D$. In particular, we

have

$$\mathcal{N}(\boldsymbol{X})$$
$$= \mathcal{E}_{out} \circ \mathcal{F}_{2K+4}^{(GFF)} \circ \mathcal{F}_{2K+3}^{(GFF)} \circ \mathcal{F}^{(GSA)} \circ \mathcal{F}_{2K+2}^{(GFF)} \circ \cdots \circ \mathcal{F}_1^{(GFF)} \circ \mathcal{E}_{in}(\boldsymbol{X})$$
$$= \mathcal{E}_{out} \circ \mathcal{F}_{2K+4}^{(GFF)} \circ \mathcal{F}_{2K+3}^{(GFF)} \circ \mathcal{F}^{(GSA)} \begin{pmatrix} \boldsymbol{Z}_1 \\ \boldsymbol{O} \end{pmatrix}$$
$$= \mathcal{E}_{out} \circ \mathcal{F}_{2K+4}^{(GFF)} \begin{pmatrix} \boldsymbol{Z}_2 \\ \boldsymbol{O} \end{pmatrix}$$
$$= \begin{pmatrix} \widetilde{G}_1 \left( 3\sum_{p=1}^{d_x}\sum_{q=1}^{n} a_{p,q}\widetilde{\phi}_K(X_{p,q}) \right) & \widetilde{G}_1 \left( 3\sum_{p=1}^{d_x}\sum_{q=1}^{n} a_{p,q}\widetilde{\phi}_K(X_{p,q}) + 2 \right) & \cdots & \widetilde{G}_1 \left( 3\sum_{p=1}^{d_x}\sum_{q=1}^{n} a_{p,q}\widetilde{\phi}_K(X_{p,q}) + 2(n-1) \right) \\ \widetilde{G}_2 \left( 3\sum_{p=1}^{d_x}\sum_{q=1}^{n} a_{p,q}\widetilde{\phi}_K(X_{p,q}) \right) & \widetilde{G}_2 \left( 3\sum_{p=1}^{d_x}\sum_{q=1}^{n} a_{p,q}\widetilde{\phi}_K(X_{p,q}) + 2 \right) & \cdots & \widetilde{G}_2 \left( 3\sum_{p=1}^{d_x}\sum_{q=1}^{n} a_{p,q}\widetilde{\phi}_K(X_{p,q}) + 2(n-1) \right) \\ \vdots & \vdots & & \vdots \\ \widetilde{G}_{d_x} \left( 3\sum_{p=1}^{d_x}\sum_{q=1}^{n} a_{p,q}\widetilde{\phi}_K(X_{p,q}) \right) & \widetilde{G}_{d_x} \left( 3\sum_{p=1}^{d_x}\sum_{q=1}^{n} a_{p,q}\widetilde{\phi}_K(X_{p,q}) + 2 \right) & \cdots & \widetilde{G}_{d_x} \left( 3\sum_{p=1}^{d_x}\sum_{q=1}^{n} a_{p,q}\widetilde{\phi}_K(X_{p,q}) + 2(n-1) \right) \end{pmatrix}.$$

**Step 4:** We now conduct an error analysis. We have

$$\|\mathcal{N} - \boldsymbol{F}\|_{L^p([0,1]^{d_x \times n})}^p$$
$$= \int_{[0,1]^{d_x \times n}} \|\mathcal{N}(\boldsymbol{X}) - \boldsymbol{F}(\boldsymbol{X})\|_F^p \, d\boldsymbol{X}$$
$$\leq \int_{[0,1]^{d_x \times n}} (d_x n)^{\max\{0, \frac{p}{2}-1\}} \sum_{u=1}^{d_x}\sum_{v=1}^{n} |\mathcal{N}_{u,v}(\boldsymbol{X}) - F_{u,v}(\boldsymbol{X})|^p \, d\boldsymbol{X}$$
$$= (d_x n)^{\max\{0, \frac{p}{2}-1\}} \sum_{u=1}^{d_x}\sum_{v=1}^{n} \left( \int_{\boldsymbol{X}: \forall X_{i,j} \in \Omega_K} + \int_{\boldsymbol{X}: \exists X_{i,j} \notin \Omega_K} \right) |\mathcal{N}_{u,v}(\boldsymbol{X}) - F_{u,v}(\boldsymbol{X})|^p \, d\boldsymbol{X}$$
$$=: (d_x n)^{\max\{0, \frac{p}{2}-1\}} \sum_{u=1}^{d_x}\sum_{v=1}^{n} (\mathrm{I} + \mathrm{II}).$$

To estimate I, using that $\widetilde{\phi}_K(X_{p,q}) = \phi_K(X_{p,q})$ when $X_{p,q} \in \Omega_K$, $\widetilde{G}_u$ interpolates $G_{u,v}$ by construction, and $G_{u,v} \in$

$\mathcal{H}^{\frac{\gamma \log 2}{d_x n \log 3}}(\mathcal{C}, 2\sqrt{d_x n} K_{\mathcal{H}})$, we have

$$|\mathcal{N}_{u,v}(\boldsymbol{X}) - F_{u,v}(\boldsymbol{X})|$$

$$= \left| \widetilde{G}_u \left( 3 \sum_{p=1}^{d_x} \sum_{q=1}^{n} a_{p,q} \widetilde{\phi}_K(X_{p,q}) + 2(v-1) \right) - G_{u,v} \left( 3 \sum_{p=1}^{d_x} \sum_{q=1}^{n} a_{p,q} \phi(X_{p,q}) \right) \right|$$

$$= \left| \widetilde{G}_u \left( 3 \sum_{p=1}^{d_x} \sum_{q=1}^{n} a_{p,q} \phi_K(X_{p,q}) + 2(v-1) \right) - G_{u,v} \left( 3 \sum_{p=1}^{d_x} \sum_{q=1}^{n} a_{p,q} \phi(X_{p,q}) \right) \right|$$

$$= \left| G_{u,v} \left( 3 \sum_{p=1}^{d_x} \sum_{q=1}^{n} a_{p,q} \phi_K(X_{p,q}) \right) - G_{u,v} \left( 3 \sum_{p=1}^{d_x} \sum_{q=1}^{n} a_{p,q} \phi(X_{p,q}) \right) \right|$$

$$\leq 2(d_x n)^{\frac{1}{2}} K_{\mathcal{H}} \left| 3 \sum_{p=1}^{d_x} \sum_{q=1}^{n} a_{p,q} \left( \phi_K(X_{p,q}) - \phi(X_{p,q}) \right) \right|^{\frac{\gamma \log 2}{d_x n \log 3}}$$

$$\leq 2(d_x n)^{\frac{1}{2}} K_{\mathcal{H}} \left| 2 \sum_{q=d_x nK+1}^{\infty} 3^{-q} \right|^{\frac{\gamma \log 2}{d_x n \log 3}}$$

$$\leq 2(d_x n)^{\frac{1}{2}} K_{\mathcal{H}} 3^{-\frac{K\gamma \log 2}{\log 3}}$$

$$= 2(d_x n)^{\frac{1}{2}} K_{\mathcal{H}} 2^{-\gamma K},$$

where the second inequality follows from the fact that, as indicated in (12), $3 \sum_{p=1}^{d_x} \sum_{q=1}^{n} a_{p,q} \phi(X_{p,q})$ and $3 \sum_{p=1}^{d_x} \sum_{q=1}^{n} a_{p,q} \phi_K(X_{p,q})$ are both in the Cantor set $\mathcal{C}$ and have the same first $d_x nK$ ternary digits. Thus,

$$\mathrm{I} = \int_{\boldsymbol{X}: \forall X_{i,j} \in \Omega_K} |\mathcal{N}_{u,v}(\boldsymbol{X}) - F_{u,v}(\boldsymbol{X})|^p \mathrm{d}\boldsymbol{X}$$

$$\leq 2^p (d_x n)^{\frac{p}{2}} K_{\mathcal{H}}^p 2^{-p\gamma K}.$$

To estimate II, noting that both $\mathcal{N}_{u,v}$ and $F_{u,v}$ are bounded, and that $\Omega_K$ has Lebesgue measure at least $1 - 2^{-p\gamma K}$, we obtain

$$\mathrm{II} = \int_{\boldsymbol{X}: \exists X_{i,j} \notin \Omega_K} |\mathcal{N}_{u,v}(\boldsymbol{X}) - F_{u,v}(\boldsymbol{X})|^p \mathrm{d}\boldsymbol{X}$$

$$\leq \int_{\boldsymbol{X}: \exists X_{i,j} \notin \Omega_K} \left( \|\mathcal{N}_{u,v}\|_{L^\infty([0,1]^{d_x \times n})} + \|F_{u,v}\|_{L^\infty([0,1]^{d_x \times n})} \right)^p \mathrm{d}\boldsymbol{X}$$

$$\leq \left( \|\mathcal{N}_{u,v}\|_{L^\infty([0,1]^{d_x \times n})} + \|F_{u,v}\|_{L^\infty([0,1]^{d_x \times n})} \right)^p \left( 1 - (1 - 2^{-p\gamma K})^{d_x n} \right)$$

$$\leq 2^{2p} (d_x n)^{\frac{p}{2}+1} K_{\mathcal{H}}^p 2^{-p\gamma K},$$

where we apply Bernoulli's inequality in the last inequality.

We combine the bounds for I and II to obtain

$$\|\mathcal{N} - \boldsymbol{F}\|_{L^p([0,1]^{d_x \times n})}^p \leq (d_x n)^{\max\{0, \frac{p}{2}-1\}} \sum_{u=1}^{d_x} \sum_{v=1}^{n} \left( 2^p (d_x n)^{\frac{p}{2}} K_{\mathcal{H}}^p 2^{-p\gamma K} + 2^{2p} (d_x n)^{\frac{p}{2}+1} K_{\mathcal{H}}^p 2^{-p\gamma K} \right)$$

$$\leq (d_x n)^{3p} 2^{2p} K_{\mathcal{H}}^p 2^{-p\gamma K},$$

where we have used $p \geq 1$ and $\max\{a,b\} \leq a+b$ for all $a, b \geq 0$, which implies

$$\|\mathcal{N} - \boldsymbol{F}\|_{L^p([0,1]^{d_x \times n})} \leq 4(d_x n)^3 K_{\mathcal{H}} 2^{-\gamma K}.$$

Choose $K \geq \frac{1}{\gamma} \log_2 \frac{1}{\varepsilon}$ so that $2^{d_x nK} = \lceil \varepsilon^{-\frac{d_x n}{\gamma}} \rceil$. Then we have

$$\|\mathcal{N} - \boldsymbol{F}\|_{L^p([0,1]^{d_x \times n})} \leq 4(d_x n)^3 K_{\mathcal{H}} \varepsilon,$$

and by (15),

$$\mathcal{N} \in \mathcal{GT}_{d_x,d_x}(D = 4d_x n, H = 1, S = d_x, W \le 3d_x n \lceil \varepsilon^{-\frac{d_x n}{\gamma}} \rceil, L \le 6 \lceil \tfrac{1}{\gamma} \log_2 \tfrac{1}{\varepsilon} \rceil).$$

This completes the proof. $\qquad\square$

