# OpenReview forum: "Approximation Bounds for Transformer Networks with Application to Regression"
_ICML.cc/2026/Conference — ICML 2026 regular_

### Official Review · Reviewer_fwxX · 2026-02-25

**Soundness:** 3
**Presentation:** 3
**Significance:** 3
**Originality:** 3
**Overall Recommendation:** 4
**Confidence:** 4

**Summary:**

This paper investigates the approximation and estimation abilities of Transformer networks with softmax attention.
The authors show that $L^p (p \in [0, \infty])$ approximation bounds for Holder and Sobolev functions.
In particular, their results include the case of $p = \infty$, which is not covered by existing works.
Then, they derive the estimation error bounds beyond i.i.d. data assumption, considering the case of $\beta$-mixing data.

**Compliance With Llm Reviewing Policy:**

Affirmed.

**Final Justification:**

My main concerns are mostly resolved. Thus, I maintain my positive score.

**Key Questions For Authors:**

1. I was unable to understand the message in Section 2.3. What is the purpose of introducing the generalized Transformers at this point?
2. In Theorem 2.3 and 2.4, the approximation bounds for Transformers with softmax attention are given in $L^p$ norm for $p \in [0, \infty]$. However, it is not clear to me why approximation bounds in $L^\infty$ norm is important. Could you please explain the motivation behind this?
3. The authors suggest that the suboptimal estimation error bounds are attributed to the loose bounds on VC dimension of Transformers. Is it possible to obtain better bounds via a different approach? For example, by using the Rademacher complexity instead of VC dimension?

**Limitations:**

yes

**Strengths And Weaknesses:**

**Strengths:**
1. The paper is well-written and well-structured. The authors fairly compare their results with existing works and clearly explain the differences.
2. The paper investigates the theoretical properties of Transformer networks
without making any unrealistic assumptions. For example, the authors consider softmax attention, which is widely used in practice, instead of hard attention, which is more commonly used in theoretical analysis but less practical.
3. Beyond i.i.d. data assumption, the authors consider the case of $\beta$-mixing data, which is more general and realistic in practice.

**Weaknesses:**
1. One of the main contributions of this paper is the approximation bounds for Transformer networks with softmax attention in $L^\infty$ norm. While theoretically interesting, it is not clear to me why approximation bounds in $L^\infty$ norm is important.
2. Estimation error bounds in this paper is not optimal as the authors themselves admit.
3. Estimation error bounds for Transformers are the same or worse than those for feedforward neural networks. Therefore, it does not explain the empirical success of Transformers in practice.

---

> ### Author Rebuttal · Authors · 2026-03-29
>
> We thank you for your thorough review of our manuscript and for your constructive suggestions. Our point-by-point responses to your comments are given below.
>
> > **Q1** I was unable to understand the message in Section 2.3. What is the purpose of introducing the generalized Transformers at this point?
>
> **Response** Thank you for raising this point. The goal of Section 2.3 is to present a new proof strategy for Transformer approximation, inspired by the Kolmogorov-Arnold representation theorem. The key observation is that our argument relies only on property of self-attention that, under a suitable choice of parameters, it can implement column-wise averaging. Once this property is isolated, the rest of the proof no longer depends strongly on the specific analytic form of the softmax. We believe this averaging property might be fundamental and might also be shared by other attention variants. For this reason, we introduce generalized Transformers. We will revise the paper to make the motivation more explicit.
>
> > **Q2** In Theorem 2.3 and 2.4, the approximation bounds for Transformers with softmax attention are given in $L^p$ norm for $p\in[0, \infty]$. However, it is not clear to me why approximation bounds in $L^\infty$ norm is important. Could you please explain the motivation behind this?
>
> **Response** Thank you for this important question. We believe that approximation in $L^\infty$-norm is a more difficult problem, and is very useful in statistical analysis. First, $L^\infty$-norm is stronger than $L^p$-norm on a bounded domain for every $1 \leq p < \infty$. Thus, $L^\infty$-approximation implies $L^p$-approximation, whereas the converse does not hold in general. Second, if one aims to study $L^\infty$-convergence or sup-norm risk bounds, then $L^\infty$-approximation is necessary.
>
> > **Q3**  The authors suggest that the suboptimal estimation error bounds are attributed to the loose bounds on VC dimension of Transformers. Is it possible to obtain better bounds via a different approach? For example, by using the Rademacher complexity instead of VC dimension?
>
> **Response** Thank you very much for this valuable question.
>
> First, we acknowledge that our VC-dimension bound may not be optimal. To the best of our knowledge, for network architectures involving exponential computations, even for standard sigmoid FNNs, the best known upper and lower bounds do not match. Obtaining a tighter VC-dimension bound would likely require an improvement of the existing proof techniques, which appears to be highly challenging.
>
> Second, there is another line of work that imposes explicit magnitude constraints on the network parameters and then bounds the covering number of the function class through a covering of the parameter space, without passing through VC-dimension; see, e.g., [1]. We believe this approach may be a promising direction for Transformers.
>
> [1] Johannes Schmidt-Hieber. Nonparametric regression using deep neural networks with relu activation function. The Annals of Statistics, 2020.
>
> > **Q4** One of the main contributions of this paper is the approximation bounds for Transformer networks with softmax attention in $L^\infty$ norm. While theoretically interesting, it is not clear to me why approximation bounds in $L^\infty$ norm is important.
>
> We refer the reviewer to our response to Q2.
>
> > **Q5** Estimation error bounds in this paper is not optimal as the authors themselves admit.
>
> We refer the reviewer to our response to Q3.
>
> > **Q6** Estimation error bounds for Transformers are the same or worse than those for feedforward neural networks. Therefore, it does not explain the empirical success of Transformers in practice.
>
> **Response** Thank you for your valuable comment. We acknowledge that our bounds for Transformers are currently not as sharp as the best known ones for FNNs. However, understanding such a complex model is itself a fundamental and challenging problem that is receiving broad attention, and we believe our work provides new insight into this problem.
>
> First, many existing theoretical works replace standard softmax attention with more mathematically tractable alternatives. By contrast, our bounds are established directly for standard softmax-activated Transformers.
>
> Second, while there have been some prior results on the approximation power of softmax attention, our work provides explicit approximation rates for a broad class of functions and extends the analysis to $L^\infty$-approximation, which previous approaches did not address.
>
> Third, we provide a careful analysis of statistical efficiency within a statistical learning framework, which has been less studied in the literature.
>
> Therefore, our goal is not to claim that the present theory fully explains the practical success of Transformers, but rather to make rigorous progress on an important and difficult problem. We believe that this already constitutes a meaningful step beyond the existing literature.

---

> > ### Author Rebuttal · Reviewer_fwxX · 2026-04-01
> >
> > Thank you for the clarification. Most of my concerns have been resolved.
> > Regarding Q2, my question is: in what situations are error bounds in the $L^\infty$ necessary?
> > Could you explain in what cases they are “very useful in statistical analysis”?

---

> > > ### Author Response · Authors · 2026-04-04
> > >
> > > Thank you for your effort in reviewing our work. Regarding the further question, we respond as follows.
> > >
> > > > Regarding Q2, my question is: in what situations are error bounds in the $L^\infty$ necessary? Could you explain in what cases they are "very useful in statistical analysis"?
> > >
> > > **Response** Thank you for the clarification. Approximation in the $L^\infty$-norm is useful in several statistical problems, including $L^\infty$-convergence and sup-norm risk bounds [1,2], as well as the construction of confidence bands [3]. The reason is that the $L^\infty$-norm guarantees uniform control over the whole domain, ensuring that every point is well approximated. Moreover, an $L^\infty$-approximation bound immediately yields uniform boundedness of the constructed approximant via the triangle inequality, whereas $L^p$-approximation bounds alone do not provide such control in general, since a function may have large spikes on sets of small measure. This boundedness property is often useful in statistical learning arguments, where one frequently works with hypothesis classes that are assumed to be uniformly bounded.
> > >
> > >
> > > [1] Masaaki Imaizumi. Sup-norm convergence of deep neural network estimator for nonparametric regression by adversarial training. arXiv, 2023.
> > >
> > > [2] Bin Yu. Density estimation in the $L^\infty$ norm for dependent data with applications to the gibbs sampler. The Annals of Statistics, 1993.
> > >
> > > [3] Tony Cai, Mark Low, and Zongming Ma. Adaptive confidence bands for nonparametric regression functions. JASA, 2014.
> > >
> > > If you have any questions, please feel free to further contact us. Thank you once again for your careful reading, professional suggestions, and inspiring recognitions.

---

### Official Review · Reviewer_kiig · 2026-03-12

**Soundness:** 3
**Presentation:** 3
**Significance:** 3
**Originality:** 3
**Overall Recommendation:** 5
**Confidence:** 3

**Summary:**

This paper presents a rigorous theoretical framework establishing approximation bounds and statistical excess-risk guarantees for standard softmax-activated Transformer networks. The paper proves that a fixed-depth Transformer can achieve an $\epsilon$ approximation error in $L^p$-norms (crucially including the $p = \infty$ endpoint) for sequence-to-sequence $\gamma$-H\"older continuous functions with $\mathcal{O}(\epsilon^{-d_x n / \gamma})$ parameters. A similar scaling of $\mathcal{O}(\epsilon^{-d_x n})$ is derived for first-order Sobolev functions. Moreover, the authors apply these approximation bounds to a nonparametric regression setting with temporally dependent data (stationary $\beta$-mixing sequences). The proof strategy relies on a novel reduction inspired by the Kolmogorov-Arnold representation theorem, framing the self-attention layer as a column-wise averaging operator.

**Compliance With Llm Reviewing Policy:**

Affirmed.

**Final Justification:**

Most of my concerns have been resolved and I maintain my recommendation.

**Key Questions For Authors:**

1. The suboptimality of your convergence rates stems from the $\mathcal{O}(N^2)$ VC-dimension bound, which arises due to the exponential function in the standard softmax attention (via Anthony & Bartlett). Do you view this quadratic growth in VC-dimension as a fundamental statistical inefficiency inherent to standard softmax attention compared to ReLU/hardmax variants, or do you suspect it is simply an artifact of loose bounding techniques in current learning theory?

2. Can your generalized attention framework be extended to construct piecewise-linear or higher-order polynomial approximations to adapt to higher-order smoothness?

**Limitations:**

Yes.

**Strengths And Weaknesses:**

## Strengths

- The paper provides explicit approximation rates for standard softmax-activated Transformers over Hölder and Sobolev functions, crucially including the previously elusive $L^\infty$-norm endpoint.

- Unlike much of the existing literature which relies on modifying the attention mechanism (e.g., substituting softmax with ReLU or hardmax) to make the mathematics tractable, this paper directly addresses the standard softmax-activated architecture used in practice.

- The established approximation rates match the best-known bounds for ambient dimensions previously achieved by fixed-depth Feed-Forward Neural Networks (FNNs) and Recurrent Neural Networks (RNNs).

- The paper also develops a unified excess-risk analysis for nonparametric regression using Transformers under the realistic assumption of temporally dependent data (stationary $\beta$-mixing sequences). This aligns well with the typical use cases of sequence modeling.


## Weaknesses

- In Appendix A, the proof of Theorem 2.4 bounding the approximation error within the trifling region $\Omega$ relies on an erroneous assumption. The authors assert that the target function $F \in \mathcal{W}^{1,p}$ implies $||\mathcal{N}(X)|| \leq \sqrt{d_xn}K_{\mathcal{W}}$, allowing them to bound the integral $\int_{\Omega} ||\mathcal{N}(X) - F(X)||_F^p dX$. However, by the Sobolev embedding theorems, functions in $\mathcal{W}^{1,p}$ are not globally bounded in $L^\infty$ when $p \leq d_x n$. They can possess local singularities. Therefore, applying a universal uniform bound is mathematically invalid for general $p < \infty$.

    The theorem itself remains true and can be patched. The authors should revise the proof to utilize the absolute continuity of the Lebesgue integral. Since $F, \mathcal{N} \in L^p$, the integral over the trifling region $\Omega$ approaches zero as the measure of the region (controlled by $\delta$) goes to zero. By choosing $\delta$ to be sufficiently small relative to the grid size $K$, the error can be suppressed without requiring an $L^\infty$ assumption.

- As the authors acknowledge, the derived rates of convergence for nonparametric regression are suboptimal compared to the known minimax optimal rates for i.i.d. and certain $\beta$-mixing data.

- The suboptimality in the convergence rates stems from the loose upper bound on the VC-dimension. Because the standard Transformer uses the exponential function in the softmax attention, the best-known VC-dimension bounds grow quadratically with the number of parameters ($\mathcal{O}(N^2)$), rather than linearly as seen in ReLU networks. The paper does not resolve whether this quadratic growth is a fundamental property or simply a limitation of current VC-dimension bounding techniques.

- The proof relies on approximating the target function with piecewise-constant functions on a uniform grid. This discretization technique creates a bottleneck due to the "saturation phenomenon," preventing the authors from obtaining faster approximation rates even if the target function possesses **higher-order smoothness** (regularity beyond first-order Sobolev or $\gamma \in (0, 1]$ Hölder).

---

> ### Author Rebuttal · Authors · 2026-03-30
>
> We thank you for your thorough review of our manuscript and for your constructive suggestions. Our point-by-point responses to your comments are given below.
>
> > **Q1** The suboptimality of your convergence rates stems from the $\mathcal{O}(N^2)$ VC-dimension bound, which arises due to the exponential function in the standard softmax attention (via Anthony \& Bartlett). Do you view this quadratic growth in VC-dimension as a fundamental statistical inefficiency inherent to standard softmax attention compared to ReLU/hardmax variants, or do you suspect it is simply an artifact of loose bounding techniques in current learning theory?
>
> **Response** Thank you for this insightful question. In our view, the answer may depend on whether the current upper bound is close to being tight. If one could prove a matching lower bound on the VC-dimension, then the observed suboptimality would indeed suggest a fundamental statistical inefficiency of standard softmax attention. Since the VC-dimension measures the complexity of the underlying function class, a larger quantity would indicate that softmax-activated Transformer classes are intrinsically more complex than the ReLU and hardmax variants, and hence potentially harder to estimate efficiently. On the other hand, if the current upper bound is not tight, then the suboptimality should be viewed as an artifact of existing bounding techniques. At present, however, the situation remains unclear. To the best of our knowledge, for network architectures involving exponential computations, even for standard sigmoid FNNs, the best known upper and lower bounds on the VC-dimension do not match. Obtaining a tighter VC-dimension bound would likely require an improvement of the existing proof techniques, which appears to be highly challenging.
>
> > **Q2** Can your generalized attention framework be extended to construct piecewise-linear or higher-order polynomial approximations to adapt to higher-order smoothness?
>
> **Response** Thank you very much for this valuable question. In the proof of Theorem 2.8, our argument relies on the Kolmogorov-Arnold representation $f(x_1, \ldots, x_d) = g(3 \sum_{p=1}^d 3^{-p} \phi(x_p))$ for any $d$-variate function $f$. Although this representation allows the transfer of smoothness properties of the represented function $f$ to the outer function $g$ for $\gamma$-Hölder $f$ with $\gamma \in (0,1]$, it remains unclear whether this representation can be generalized to higher order smoothness.
>
> > **Q3** In Appendix A, the proof of Theorem 2.4 bounding the approximation error within the trifling region $\Omega$ relies on an erroneous assumption. The authors assert that the target function $F\in\mathcal{W}^{1,p}$ implies $\|\mathcal{N}(X)\| \leq \sqrt{d_x n} K_{\mathcal{W}}$, allowing them to bound the integral $\int_{\Omega} \|\mathcal{N}(X)-F(X)\|_F^p d X$. However, by the Sobolev embedding theorems, functions in $\mathcal{W}^{1,p}$ are not globally bounded in $L^\infty$ when $p \leq d_x n$. They can possess local singularities. Therefore, applying a universal uniform bound is mathematically invalid for general $p<\infty$.
> >
> > The theorem itself remains true and can be patched. The authors should revise the proof to utilize the absolute continuity of the Lebesgue integral. Since $F, \mathcal{N} \in L^p$, the integral over the trifling region $\Omega$ approaches zero as the measure of the region (controlled by $\delta$) goes to zero. By choosing $\delta$ to be sufficiently small relative to the grid size $K$, the error can be suppressed without requiring an $L^\infty$ assumption.
>
> **Response** Thank you very much for your valuable comment and helpful suggestion. We will carefully revise the manuscript and check the proofs accordingly.
>
> > **Q4**
> > - As the authors acknowledge, the derived rates of convergence for nonparametric regression are suboptimal compared to the known minimax optimal rates for i.i.d. and certain $\beta$-mixing data.
> > - The suboptimality in the convergence rates stems from the loose upper bound on the VC-dimension.
>
> We refer the reviewer to our response to Q1.
>
> > **Q5** The proof relies on approximating the target function with piecewise-constant functions on a uniform grid. This discretization technique creates a bottleneck due to the "saturation phenomenon," preventing the authors from obtaining faster approximation rates even if the target function possesses higher-order smoothness (regularity beyond first-order Sobolev or $\gamma \in(0,1]$ Hölder).
>
> **Response** Thank you for your valuable comment. We agree with this point. Extending the approximation analysis to accommodate higher order smoothness, for example via partitions of unity and multiplication constructions, is an interesting and challenging direction for our future research.
>
> If you have any questions, please feel free to further contact us. Thank you once again for your careful reading, professional suggestions, and inspiring recognitions.

---

> > ### Author Rebuttal · Reviewer_kiig · 2026-04-03
> >
> > Thank you for your response. I believe my initial assessment already reflects my evaluation, and I will maintain my initial score.

---

### Official Review · Reviewer_kL8d · 2026-03-12

**Soundness:** 3
**Presentation:** 3
**Significance:** 3
**Originality:** 3
**Overall Recommendation:** 5
**Confidence:** 3

**Summary:**

Prior universality results for Transformers with sigmoidal activations show that they can approximate continuous sequence-to-sequence maps, but these results typically do not provide quantitative estimates on how the approximation accuracy scales with the network size. This paper studies approximation rates for Transformers when the target function has additional regularity, specifically Holder or Sobolev regularity. The authors show that Transformers with sigmoidal activations and fixed depth can approximate such targets with explicit bounds on the required width (and hence parameter count) as a function of the desired accuracy. The paper also extends these approximation results to a broader class of generalized Transformers that allow more flexible activation functions and biases.

In addition to the approximation results, the paper analyzes a regression setting where data are generated from a stationary beta-mixing process. The authors derive excess risk bounds showing that the estimation error decreases with the sample size. Combined with the approximation results, this implies that the learned Transformer converges to the target function at a rate determined by both the number of samples and the regularity of the target.

Overall, the paper establishes quantitative bounds on both the network size required to approximate smooth sequence-to-sequence maps and the sample complexity required to learn such models from data.

**Compliance With Llm Reviewing Policy:**

Affirmed.

**Key Questions For Authors:**

- (Q1) What were the main technical contributions in the proofs of the main approximation theorems?

**Limitations:**

Yes.

**Strengths And Weaknesses:**

The paper is generally well written and the technical arguments appear carefully developed. In addition to extending existing universality results, the work provides explicit parameter scaling estimates and statistical risk bounds, which are valuable contributions to the theoretical understanding of Transformers. In particular, it shows that the transformers in question can be obtained through training.

The paper states that approximation in the uniform norm (i.e. $L^\infty$) is a key contribution. However, uniform approximation for Transformers over continuous functions has already been established in "Transformers are Universal In-Context Learners" (de Hoop, Furuya, Peyre), where approximation is shown for infinite contexts. However, that work does not provide quantitative estimates on how the number of parameters scales with the desired accuracy, nor does it establish statistical learning guarantees. The present paper shows that once additional regularity assumptions (Holder or Sobolev) are imposed on the target function, one can obtain explicit parameter scaling and risk bounds. Stated this way, the results provide a meaningful and complementary addition to the existing theoretical literature.

In a related point, Table 1 lists several prior results that already establish uniform approximation for similar function classes. From reading the results more carefully, it appears that the main distinction is architectural: previous works typically require both width and depth to scale with the desired accuracy, whereas the results in this paper achieve the approximation rate using fixed depth and increasing width. In addition, the present work focuses on sigmoidal activations that are commonly used in practice. Clarifying this distinction explicitly would make the contribution clearer to readers.

The importance of the approximation result for generalized transformers (Theorem 2.8) could be elaborated on.

The following are minor comments that the authors may or may not want to follow to improve readibility:
- it could make Theorem 2.7 more clear if the list, 1. 2. 3., is replace with more meaningful titles, e.g. "geometric beta-mixing:", "algebraic ...", etc.
- in a similar spirit, it would be nice to include titles in theorems, e.g. "approximation for generalized transformers" in Theorem 2.8. It easy to miss the difference between Theorem 2.8 and the prior approximation theorems on a quick read.

---

> ### Author Rebuttal · Authors · 2026-03-30
>
> We thank you for your thorough review of our manuscript and for your constructive suggestions. Our point-by-point responses to your comments are given below.
>
> > **Q1** What were the main technical contributions in the proofs of the main approximation theorems?
>
> **Response** Thank you for this important question.
>
> First, in Theorems 2.3 and 2.4, we derive explicit approximation rates for a broad class of target functions and extend existing $L^p$-approximation results to the $L^\infty$ setting. The former follows from our analysis on parameter scaling. For the latter, the previous technical difficulty is that continuous piecewise linear functions cannot approximate a discontinuous step function uniformly well. We use the horizontal shift technique to show that, as long as good uniform approximation by a Transformer can be obtained outside the mismatching region, the uniform approximation error can also be well controlled inside that region when the network size is slightly increased.
>
> Second, in Theorem 2.8, we use the Kolmogorov-Arnold representation to provide a potentially new perspective on the role of softmax attention. Specifically, every $d$-variate continuous function admits a representation of the form $f(x_1, \ldots, x_d) = g(3 \sum_{p=1}^d 3^{-p} \phi(x_p))$. We use feed-forward layers to approximate the inner function $\phi$ and the outer function $g$, while the self-attention layer is reduced to implement column-wise summation. To the best of our knowledge, this construction has not appeared in the previous literature and constitutes a technical novelty of our proof.
>
>
> > **Q2** The paper states that approximation in the uniform norm (i.e. $L^\infty$) is a key contribution. However, uniform approximation for Transformers over continuous functions has already been established in "Transformers are Universal In-Context Learners" (de Hoop, Furuya, Peyre), where approximation is shown for infinite contexts. However, that work does not provide quantitative estimates on how the number of parameters scales with the desired accuracy, nor does it establish statistical learning guarantees. The present paper shows that once additional regularity assumptions (Holder or Sobolev) are imposed on the target function, one can obtain explicit parameter scaling and risk bounds. Stated this way, the results provide a meaningful and complementary addition to the existing theoretical literature.
>
> **Response** Thank you very much for recognizing the value of our work and for pointing out this relevant reference. We will cite this paper in the revision and clarify the distinction from our results more carefully.
>
> > **Q3** In a related point, Table 1 lists several prior results that already establish uniform approximation for similar function classes. From reading the results more carefully, it appears that the main distinction is architectural: previous works typically require both width and depth to scale with the desired accuracy, whereas the results in this paper achieve the approximation rate using fixed depth and increasing width. In addition, the present work focuses on sigmoidal activations that are commonly used in practice. Clarifying this distinction explicitly would make the contribution clearer to readers.
>
> **Response** Thank you for your helpful suggestion. This is indeed the contribution we intended to highlight. We will make the architectural distinction more explicit in the revision so as to clarify our contribution.
>
> > **Q4** The importance of the approximation result for generalized transformers (Theorem 2.8) could be elaborated on.
>
> **Response** Thank you for your helpful comment. This point is indeed our theoretical contribution. A challenge in understanding the expressivity of Transformers is to explain why the self-attention layer can effectively capture complex token-wise interactions, since self-attention is the only component in the architecture that explicitly aggregates information across tokens. In Theorem 2.8, we show that column-wise averaging alone already suffices to establish nontrivial approximation properties. Thank you for your understanding. We will revise the paper to make the motivation and contribution more explicit.
>
> > **Q5**
> > - It could make Theorem 2.7 more clear if the list, 1. 2. 3., is replace with more meaningful titles, e.g. "geometric beta-mixing:", "algebraic ...", etc.
> > - In a similar spirit, it would be nice to include titles in theorems, e.g. "approximation for generalized transformers" in Theorem 2.8. It easy to miss the difference between Theorem 2.8 and the prior approximation theorems on a quick read.
>
> **Response** Thank you very much for your valuable suggestion. We will revise the manuscript accordingly to improve readability.
>
> If you have any questions, please feel free to further contact us. Thank you once again for your careful reading, professional suggestions, and inspiring recognitions.

---

> > ### Author Rebuttal · Reviewer_kL8d · 2026-04-03
> >
> > Thank you for the response, I maintain my recommendation.

---

### Official Review · Reviewer_nXBF · 2026-03-13

**Soundness:** 3
**Presentation:** 4
**Significance:** 3
**Originality:** 4
**Overall Recommendation:** 5
**Confidence:** 4

**Summary:**

The proposed submission develops novel approximation and statistical theory for standard softmax-activated Transformer networks in
sequence modeling, establishing explicit $L^p$-approximation bounds (including the challenging $p=\infty$ endpoint) for target
sequence-to-sequence mappings in Hölder and Sobolev classes. This demonstrates that achieving an error of $\epsilon$ requires
$\mathcal{O}(\epsilon^{-d_xn/\gamma})$ parameters for $\gamma$-Hölder functions, while also extending the analysis to nonparametric
regression with temporally dependent observations using a sliding-window empirical risk minimization procedure, deriving explicit
excess-risk guarantees. A core contribution is the introduction of a proof strategy inspired by the Kolmogorov-Arnold representation
theorem, which abstracts self-attention layers as column-wise averaging operators and simplifies the analysis to enable extensions to
beyond-softmax attention mechanisms.

**Compliance With Llm Reviewing Policy:**

Affirmed.

**Key Questions For Authors:**

**Addressing Optimality Gap:**

1.  **Tighter VC-Dimension Bound:** Do you believe it's possible to obtain a tighter VC-dimension bound for Transformers, or would an
alternative capacity measure be more suitable?
2.  **Fundamental Capacity Measures:** Would using Rademacher complexity or covering numbers as the capacity measure help close the
optimality gap?

**Circumventing Curse of Dimensionality:**

1.  **Low-Dimensional Manifolds:** Have you considered extending the analysis to target functions defined on low-dimensional
manifolds?
2.  **Compositional Structures:** Could you explore target functions with specific compositional structures, which could
theoretically mitigate the curse of dimensionality?

**Self-Attention Expressiveness:**

1.  **Dynamic Routing Benefits:** Does the column-wise averaging abstraction overlook some benefits of dynamic, data-dependent routing
in self-attention?
2.  **Potential Overlooked Advantages:** Are there any other advantages that self-attention provides over static averaging that might
be relevant to the analysis?

**Limitations:**

yes

**Strengths And Weaknesses:**

**Strengths:**
* The proof strategy inspired by the Kolmogorov-Arnold representation theorem is highly creative and allows for a clean decoupling of
approximation capabilities from specific nonlinearities, making the bounds robust and easily extendable to other attention variants.

* Providing rigorous $L^\infty$-norm approximation bounds for standard softmax-activated Transformers addresses a notable gap in the
theoretical deep learning literature. Analyzing statistical performance under temporally dependent sequences aligns with the primary
practical use-case of Transformers, moving beyond restrictive i.i.d. assumptions.

* Theoretical results are clearly formalized, and the progression from basic architecture definitions to function class approximation,
and finally to statistical excess-risk bounds provides a cohesive narrative.

**Weaknesses:**
* Statistical rates derived for nonparametric regression are suboptimal compared to known minimax rates for i.i.d. data due to the
reliance on a loose upper bound for the VC-dimension, weakening excess-risk guarantees.

* Approximation and statistical bounds exhibit an exponential dependence on ambient sequence dimension $d_x n$, failing to explain why Transformers succeed remarkably in high-dimensional real-world tasks that likely require analyzing target functions with
low-dimensional structural priors.

* The paper is purely theoretical, and a small-scale empirical study verifying the tightness of approximation bounds or observing
behavior under various $\alpha$-mixing conditions would make practical implications more compelling.

---

> ### Author Rebuttal · Authors · 2026-03-30
>
> We thank you for your thorough review of our manuscript and for your constructive suggestions. Our point-by-point responses to your comments are given below.
>
> > **Q1** Tighter VC-Dimension Bound: Do you believe it's possible to obtain a tighter VC-dimension bound for Transformers, or would an alternative capacity measure be more suitable?
>
> **Response** Thank you for this insightful question. Regarding the first point, in the paper we only derive an upper bound on the VC-dimension of Transformers, following existing techniques, and we conjecture that this upper bound is not tight. To the best of our knowledge, for network architectures involving exponential computations, even for standard sigmoid FNNs, the best known upper and lower bounds do not match. Obtaining a tighter VC-dimension bound would likely require an improvement of the existing proof techniques, which appears to be highly challenging. Regarding the second point, we refer the reviewer to our response to Q2 below.
>
> > **Q2** Fundamental Capacity Measures: Would using Rademacher complexity or covering numbers as the capacity measure help close the optimality gap?
>
> **Response** Thank you very much for this valuable question.
>
> First, using covering numbers directly as the complexity measure may potentially lead to improved results. In particular, there is another line of work that imposes explicit magnitude constraints on the network parameters and then bounds the covering number of the function class through a covering of the parameter space, without passing through VC-dimension; see, e.g., [1]. We believe this approach may be a promising direction for Transformers.
>
> Second, regarding Rademacher complexity, the commonly used approach is to bound it by covering numbers and, in turn, VC-dimension. Using Rademacher complexity may not in itself lead to an essential improvement.
>
> [1] Johannes Schmidt-Hieber. Nonparametric regression using deep neural networks with relu activation function. The Annals of Statistics, 2020.
>
> > **Q3** Low-Dimensional Manifolds: Have you considered extending the analysis to target functions defined on low-dimensional manifolds?
>
> **Response** Thank you for this interesting question. We believe that extending the analysis to target functions defined on low-dimensional manifolds is feasible, following existing techniques. A possible route is to combine local decomposition, local approximation, and a patching argument to obtain a global result. Ideally, the ambient dimension in the approximation bounds could be replaced by the intrinsic dimension of the manifold, thereby mitigating the curse of dimensionality. Understanding how to overcome the curse of dimensionality is an important problem and part of our ongoing research.
>
> > **Q4** Compositional Structures: Could you explore target functions with specific compositional structures, which could theoretically mitigate the curse of dimensionality?
>
> **Response** Thank you for this insightful question. We agree that compositional structure is another important way to mitigate the curse of dimensionality, and we believe that extending our analysis in this direction may be feasible.
>
> > **Q5** Dynamic Routing Benefits: Does the column-wise averaging abstraction overlook some benefits of dynamic, data-dependent routing in self-attention?
>
> **Response** Thank you for this insightful question. We agree that self-attention may have additional properties beyond column-wise averaging that could potentially lead to stronger results. In our paper, we show that column-wise averaging alone already suffices to establish nontrivial approximation properties. Further understanding and exploiting the finer structure of self-attention is an interesting direction for future work.
>
> > **Q6** Potential Overlooked Advantages: Are there any other advantages that self-attention provides over static averaging that might be relevant to the analysis?
>
> **Response** Thank you for this thoughtful question. Self-attention may have additional structural advantages, and exploring them is an interesting direction for future research.
>
> > **Q7** The paper is purely theoretical, and a small-scale empirical study verifying the tightness of approximation bounds or observing behavior under various $\alpha$-mixing conditions would make practical implications more compelling.
>
> **Response** Thank you for your valuable suggestion. We will include small-scale regression experiments as a complement to the theory.
>
>
> If you have any questions, please feel free to further contact us. Thank you once again for your careful reading, professional suggestions, and inspiring recognitions.

---

### Decision · Program_Chairs · 2026-04-30

**Decision:**

Accept (regular)

**Comment:**

This paper proves bounds on the approximation error of transformers when estimating Holder continuous or Sobolev sequence to sequence functions. In addition, they also prove excess risk bounds for non-parametric regression with a single long input sequence if either i.i.d., $\beta$ geometrically mixing or $\alpha$ algebraically mixing tokens, generalizing results from [1,2,3]. The proof strategy via the Kolmogorov Alrnold Representation theorem appears distinct from the previous literature, strengthening the approximation results.



The reviewers are **unanimously positive**, praising the clarity of the formalism (nXBF)  and proof techniques (kL8d :“the technical arguments appear carefully developed”), the originality of the proof technique and the strength of the results, which manage to match the known DNN results exactly (Rev. kiig).


On the other hand, as reviewers nXBF and kiig (and to lesseer extent, reviewer fwxX: “the estimation error bounds are not as tight as the authors admit”) both mention, the **VC dimension upper bound on transformers from Lemmas A.11 and A.12 appear extremely loose** (worse than what would be expected from parameter counting, as the dependence on the number of layers is quadratic). I do think this is a significant and unnecessary limitation: it is likely not too hard to achieve much tighter bounds with a dedicated covering number approach (cf. eg. [4] for CNNs and [6] for rudimentary transformer architectures). Reviewer kL8d also mentions that the particular case $p=\infty$ has already been covered in [7], which should affect the positioning in the camera-ready revision.



Overall, the paper makes a **meaningful contribution** with a highly non trivial and **innovative proof technique**, and fully clearly deserves a place in the ICML programme.


**References**:

[1] Jiao, Y., Lai, Y., Wang, Y., and Yan, B. Convergence analysis of flow matching in latent space with transformers

 [2] Jiao, Y., Wang, Y., and Yan, B. Approximation bounds for recurrent neural networks with application to regression.

[3] Ren, Y., Lu, Y., Ying, L., and Rotskoff, G. M. Statistical spatially inhomogeneous diffusion inference


[4] Long and Sedhi. Generalization bounds for Convolutional Neural Networks.
generally well written and the technical arguments appear carefully developed


[5] Ledent et al. Generalization bounds for rank sparse neural networks.

[6] Zhang, Y., Liu, B., Cai, Q., Wang, L., and Wang, Z. (2022). An analysis of attention via the lens of exchangeability and latent variable models

[7] Transformers are Universal In-Context Learners.